# Arctic Delta Reduced Complexity Model and its Reproduction of Key Geomorphological Structures

Ngai-Ham Chan[1,2], Moritz Langer[2,3,4], Bennet Juhls[2], Tabea Rettelbach[2,5], Paul Overduin[2], Kimberly Huppert[1], and Jean Braun[1]

[1]German Research Centre for Geosciences GFZ, Telegrafenberg, 14473 Potsdam, Germany
[2]Alfred Wegener Institute for Polar and Marine Research, Telegrafenberg, 14473 Potsdam, Germany
[3]Geography Department, Humboldt-Universität zu Berlin, Rudower Chaussee 16, 12489 Berlin, Germany
[4]Department of Earth Sciences, Vrije Universiteit Amsterdam, De Boelelaan 1085, 1081 HV Amsterdam, The Netherlands
[5]Institute of Geosciences, University of Potsdam, Karl-Liebknecht-Str. 24-25, 14476 Potsdam, Germany

**Correspondence:** Ngai-Ham "Erik" Chan (echan@gfz-potsdam.de)

**Abstract.** Arctic river deltas define the interface between the terrestrial Arctic and the Arctic Ocean. They are the site of sediment, nutrient, and soil organic carbon discharge to the Arctic Ocean. Arctic deltas are unique globally because they are underlain by permafrost, acted on by river and sea ice, and many are surrounded by a broad shallow ramp. Such ramps may buffer the delta from waves, but as the climate warms and permafrost thaws, the evolution of Arctic deltas will likely take a different course, with implications both local in scale and on the wider Arctic Ocean. One important way to understand and predict the evolution of Arctic deltas is through numerical models. Here we present ArcDelRCM.jl, an improved reduced complexity model (RCM) of arctic delta evolution based on the DeltaRCM-Arctic model (Lauzon et al., 2019), which we have reconstructed in Julia language using published information. Unlike previous models, ArcDelRCM.jl is able to replicate the ramp around the delta. We have found that the delayed breakup of the so-called "bottom-fast ice" (i.e., ice that is in direct contact with the bed of the channel or the sea, also known as "bed-fast ice") on and around the deltas is ultimately responsible for the appearance of the ramp feature in our models. However, changes made to the modelling of permafrost erosion and protective effects of bottom-fast ice are also important contributors. Graph analyses of the delta network performed on ensemble runs show that deltas produced by ArcDelRCM.jl have more interconnected channels and contain less abandoned subnetworks. This may suggest a more even feeding of sediments to all sections of the delta shoreline, supporting ramp growth. Moreover, we showed that the morphodynamic processes during the summer months remain active enough to contribute significant sediment input to the growth and evolution of Arctic deltas, thus should not be neglected in simulations gauging the multi-year evolution of delta features. Finally, we tested a strong climate-warming scenario on the simulated deltas of ArcDelRCM.jl, with temperature, discharge, and ice conditions consistent with RCP 7-8.5. We found that the ramp features degrade on the time scale of centuries and effectively disappear in under a millennium. Ocean processes, which are not included in these models, may further shorten the time scale. With the degradation of the ramps, any dissipative effects on wave energy they offered would also decrease. This could expose the sub-aerial parts of the deltas to increased coastal erosion, thus impacting permafrost degradation, nutrients and carbon releases.

# 1 Introduction

Arctic deltas are key components of the permafrost landscape, connecting the permafrost areas upstream and the Arctic Ocean. They act as records and filters of the particulate and dissolved matter, such as sediments and nutrients, that originated from the Arctic and sub-Arctic regions — regions that contain a substantial portion of the Earth's soil organic carbon (Hugelius et al., 2014; Schuur et al., 2015) — which could potentially exacerbate climate warming through positive feedback. As permafrost thaws and the Arctic ocean trends towards being free of ice, especially under amplified warming in polar regions (Stocker, 2014; Rantanen et al., 2022), Arctic deltas face an uncertain future. On a local level, the ecosystems surrounding Arctic deltas will also face significant impacts as a result of climate change (Pisaric et al., 2011). The largest Arctic delta is the Lena Delta, which has fan-shaped morphology with multiple channels leading discharge from its epicentre to the delta edge. Although complicated by neotectonics (Are and Reimnitz, 2000), its large-scale structure is probably determined by regional relative sea level effects on fluvial sediment budgets within the delta and at its edge (Whitehouse et al., 2007).

An important feature ubiquitous to Arctic deltas is the 2-m ramps, which we will interchangeably refer to as "ramp features", or simply "ramps" (Are and Reimnitz, 2000). Figure 1 shows such a ramp feature surrounding the Lena Delta, which dips gradually from below the sea surface to roughly 2-metre depth, but with localised variations on the order of a metre (Fuchs et al., 2021). These ramps extend from the above-water shoreline of Arctic deltas over tens of kilometres towards the open ocean (Reimnitz, 2002). The shallow inclination of the ramp serves to diffuse wave energy off-shore and may protect the delta from direct wave impact. The shallow water depth of the ramp corresponds to the range of maximum winter ice thickness (Wegner et al., 2017), meaning that ice freezes to the seabed at some point during winter, which is called "bottom-fast" or "bed-fast" ice (Dammann et al., 2018, 2019; Eicken et al., 2005), and that a seasonally frozen layer can develop in the sediment. When seasonal freezing exceeds thawing in subsequent years, permafrost can develop below the seabed (Osterkamp et al., 1989). Land- or bottom-fast ice and seasonal or permanent freezing may all act to stabilise the ramp (Overeem et al., 2022). Therefore, the ramp features, aside from being an integral part of Arctic deltas, may also play an important role in protecting Arctic shorelines from coastal erosion (Dean and Dalrymple, 2002) and could enhance carbon sequestration (Overeem et al., 2022). Moreover, the shallow-water platform provided by the ramp could play an important role in the surrounding ecosystems (Lopez et al., 2006). The origin, evolution, and stability of the ramp features are therefore an important feature of Arctic deltas. A better understanding of all three are required to predict how changing sea ice cover and air temperatures will impact Arctic delta morphology.

In order to explore the possible origin of these ramps, numerical models form an important tool in the toolbox. Due to the complexity of the system (which involves permafrost, flow on low-slope environments, ice cover, spring floods, and more), delta models are typically divided into two classes. The first is built on physically-based equations, simplified to be computationally tractable (e.g., models involving Delft 3D; Lesser et al., 2004). They are able to simulate more directly delta dynamics, but at a cost of computational resources and the ability to cover time scales of years or longer. To address these issues, the second class of models – reduced complexity models (RCMs) – simulate phenomenological processes of arctic delta evolution using rule based trajectories of cellular automata (originating works in this field include, e.g., Murray and Paola, 1994, 2003; Murray,

2007). The rules governing the automata units are typically informed by physical equations under specific sets of conditions and by empirical observations. Due to the simplified construct, there are greater flexibilities in choosing the spatial and temporal step sizes, resulting in much greater spatial and temporal coverage whilst keeping computational requirements feasible. As Bokulich (2013) pointed out, there are on-going debates about the predictive power and explanatory insights offered by these classes of models, and their usage by researchers are typically with "division of cognitive labour"—to use a term coined by Bokulich—in mind. Within this context, we take the RCM approach in order to explore what physical-process rules could be favourable to the formation of a ramp feature.

One representative RCM for understanding deltas is the DeltaRCM (Liang et al., 2015b), the Arctic extension of which (called DeltaRCM-Arctic; Lauzon et al., 2019) will serve as the starting point of our work herein. DeltaRCM and its Arctic extension have been demonstrated to efficiently reproduce numerous observed features of natural and experimental deltas (Liang et al., 2015b, a; Lauzon et al., 2019; Piliouras et al., 2021). Our goal is to build upon previous work to reproduce the ramp features in order to explore their physical-process origin.

We start by rewriting the DeltaRCM base and reconstruct its Arctic extension in Julia, which has comparable performance as C and FORTRAN but retains the syntactical convenience of MATLAB and Python. We then explored the effects of modifying the rules to include additional processes and physics, with the hope of identifying and understanding the circumstances that favour the formation of the ramp feature. Moreover, motivated by the high flow rate during summer in large Arctic deltas, we also address the importance of summer months in Arctic delta evolution when examining growth and evolution over time scales of years or longer. We call the modified model ArcDelRCM.jl for succinctness and in keeping with conventions of Julia code packages, and to signify that it is a reconstruction of the Arctic extension of DeltaRCM based on published articles (Lauzon et al., 2019; Piliouras et al., 2021) and not a direct translation of the original DeltaRCM-Arctic source code (which is not publicly available).

The purpose of this article is two fold. First, we present and motivate the physical-processes rule changes we made to our reconstructed version of DeltaRCM-Arctic to arrive at the ramp-producing ArcDelRCM.jl version[1]. Second, we present the model outputs, including ones that are intended to simulate the evolution of large-scale Arctic deltas. Through these outputs, we demonstrate the model's capability in reproducing the 2-m ramp, identify the processes that led to their formation in the model, and make exemplary comparisons of these outputs with a real-world case, the Lena Delta. Aside from being the largest Arctic delta, Lena Delta is chosen for the real-world case example because of its fit to the modelled geometry. Many of the other Arctic river mouths (e.g., Ob, Yenisei, Mackenzie) are estuarine or geologically confined, and thus match the modelled geometry poorly, which may confound the analyses herein. Other Arctic deltas such as Olenyok and Colville are much smaller, and therefore more likely to be influenced in their growth by processes other than those modelled here (we discuss some of these processes at the end of Sect. 4). Through the exemplary case of the Lena Delta, we also argue for the importance of the summer months in modelling delta evolution on time scales of years or longer. Finally, we gauge the possible fate of these ramps under a warming climate.

---

[1]The specific version of the source code used in this study is published as a supplement as Chan (2023), archived by the GFZ Data Service. Any future updates can be found on the GitLab project page: https://gitlab.com/nhchan/arcdelrcm.jl.

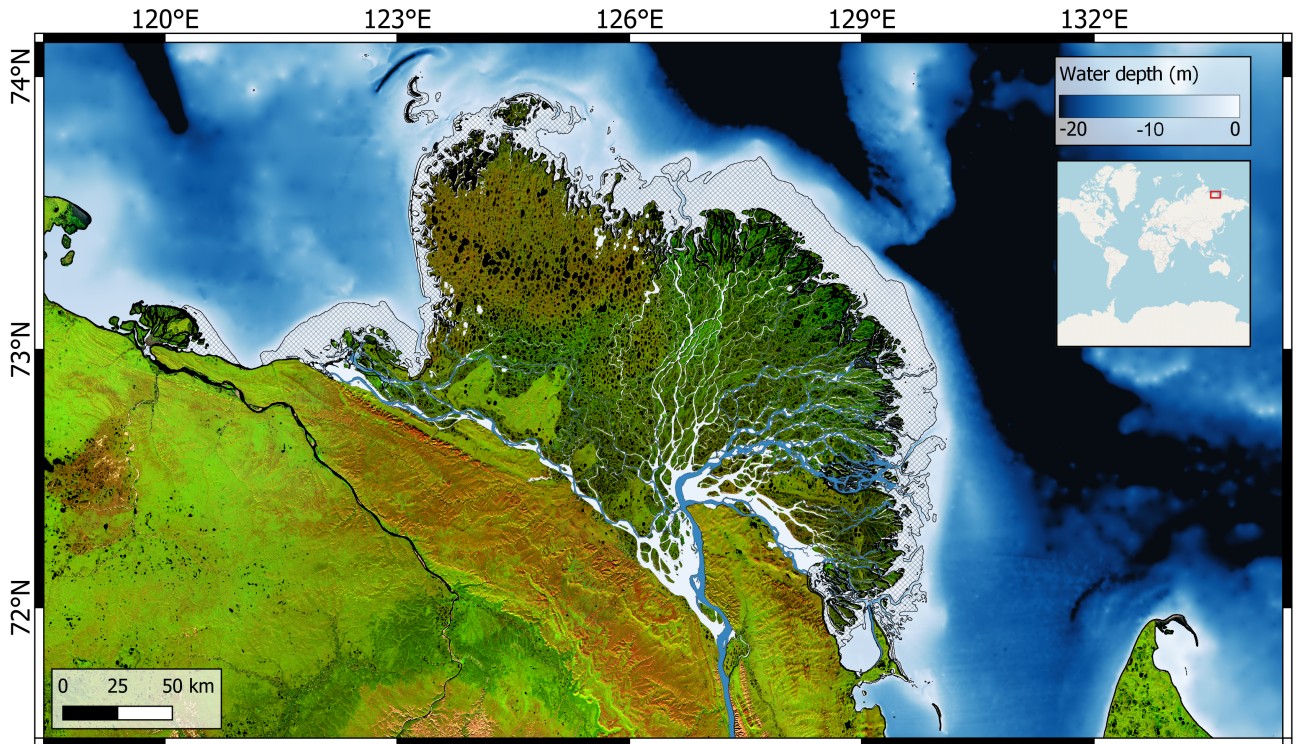

**Figure 1.** Map of the Lena Delta and the bathymetry of the southern Laptev Sea region (Fuchs et al., 2021). The orange-to-green relief shows the subaerial portion of the Lena Delta and its surrounding land. The blue-to-white colour scale shows the bathymetry. Dark blue channels within the delta show deep channels that do not freeze in winter (Juhls et al., 2021). Light blue within the delta shows the maximum channel area during the spring flood. The hashed area displays the shallow water 2-m ramp feature. Some deeply incised channels are visible within the ramp feature. The land area relief is visualised in a false colour Landsat-8 mosaic, courtesy of the United States Geological Survey, processed in Google Earth Engine. The red box in the inset shows the location on the world map.

## 2   Methods

### 2.1   Description of DeltaRCM(-Arctic)

In this section, we provide an overview of DeltaRCM and its Arctic extension (referred to as "DeltaRCM-Arctic" by Piliouras et al., 2021) based entirely on the model-describing publications (Liang et al., 2015b, a; Lauzon et al., 2019; Piliouras et al., 2021), the source codes of the (non-Arctic) DeltaRCM (Liang, 2015; Perignon, 2018), and our observations during the process of reproducing these models prior to extending them. These descriptions will facilitate the introduction of all of our modifications, left to Section 2.2, and the specifications of our simulations in Section 3.

### 2.1.1 Simulation Domain

In the most basic setup, the simulation domain (depicted in Figures 1 and 2 of Liang et al., 2015b) consists of a rectangular grid of $N_x$ by $N_y$ cells (typically, $N_y = 2N_x$). Each cell is a square with a width of $\delta c$. Along the $y$ dimension, the first $N_{wall}$ (typically 3) cells in the $x$ direction are defined as the inlet wall, which are impermeable and static. Centred around the $\left(\frac{1}{2}N_y\right)^{\text{th}}$ cell, there is an opening in the inlet wall $N_0$ cells wide, from where the water (volume) discharge, $Q_w$, and the sediment (volume) discharge, $Q_s$, enter the simulation domain. This opening is the inlet channel.

The domain is initialised with a water-surface elevation of $H$ (typically 0 m, taken to be the sea-surface height), a water depth of $h_B$ ('B' for ocean basin), and a corresponding bed elevation of $\eta = H - h_B$. Within the inlet channel, an initial surface slope, $S_0$, is added to mimic the backwater slope. An inlet flow depth, $h_0$, is given and reimposed at the start of each time step, such that the inlet flow speed, $u_0$, can be determined in conjunction with $Q_w$ and $N_0 \times \delta c$.

### 2.1.2 Flow Field

To build the flow field within each time step, the input discharge is divided into $n_w$ (which is 2000 in all examples) packets and sent through the simulation domain using a weighted random-walk scheme. The weights of each of the eight neighbouring grid cells are determined by a linear combination of two factors: (i) Water-surface gradient (as a proxy to gravity)

$$w_{i,\text{surface}} = \max(0, -\nabla H_i), \tag{1}$$

where $\nabla H_i$ is the gradient of water-surface elevation from the current cell towards its $i^{\text{th}}$ neighbouring cell. (ii) Flow depth, expressed as a resistance measure (as a proxy to inertia), scaled by both the projection of the local flow field to the eight neighbouring cells and by the distances to those neighbouring cells

$$w_{i,\text{inertia}} = \frac{1}{R_i} \frac{\max(0, \mathbf{q_w} \cdot \mathbf{d}_i)}{\Delta_i}, \tag{2}$$

where $R_i^{-1} = (h_i - h_{\text{ice},i})\left(1 - \frac{h_{\text{ice},i}}{h_i}\right)$ is the flow resistance measure of the $i^{\text{th}}$ neighbouring cell, taking into account the full water depth, $h_i$, and the portion of it that is ice, $h_{\text{ice},i}$ (Lauzon et al., 2019); $\mathbf{q_w}$ is the unit discharge vector at the current cell (serving as the flow-direction vector); $\mathbf{d}_i$ is the unit vector pointing from the current cell towards the $i^{\text{th}}$ neighbouring cell; and $\Delta_i$ is the distance between the centres of the current cell and that of its $i^{\text{th}}$ neighbour. Cells with water depth shallower than $0.1h_0$ (up to a maximum of 0.1 m) are classified as "dry" and thus assigned a weight of zero. Discharge is conserved throughout, with each packet's contribution ($q_w$) to each visited grid cell recorded in an additive manner. The total random-walk weight is combined through a "partitioning coefficient", $\gamma$:

$$w_i = \gamma w_{i,\text{surface}} + (1 - \gamma)w_{i,\text{inertia}}. \tag{3}$$

Liang et al. (2015b, a) described $\gamma$ as a free parameter that controls the lateral spread of water. By increasing the importance of the water-surface gradient, its cross-channel component is also emphasised. The value of $\gamma$ is typically small (e.g., 0.05 according to Liang et al. 2015b), and is also given by Liang et al. (2015a) as

$$\gamma = \frac{g\delta c S_0}{u_0^2} \tag{4}$$

as a guideline to choose an appropriate value, where $g$ is the gravitational acceleration. The latter expression may have arisen from taking the ratio between the pressure gradient and inertia terms (without local acceleration) of the shallow water equations.

The latter expression is the version for $\gamma$ implemented in the source codes of DeltaRCM (Liang, 2015; Perignon, 2018), which would result in, for example, a value of 0.098 in the demonstration cases with 50% sand fraction[2] instead of 0.05 in Liang et al. (2015b). We note that this remains a free parameter that has been given a range of values (e.g. 0.02 to 0.15) in various tests of its influence on delta planforms (Liang et al., 2015b, supplementary material). In our own experiments mimicking the size and conditions of the Lena Delta, with much larger $\delta c$, we have found that $\gamma = 0.135 \, (\pm 0.02)$ works best in producing the planform of deltas similar to the Lena Delta.

The slope $S_0$ is imposed along each water packet's path, so that an averaged, approximate backwater slope is formed in $H$ in each time step. The details of this procedure can be found in Liang et al. (2015b, Sect. 3.2.2)[3].

Given the bed-elevation field at the time, $\eta$ (either from initialisation or from the previous time step), the water depth is determined from $h = H - \eta$. Using the accumulated $q_w$ from all passing water packets and the flow depth $h$ (minus any portion that is $h_{ice}$), the flow speed $u$ is calculated. Finally, the direction of flow at each grid cell is the average entry and exit directions of all water packets that passed through that cell (Figure 4 of Liang et al., 2015b). We note that, in practice, the unit-discharge vector field serves as the flow-direction field and thus computed during the routing of the water packets. Its underrelaxation by the free parameter $\gamma$ is done directly when computing the routing weights (Equation 3).

The flow-field determination process is iterated multiple ($n_{iter}$) times per time step to suppress any instability due to the randomised nature of the scheme (Liang et al., 2015b). Note that there is one more step that is not documented in the original article, but exists in the non-Arctic source codes (Liang, 2015; Perignon, 2018): the flow field $q_w$ also undergoes an "underrelaxation" identical to that undergone by $H$ (Liang et al., 2015b, Equation 12), except that it is with a different coefficient and is applied between each iteration instead of merely across time steps. The underrelaxation coefficient for the $q_w$ field is 0.9 for the first iteration (in each time step) and $2/n_{iter}$ for all subsequent iterations.

### 2.1.3 Ice Cover

Lauzon et al. (2019) extended DeltaRCM to simulate Arctic deltas. They simulate only the spring-flood period (assumed to be 10 days). At the beginning of each flood, the maximum ice cover is defined with a pair of parameters: maximum ice thickness, $h_{ice,max}$, and maximum ice extent, $f_{ice}$, which is a fraction between 0 and 1. At maximum, ice thickness is $h_{ice,max}$ everywhere except on "dry" grid cells and except in cells within $(1 - f_{ice})$ of the mean distance between the inlet channel and the (average) coast of the delta. To improve simulation stability, a taper from 0 to $h_{ice,max}$ over the equivalent of 10 $\delta c$ is imposed from the edge of the ice-free zone towards the ocean. For similar purposes, $h_{ice}$ may not be equal to the local water depth (e.g., $h_{ice} < 0.9999h$).

---

[2]Note that their $S_0$ is influenced by the sand fraction, and can vary by up to a factor of 3 between $0 - 100\%$ sand fractions (Liang, 2015; Perignon, 2018). This contributes strongly, through $\gamma$, to how the sand/mud fractions influence the planform of the delta, between fan-like and elongated channels-like.

[3]In the published version of Liang et al. (2015b), the water-depth condition defining the edge of the delta is given as having the bed elevation $\eta_{shore} = H_{sea-level} - 0.9h_0$, or equivalently, $h = 0.9h_0$. That is a typo, and the factor of 0.9 should have been 0.1, based on the source codes of DeltaRCM (Liang, 2015; Perignon, 2018).

Two processes contribute additively to the melting of the ice: discharge-based heat flux and atmospheric heat. The former is proportional to the flow speed field, $u$, of the current time step, and is given by an expression from Searcy et al. (1996, Equation 2 and the subsequent unnumbered equation) (also in Lauzon et al., 2019, supplementary material, and Piliouras et al., 2021). The latter (i.e., atmospheric contribution) is given by

$$\left(\frac{dh_{\text{ice}}}{dt}\right)_{\text{atmospheric}} = -a\frac{h_{\text{ice,max}}}{t_{\text{melt}}}, \tag{5}$$

where $t_{\text{melt}}$ is the period during which the entire ice thickness would melt due to atmospheric heat alone, and $a$ is a scaling factor, between $0$ and $1$, to tune the contribution of atmospherically induced melting towards the total melt. In the model result of Searcy et al. (1996), $a \approx 0.58$, and in Piliouras et al. (2021), $a = 0.5$. In DeltaRCM-Arctic, $t_{\text{melt}}$ is taken to be 10 days (i.e., the entire simulation period for each model year) (Lauzon et al., 2019; Piliouras et al., 2021).

### 2.1.4 Sediments and Permafrost

The sediment entering the delta in DeltaRCM(-Arctic) is split into two categories: sand (representing bed load) and mud (representing suspended load). Their relative fraction of the total is given by the "sand-fraction" parameter, $f_{\text{sand}}$. The routing of sediment packets is similar to the routing of water packets, and is described in (Liang et al., 2015b, Sect. 3.2.4).

Deposition and erosion of sediments are determined by a set of threshold relations (detailed in Liang et al., 2015b, Sect. 3.2.5). Deposited sediments are tracked and stored in a grid with the same lateral spatial coverage as the simulation domain ($N_x$ by $N_y$), and with a depth-wise dimension of $\delta z$ per cell. After all sediment packets have passed through during one time step, any bed-elevation gains are added to the sediment grid. Each of these added grid elements records the "sand fraction" (i.e., the relative fraction of sand in the total volume of sand and mud) of all sediments deposited by all passing packets. Grid cells corresponding to eroded sediments (i.e., the volume picked up by passing packets) are simply removed.

In DeltaRCM-Arctic, Lauzon et al. (2019) assume a constant active layer (or thaw depth) of $0.5$ m. A sediment cell is considered a permafrost cell if it has remained below the thaw depth for at least two years. If a column of sediment contains $75\%$ permafrost cells or if the permafrost cells amount to $\geq 75\%$ of the inlet-channel depth $h_0$, the corresponding planar grid cell (i.e., in the $x,y$ dimensions) are flagged as a "permafrost cell".

To simulate the erosion of a permafrost bed, DeltaRCM-Arctic (Lauzon et al., 2019) used a multiplicative erodibility factor, $E \leq 1$, to scale the erosional thresholds (given in Liang et al., 2015b, Sect. 3.2.5) in (planar) permafrost grid cells, such that erosion is harder to achieve.

### 2.1.5 Bed Diffusion and Flood Correction

Immediately after each round of sediment packet routing, a bed-diffusion process is applied "to take into account the influence of topographical slope on sediment flux", and to introduce "lateral erosion by allowing sediment on the bank to be removed and added to the channels" (Liang et al., 2015b). This is achieved by calculating the "diffusive sand flux"

$$q_{\text{sand,diff}} = \alpha \left|\nabla\eta\right| q_{\text{sand}}, \tag{6}$$

where $\alpha$ is a coefficient set to 0.1 in all demonstrated cases, $|\nabla\eta|$ is the absolute slope of the bed, and $q_{\text{sand}}$ is the sand flux into and out of the grid cell in concern. Both $|\nabla\eta|$ and $q_{\text{sand}}$ are calculated across the boundary between the grid cell in concern and each of its neighbouring cells. Contributions of each of the neighbours are summed to give $q_{\text{sand,diff}}$, which is then used to determine the diffusion-induced bed-elevation change $\Delta\eta$ of the grid cell in concern. In DeltaRCM-Arctic, $\alpha$ is further scaled by the erodibility factor, $E$.

After each full update of the water surface and each full update of the bed elevation, some grid cell may contain water surface that is higher than the dry bed in a neighbouring cell, causing an unphysical shoreline or river-bank location. In the source codes of DeltaRCM, this is referred to as "flood correction" (Liang, 2015; Perignon, 2018), but not explicitly described in the article (Liang et al., 2015b). This means that any "dry" grid cells are refilled with water when one or more neighbouring cells have higher water surfaces. We assume that DeltaRCM-Arctic also inherited this mechanism, and describe our interpretation of it in Sect. 2.2.4.

## 2.2 Modifications and Interpretations in ArcDelRCM.jl

We completely re-wrote the model in the Julia language, first according to the model descriptions of Liang et al. (2015b) and Lauzon et al. (2019) summarised above. Due to the non-availability of the source code of DeltaRCM-Arctic, whenever we present simulations results of "DeltaRCM-Arctic", we mean the DeltaRCM-Arctic configuration of ArcDelRCM.jl. Aside from reconstructing DeltaRCM-Arctic, we also made significant changes to the model itself to improve its ability to account for processes that are climate-sensitive, with the goal of exploring the physical-processes that give rise to ramp features. We describe them in turn in this section[4]. Since the ramps appear to be related to winter ice, its depth being the same as the thickness of winter sea ice, we begin by examining ice-related processes.

### 2.2.1 Bottom-fast Ice Protection and Shielding

Due to the weighted random walk scheme and the limit of $h_{\text{ice}}$ to 99.99% of the water depth (for numerical stability), water packets can still go through grid cells where the entire water depth is effectively in the form of ice (albeit with decreased probability, since the un-frozen water depth plays an important role in determining random-walk weights). This can generate unrealistic flow pathways with anomalously high speeds (due to ice constriction) and consequent ice melting. To eliminate this unrealistic behaviour, we prohibit flow-speed induced melting when ice is effectively in contact with the bed (i.e., $h_{\text{ice}} \approx h$). We call this "bottom-fast ice protection". In the same locations, the cell is considered entirely blocked by bottom-fast ice and no erosion or deposition can occur. We call this "bottom-fast ice shielding" of the bed.

### 2.2.2 Time-dependent Thaw Depth

During winter months, bottom-fast ice can bond with the sediments below through conductive heat transfer. Since we introduced ice protection and ice shielding, we deemed it logically necessary to also introduce a time-dependent thaw depth to

---

[4]Three more user-feature additions implemented during testing, but are not utilised in our results, are described in Appendix A.

avoid an abrupt jump (in time) from ice-bonded top sediment layers to a finite thaw depth (or active-layer depth, as defined

in DeltaRCM-Arctic) when bottom-fast ice ceases to be bottom-fast. We do so according to the Stefan Model (Riseborough et al., 2008; Lunardini, 1981): $X = \sqrt{\frac{2\lambda I}{L}}$, where $X$ is the thaw depth, $\lambda \equiv 2.22$ W m$^{-1}$ K$^{-1}$ is the thermal conductivity of ice near $0°$C, $L \equiv 3.3355 \times 10^8$ J m$^{-3}$ is the volumetric latent heat of fusion of water, and $I$ is the "positive degree day index", which is the integrated number of days times the positive temperature since winter. For $I$ in our simulations, we use a mean temperature of $4°$C to get $I$ (see Appendix B for the reason for this choice and the sensitivity of the model output to this value).

Where there is bottom-fast ice at the start of the simulation, the thaw depth starts at $0$ and $I$ only begins to increment when the ice in the pixel is no longer bottom-fast. Otherwise, $I$ starts at 10 days in our standard simulation (see Appendix B for the reason behind this choice and the sensitivity of the model output to this value). As a result of the time-variable thaw depth, we forgo the "permafrost" label for horizontal (or planar) grid cells. In usages where the classification of vertical sediment cells as permafrost is relevant, it can be defined as the vertical cells that stayed below the maximum thaw depth (instead of a static

depth of 0.5 m in DeltaRCM-Arctic) for at least 2 years, although we do not use such labels in this study. To avoid confusion, we will use the terms "frozen" or "thawed" (as opposed to "permafrost") in the context of the erosional rules in ArcDelRCM.jl.

### 2.2.3 Erosion of Frozen Ground

To simulate the erosion of the permafrost bed, the original model used a scaled erodibility factor, $E \leq 1$, for horizontal grid cells with over 75% permafrost content in the sediment column. With the introduction of time-dependent thaw depth, we find

it more self-consistent to check the calculated erosional depth of the grid cell against its corresponding sediment column: If the calculated erosion reaches deeper than the available thawed layers, the erosional depth is limited to the thawed (vertical-cell) layers only. Whilst the sediment column is immediately updated as a sediment packet passes through, which prevents duplicate erosion/deposition by successive packets, the value of the bed elevation is kept in memory unchanged during each time step, thus tracking the exact layer that is at the bottom of the thawed section. As the model proceeds to the next time step, more

layers are thawed as the thaw depth deepens as described in the previous subsection. This approach is taken in both erosion by packets routed through individual pixels, and the bed-diffusion process explained in Sect. 2.1.5. We thus forgo the use of the erodibility factor, $E$ (and, as mentioned in Sect. 2.2.2, the labelling of any pixel as "permafrost"), and consider only the depth of the boundary between frozen and thawed grounds.

Conceptually, both DeltaRCM-Arctic and ArcDelRCM.jl allow erosion of sediment columns with frozen (vertical) cells.

Consider a sediment column with a significant fraction of its cells frozen during some number of time steps (i.e., with a shallow thaw depth in ArcDelRCM.jl, or with the corresponding horizontal grid cell is labelled as "permafrost" in DeltaRCM-Arctic). DeltaRCM-Arctic would allow for erosion with a scaled-up flow-speed threshold through $E$ (and a similarly scaled-down bed diffusion); the cumulative erosional effect on the corresponding column can thus be thought of as a fraction of the equivalent no-permafrost case (i.e., $E = 1$). By allowing only thawed (vertical) sediment cells to be eroded, the same sediment column

in ArcDelRCM.jl also undergoes erosion that is a fraction of an equivalent case without any frozen (vertical) cells. However, the main difference between the two is the timing of the erosional events. ArcDelRCM.jl allows the thawed vertical cells to be eroded unhindered and thus earlier (considering the chance of having lower versus higher flow speed across the domain at

any given moment), but then delay the erosion of the at-the-time frozen vertical cells, which would become thawed in the next time step and freely erodible again.

### 2.2.4  Flood Correction

-A "flood correction" mechanism is built into the original, non-Arctic source codes of Delta RCM (Liang et al., 2015a; Perignon, 2018), but not explicitly described in the article of Liang et al. (2015b). This mechanism ensures that "dry" grid cells are refilled with water when one or more neighbouring cells have higher water surfaces. Without access to the source code of DeltaRCM-Arctic (Lauzon et al., 2019), how they handled the presence of ice in this context is unclear. In ArcDelRCM.jl, we interpret the water surface elevation in this context as the below-ice surface. Therefore, only dry cells with at least one neighbouring cell that has a higher *liquid* water-surface elevation are rewetted. The water-surface of the rewetted cell is calculated from the average of all surrounding wet cells.

### 2.2.5  Time-step Size

To keep the simulation numerically stable, the original model determines the time-step size based on the volume of sediments entering the simulation domain. Specifically, $\Delta t = \frac{N_0^2 h_0 \delta_c^2}{10 Q_{s_0}}$ is given as a guideline by Liang et al. (2015b) to prevent too much sediments from entering in each time step relative to the accommodation space for deposition in the grid cells. We discovered in our simulations intended to mimic the Lena Delta, where the grid-cell dimensions (which are terms in the numerator) are several times larger than in Liang et al. (2015b) and Lauzon et al. (2019), that the $10$ in the denominator of expression for $\Delta t$ needed to be increased by a factor of a few (e.g., to $40 - 80$, depending on the volume of sediments entering the domain in each time step in the specific simulation). Otherwise, the model becomes numerically unstable. In order to facilitate the use of time-dependent input discharges (described in the next subsection), we let $\Delta t$ be user-determined, but implemented an internal checking procedure to warn users of potential numerical instability. For this check, we introduce a quantity that we call "scale-height measure": $\zeta \equiv \frac{Q_{s_0} \Delta t}{\delta_c^2 h_B}$. $\zeta$ is a rough scaling between the volume of sediment entering through the inlet channel in each time step and the "available" volume of an average single ocean-basin grid cell. This is in the same spirit as the "reference volume" introduced by Liang et al. (2015b, Equation 1) that led to their guideline expression for $\Delta t$. Based on experiments with both the dimensions of Liang et al. (2015b), Lauzon et al. (2019), and our Lena-like dimensions, we found that $\zeta$ should be $\lesssim 3$ for the model to be numerically stable. Due to the random-walk nature of the RCMs, this threshold is not sharp.

### 2.2.6  Input Discharges as Time Series

In large drainage basins such as the Lena watershed, discharge beyond the spring flooding season remains significant through the summer. In order to capture these deltas' summer evolution, we modify the model to take in time series of input discharges (both water and sediments) and extend the simulation model year to include the summer months. Under this setup, the inlet flow speed, $u_0$, the inlet flow depth, $h_0$, and the reference water-surface slope (used as an approximation of the backwater

slope), $S_0$, all become time series themselves and are dependent on the water discharge $Q_{w_0}$ time series[5]. To simplify the process for the user and reduce chances of mistakes, ArcDelRCM.jl users input the minima of $u_0$, $h_0$, and $S_0$, corresponding to the minimum $Q_{w_0}$. The simulation then uses the time series of $Q_{w_0}$ to calculate the corresponding time series of $u_0$ and $h_0$ based on a scaling derived from the Gauckler-Manning formula (Gauckler, 1867; Manning et al., 1890), and of $S_0$ based on simple geometric arguments. We describe these in order.

We first assume open-channel flow and that the (overall) channel bed slope is approximately constant. The average flow velocity is given by the Gauckler-Manning formula (Gauckler, 1867; Manning et al., 1890), which, for our purpose and under our assumptions, can be expressed as a proportionality:

$$\bar{u} \propto R_h^{\frac{2}{3}}, \tag{7}$$

where $R_h$ is the hydraulic radius:

$$R_h = \frac{A}{P},$$

where $A$ is the cross-sectional area of the channel, and $P$ is the "wetted perimeter" of the channel. Assuming a simplistic, rectangular channel shape, with width, $w$, and flow depth, $h$, we have $A = hw$ and $P = 2h + w$. Then, using the expression for discharge, $Q = A\bar{u}$, we calculate the flow depth under some new discharge, $Q_n$, in relation to an "original" discharge, $Q_o$, as follows:

$$\frac{Q_n}{Q_o} = \frac{A_n \bar{u}_n}{A_o \bar{u}_o} = \frac{h_n \bar{u}_n}{h_o \bar{u}_o} = \frac{h_n \left( \frac{h_n w}{2h_n + w} \right)^{\frac{2}{3}}}{h_o \left( \frac{h_o w}{2h_o + w} \right)^{\frac{2}{3}}} \approx \left( \frac{h_n}{h_o} \right)^{\frac{5}{3}},$$

where we have assumed a constant $w$ and that $w$ is much greater than both $h_o$ and $h_n$. Similarly, the transformation of $\bar{u}$ from $Q_o$ to $Q_n$ can be calculated, via Equation 7, by

$$\frac{\bar{u}_n}{\bar{u}_o} = \left( \frac{h_n}{h_o} \right)^{\frac{2}{3}}.$$

Given an "original" slope, $S_o$, corresponding to $h_o$ under the discharge $Q_o$ (with $\bar{u}_o$), one can define a "baseline reach", $L_B \equiv h_o / S_o$, such that the new slope under the discharge $Q_n$ (with $h_n$ and $\bar{u}_n$) can be approximated by

$$S_n \approx \frac{h_n}{L_B}.$$

In the manner described above, the inlet flow depth, the inlet flow speed, and the reference slope corresponding to the input discharge time series, $Q_w$, can be calculated as long as $h_0$, $u_0$, and $S_0$ corresponding to the minimum $Q_w$ are supplied.

The sediment discharge, $Q_{s_0}$, can also be input as a time series. However, many large Arctic deltas have sediment fluxes that scale approximately with water discharge (Holmes et al., 2002). Therefore, following DeltaRCM, a simple multiplicative factor is used to translate $Q_{w_0}$ to $Q_{s_0}$.

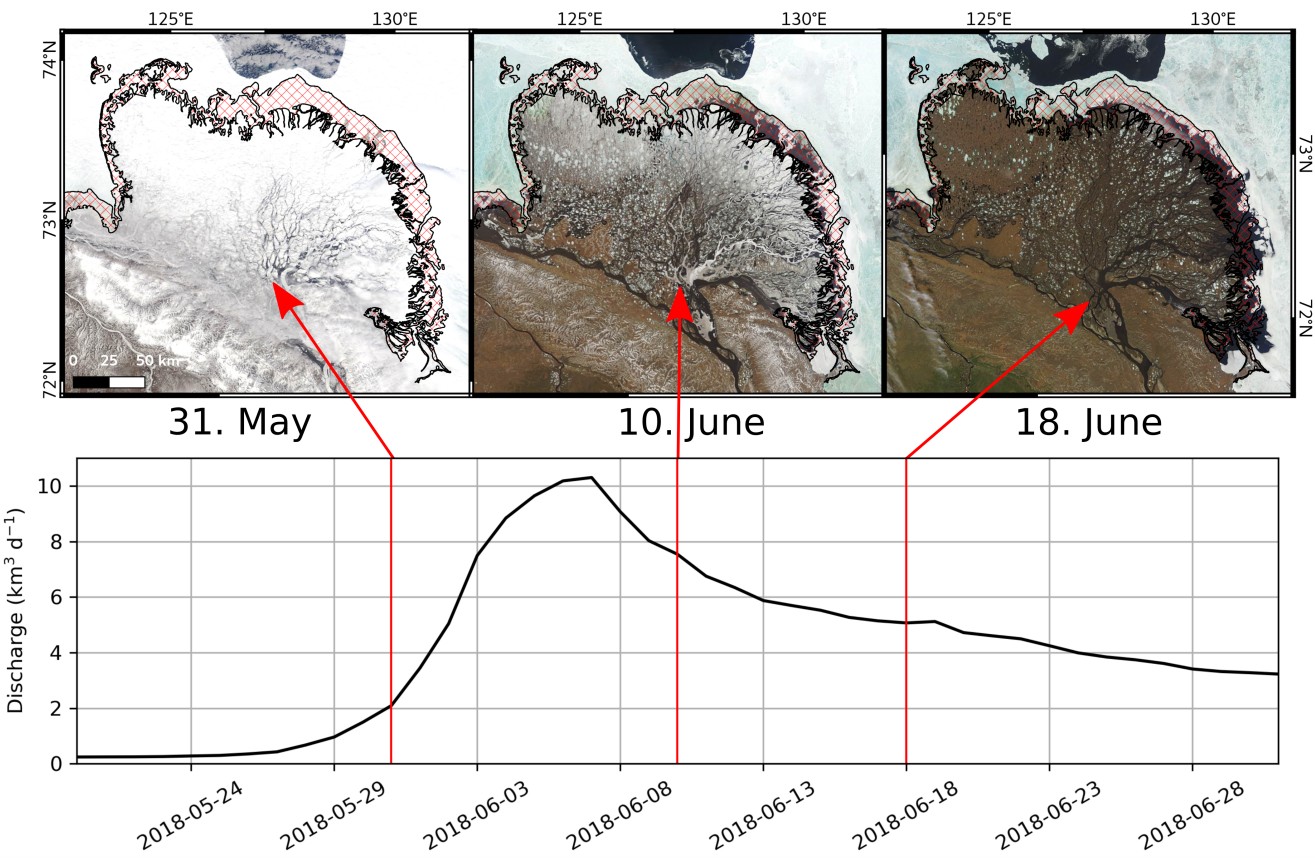

**Figure 2.** Satellite imagery of the Lena Delta from 2018 showing the delayed break-up of bottom-fast ice on the ramp feature. The upper panels show the imagery of the Lena Delta. The hash pattern in red marks the location of the ramp, and the left, middle, and right panels correspond respectively to the start, middle, and completion of ice break-up on the delta itself. The corresponding dates and discharge (measured at Kyusyur station and corrected for the distance to the Lena Delta; Juhls et al., 2020) is shown in the bottom panel, marked by the red arrows. Satellite imagery is acquired by the Moderate Resolution Imaging Spectroradiometer (MODIS) and obtained from NASA Worldview.

### 2.2.7 Controls on the Melting of Ice

We have added the capability for the users to shift the timing and duration of the ice cover's melt. This modification is motivated by the observations that bottom-fast ice just off-shore from the delta remains intact longer than the ice cover on the delta, and does not disintegrate until the delta itself is almost ice free or containing only the so-called "serpentine (floating) ice" in the channels (Figure 2). This ice remains bottom-fast longer perhaps due to its additional thickness from congelation from below during surges, or from water overflowing its top and depressing it during floods (Reimnitz, 2002). Moreover, based

---

[5]Static values of these quantities remain possible as inputs in ArcDelRCM.jl.

on examinations of satellite imagery, the duration of ice-cover melting on the Lena Delta is closer to 20 days instead of 10 days. During this time, both the water temperature and air temperature remain close to the freezing point (Reimnitz, 2002; Yang et al., 2005; Juhls et al., 2020). An example observation of this breakup sequence and duration is shown in Figure 2. In ArcDelRCM.jl, users can specify the length of the ice-cover melting period, during which atmospheric heat contributes to the

melting. Note that flow-speed induced melting is always active, subject to the protection described in Sect. 2.2.1.

## 2.3   Graph Analyses on Ensemble Runs

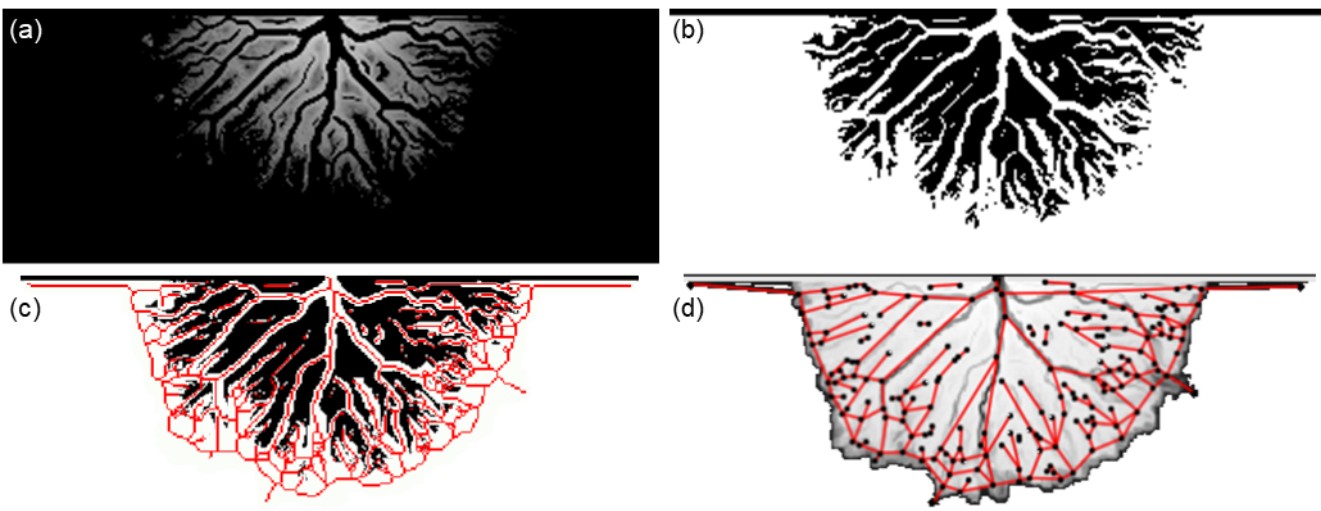

**Figure 3.** Exemplary overview of workflow to extract the hydrological graph from the imaged delta simulation. (a) bed elevation, (b) binarization of bed elevation to differentiate between channels and non-channels, (c) skeleton (in red) of the channels overlaid on the binary image (b), (d) graph of the channel network derived from the skeleton in (c) with edges in red and nodes in black overlaid on the delta elevation.

To assess quantitatively and statistically the effects of the various modifications described in Sect. 2.2, we apply graph theory to derive metrics on a collection of ensemble runs, each set with a different process (corresponding to Sect. 2.2.1 to 2.2.4), or combination thereof, enabled. Previous work by, for example, Smart and Moruzzi (1971), Edmonds et al. (2011), Tejedor et al.

(2015), and Nesvold et al. (2019) has shown that the topologies of deltas can be described with quantitative graph metrics, such as the "loopiness" and the structural overlapping of the subnetworks, the "recombination factor" describing the ratio between the number of junctions and the number of forks in the delta systems, or the fractal dimension characterising a delta's self-similarity. While these metrics give interesting insights to the environmental properties of the real-world deltas, we sought a holistic approach that would quantify the differences between simulation results with simplest, yet meaningful descriptors. We

thus made use of the approach introduced by Rettelbach et al. (2021), which provides an end-to-end approach starting from the

extraction of the graph from the delta images to providing the quantitative metrics of interest for the comparison of the ensemble runs with different parameters and forcings. While the approach by Rettelbach et al. was initially developed for characterising hydrological networks in polygonal permafrost landscapes, the methodology of automated graph extraction from underlying imagery remains exactly the same. In the original publication, the authors used binarised digital elevation models to compute the skeleton of the channel network, while we here extracted the graph from the deltas binarised based on the location of the channels (see Figure 3). In this context, channel pixels are defined as those having a water depth of $0.1$ m or over, which is the threshold value for "dry"/"wet" pixel labels used in DeltaRCM (Liang et al., 2015b) and inherited by both DeltaRCM-Arctic and ArcDelRCM.jl. "Open-ocean" pixels without any depositions (i.e., with depth equalling the ocean-basin depth, $h_B$) are excluded. From the derived graph (as seen in Figure 3d in red and black), we then calculated the following metrics:

- Number of nodes and edges, giving us an idea of the size and complexity of the underlying hydrological network.

- The number of connected components, where each connected component represents a hydrological subnetwork. One – and only one – component will always be connected with the apex of the delta, and thus any edges in this component (channels in this subnetwork), are considered "active". Any other component will represent an abandoned channel (or cluster of channels) that is no longer fed by the upstream river. Evaluating this number in combination with the graph density (see below), we can gather similar information as Tejedor et al. do with their metric of resistance distance.

- The total length of all channels combined, which makes quantification of the amount of all potential waterways possible.

- The graph's density, which lets us estimate the network flow effectiveness. It is defined via the ratio of the number of existing edges over the number of edges that could exist based on the number of nodes, $n$. For planar graphs, this equals $3(n-1)$. It can be seen as a simpler alternative to the self-similarity measure of Fractal Dimension introduced by Edmonds et al. (2011).

- The graph's diameter, representing the longest of all shortest path lengths between vertices. In combination with the total length of channels, this gives an idea on the asymmetry of the delta.

These metrics from basic graph theory are efficient to calculate and can already provide valuable insights into the properties of deltaic networks and accurate parameters to quantify and compare the results of the different ensemble runs, thus providing a simple yet powerful alternative to the more complex ones introduced and described by Smart and Moruzzi (1971), Edmonds et al. (2011), Tejedor et al. (2015), and Nesvold et al. (2019).

## 3 Results

We present here the outputs of ArcDelRCM.jl and its "DeltaRCM-Arctic" configuration (for comparison), with particular attention paid to the ramp feature. All simulations share the following parameters: $(N_x, N_y) = (150, 300)$, $N_{wall} = 3$, the number of water and sediment packets (sand and mud separately) are $n_w = n_s = 2000$, the coefficient for bed diffusion $\alpha = 0.1$, and the scaling factor for atmospherically induced melting $a = 0.5$. Further parameters applicable to individual cases are

specified in the respective subsections below. Any remaining parameters and conditions not explicitly listed take on values and specifications given in Liang et al. (2015b). Note that the colour-blindness friendly colour scheme, uniform across all the filled-contour figures in this section, is chosen to highlight the per-metre gradation of elevations below the water surface, with focus on the shallower depths where the existence or absence of the ramp can be seen. Interpretations and speculations are left to Sect. 4.

## 3.1 Analogous Setup to DeltaRCM-Arctic Demonstrations

In this subsection, we present comparisons of simulations, run with identical parameters and identical random seeds, in ArcDelRCM.jl and its DeltaRCM-Arctic configuration. Specifically, we adopt $\delta c = 50$ m, $N_0 = 5$, $h_0 = 5$ m, $h_B = 5$ m, $u_0 = 1$ m s$^{-1}$, $S_0 = 1.5 \times 10^{-4}$, $Q_w = 1250$ m$^3$s$^{-1}$, $Q_s/Q_w = 0.001$, a sand fraction (of the total sediment volume) of 25%, a maximum ice extent of 40%, $h_{\text{ice,max}} = 2$ m, and $\gamma = 0.0735$. With the exception of $h_{\text{ice,max}}$, these values are chosen after the demonstrated cases of DeltaRCM-Arctic in Piliouras et al. (2021) and Lauzon et al. (2019). In the case of DeltaRCM-Arctic, we use an erodibility factor $E = 0.8$, which is the "middle" value used in Lauzon et al. (2019) and Piliouras et al. (2021). For ArcDelRCM.jl, we specify $\Delta t = 25000$ s, matching the $\Delta t$ given by the DeltaRCM(-Arctic) guideline. All cases are run for 5000 time steps, with an additional 300 "lead-up" steps under non-Arctic conditions to build up a seed flow field (following Lauzon et al., 2019; Piliouras et al., 2021).[6]

The first row of Figure 4 shows the output from DeltaRCM-Arctic (Figure 4a) and ArcDelRCM.jl (Figure 4b). The second row of Figure 4 shows the output from the same simulations as the first row, but with $h_{\text{ice,max}} = 3$ m (Figures 4c and 4d). The last row of Figure 4 shows the output of ArcDelRCM.jl under identical configurations as in Figure 4b, except $h_B$ is increased to 7m in Figure 4e, and the ice extent is increased to 100% in Figure 4f.

The ramp features form continuous bands around all the deltas in ArcDelRCM.jl except in the case where $h_B = 7$ m, in which the ramp is not well-formed (Figure 4e). The ramp appears to be more prominent in the case with 100% ice extent (Figure 4f) and the case with $h_{\text{ice,max}} = 3$ m (Figure 4d). The deltas of DeltaRCM-Arctic do not show such ramps (Figure 4a), but rather display lobes or tentacles of "off-shore depositions" (as they are called in Piliouras et al., 2021) around channel outlets (Figure 4c). The $h_{\text{ice,max}} = 3$ m case (Figure 4c) has prominent off-shore deposits around channel outlets that are at the right depth for the ramp feature (also observed by Piliouras et al. 2021), but has less deposition in the lateral directions to join with neighbouring lobes to form a band. The lobes are also more disrupted by additional deposits to higher elevations. As we will explore in the next subsection, we find that the lateral joining appears after switching on the protection of bottom-fast ice (Sect. 2.2.1). A visual representation of the thaw-depth patterns corresponding to Figures 4b and 4f is shown in Appendix C.

## 3.2 Individual Modifications in ArcDelRCM.jl

Figure 5 shows the effects on the simulated deltas arising from the individual modifications described in Sect. 2.2.1 to 2.2.4. The model settings are identical to the $h_{\text{ice,max}} = 2$ m cases of Figure 4, but with a different random seed. The random seed

---

[6]We do not find that the lead-up phase has noticeable impacts on the numerical stability or the resulting deltas in our numerical experiments.

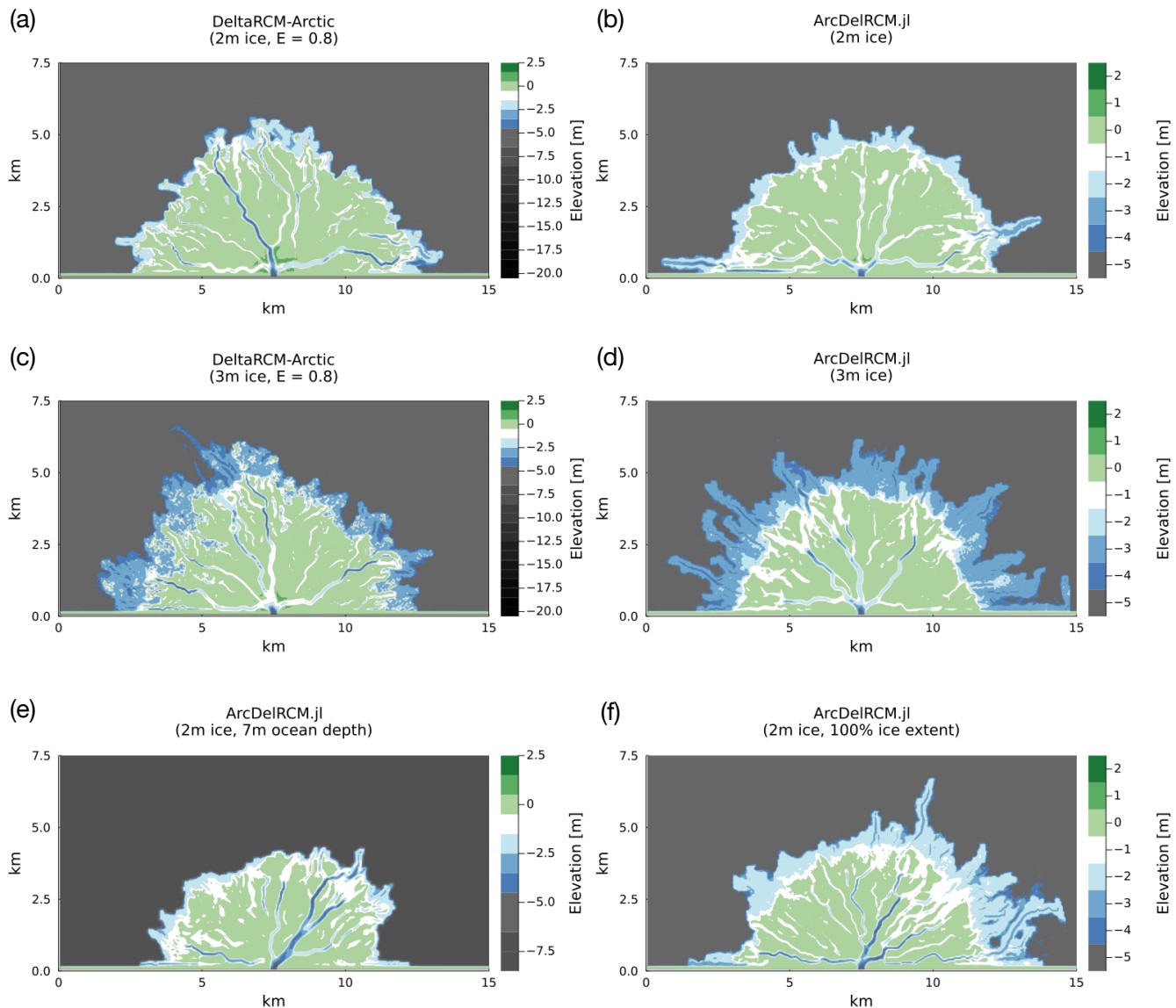

**Figure 4.** Bed-elevation output of (a, c) DeltaRCM-Arctic and (b, d-f) ArcDelRCM.jl after 5000 time steps (following a 300-step lead up under non-Arctic conditions). All runs have identical parameters (see text for full configuration), except the following differences: (a-b) $h_{\text{ice,max}}$ is 2 m; (c-d) $h_{\text{ice,max}}$ is 3 m; (e) $h_B$ is 7 m; and (f) the ice extent is 100%. Note the depths of the ramp features in panels b and d, which correspond to $h_{\text{ice,max}}$. Tentacle-like "off-shore depositions" (as described in Piliouras et al., 2021) are visible in the $h_{\text{ice,max}} = 3$ m cases of DeltaRCM-Arctic (in panel c) as well as outside the ramp of ArcDelRCM.jl (in panel d). Also, the ramp feature has vanished in panel e, but has become more prominent in panel f.

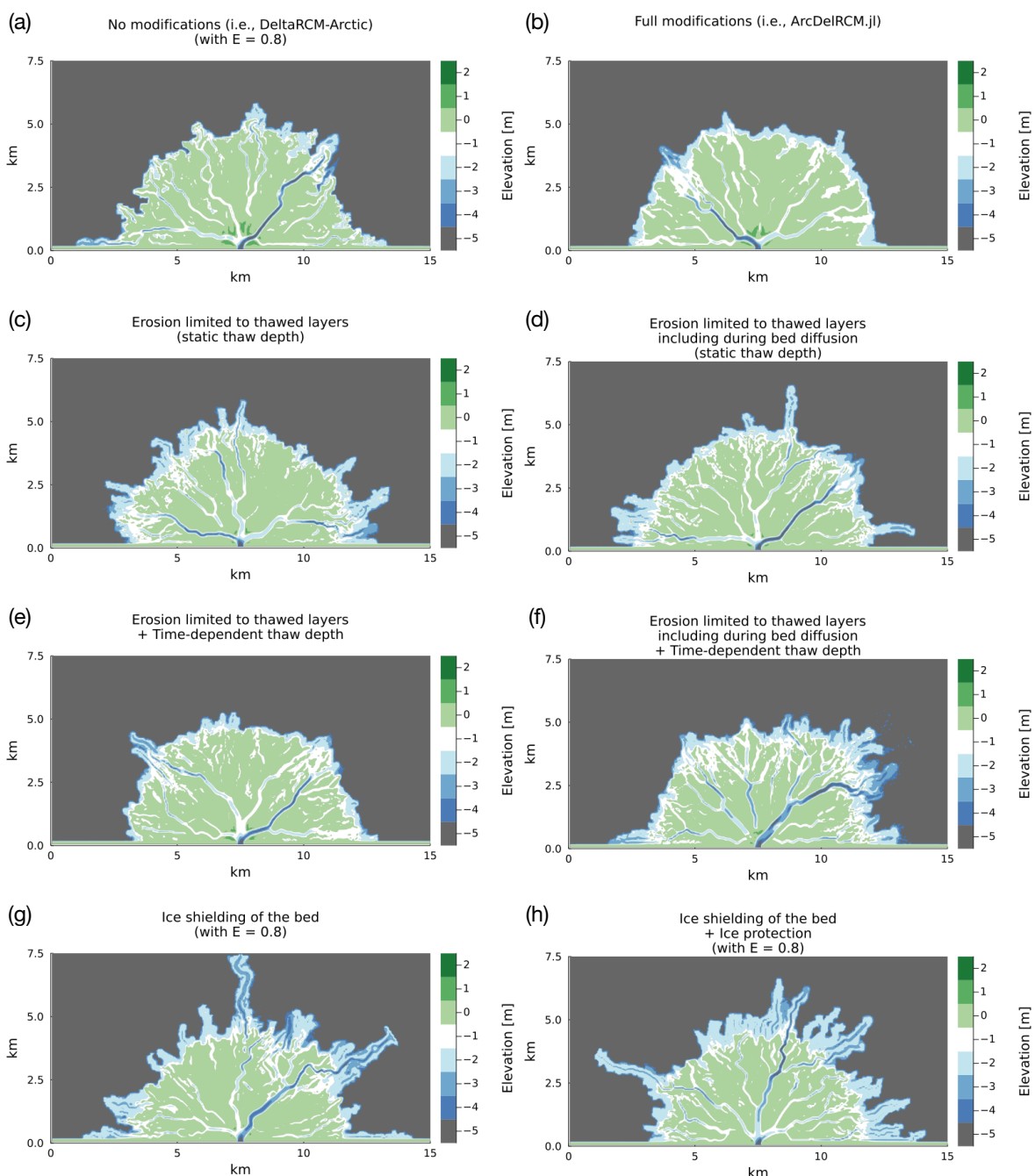

**Figure 5.** Bed-elevation output of DeltaRCM-Arctic (a), ArcDelRCM.jl (b), and with modifications described in Sect. 2.2.1 to 2.2.4 applied individually or in tandem, as indicated by the title of each panel (c-h). Note the 2 m ramp in the full model and the case in which ice shielding and ice protections are applied together.

across all cases in Figure 5 are identical, however. The cases where the erosion is limited to thaw depth (Sect. 2.2.3) but retain DeltaRCM-Arctic rules for bed diffusion (Sect. 2.1.5) uses $E = 0.8$ to match the other cases (Figures 5c, e).

All the cases with erosion limited to thawed layers instead of using the erodibility factor (Sect. 2.2.3) appear similar to each other (Figures 5c to 5f), but the cases in which the thaw depth is time-dependent (Sect. 2.2.2) has less tentacle-like deposits reaching seaward from channel outlets (Figures 5e and 5f).

The case with only ice shielding of the bed (Sect. 2.2.1) displays tentacle-like deposits (Figure 5g), similar to those in Figures 5c and 5d. The case in which bottom-fast ice is protected from flow-induced melt (i.e., ice protection; Sect. 2.2.1) in addition to ice shielding of the bed exhibits joining of deposition lobes around channel outlets to form a visible 2-m ramp around the delta, in addition to further tentacle-like deposits seaward (Figure 5h). The cases for DeltaRCM-Arctic and the full ArcDelRCM.jl are given for visual comparison (Figures 5a and 5b).

## 3.3 Quantitative Comparison of Delta Ensemble Runs

To assess quantitatively and statistically the individual modifications represented in Figure 5 and their potential influence on the ramp feature, we applied graph theory analyses as described in Sect. 2.3 on an ensemble of 105 realisations for each case. Figure 6 shows the resulting delta metrics. In addition to the cases shown in Figure 5, we also run variants of them that are not shown, but the metrics of which are included here. These variants are: all the runs where the erodibility factor is applicable with $E = 0.65$ (i.e., the cases that use DeltaRCM-Arctic rules for all erosion), the non-Arctic DeltaRCM runs, and the ArcDelRCM.jl run with unhindered bed diffusion (i.e., no thaw-depth limits, equivalent to $E = 1$, during the bed-diffusion step).

In terms of the number of nodes and edges, the cases divide roughly into three clusters: the cases with individual modifications turned on (Figures 5c-f), the DeltaRCM family, and the ArcDelRCM.jl family. Most of the variability overlap each other, however.

The number of connected components (reflecting the number of abandoned channels or subnetworks) and the graph density (actual versus all possible connections between nodes) display an inverse relation with each other. Here the cases can also be viewed as having the same three clusters as the number of nodes and edges. However, the separation is weaker (i.e., with more overlap across variability). Across all the cases that use the erodibility factor, $E$, those with $E = 0.8$ show a tendency towards less abandoned channels or subnetworks and thus are better connected (i.e., have higher graph density). The non-Arctic case (effectively meaning $E = 1$, but also ice-free) occupy roughly the same range as the $E = 0.65$ cases of DeltaRCM-Arctic.

The maximum diameters, measuring the longest of all the shortest paths between vertices, are not significantly different between all the cases considered. There is a tendency for longer single channels for the cases with ice-shielding and/or bottom-fast ice protection (also see Figures 5g-h), affecting the maximum diameter. The total lengths of all channels are also longer in these cases (which all use the erodibility factor as in DeltaRCM-Arctic). These are, in turn, similar to the lengths of channels in the DeltaRCM cases (including non-Arctic). On the other hand, the cases with erosion limited to thaw depth (but without ice-related modifications), regardless of specific variants, display tendencies to have shorter total lengths of channels. All ArcDelRCM.jl cases occupy the range of total channel lengths in between.

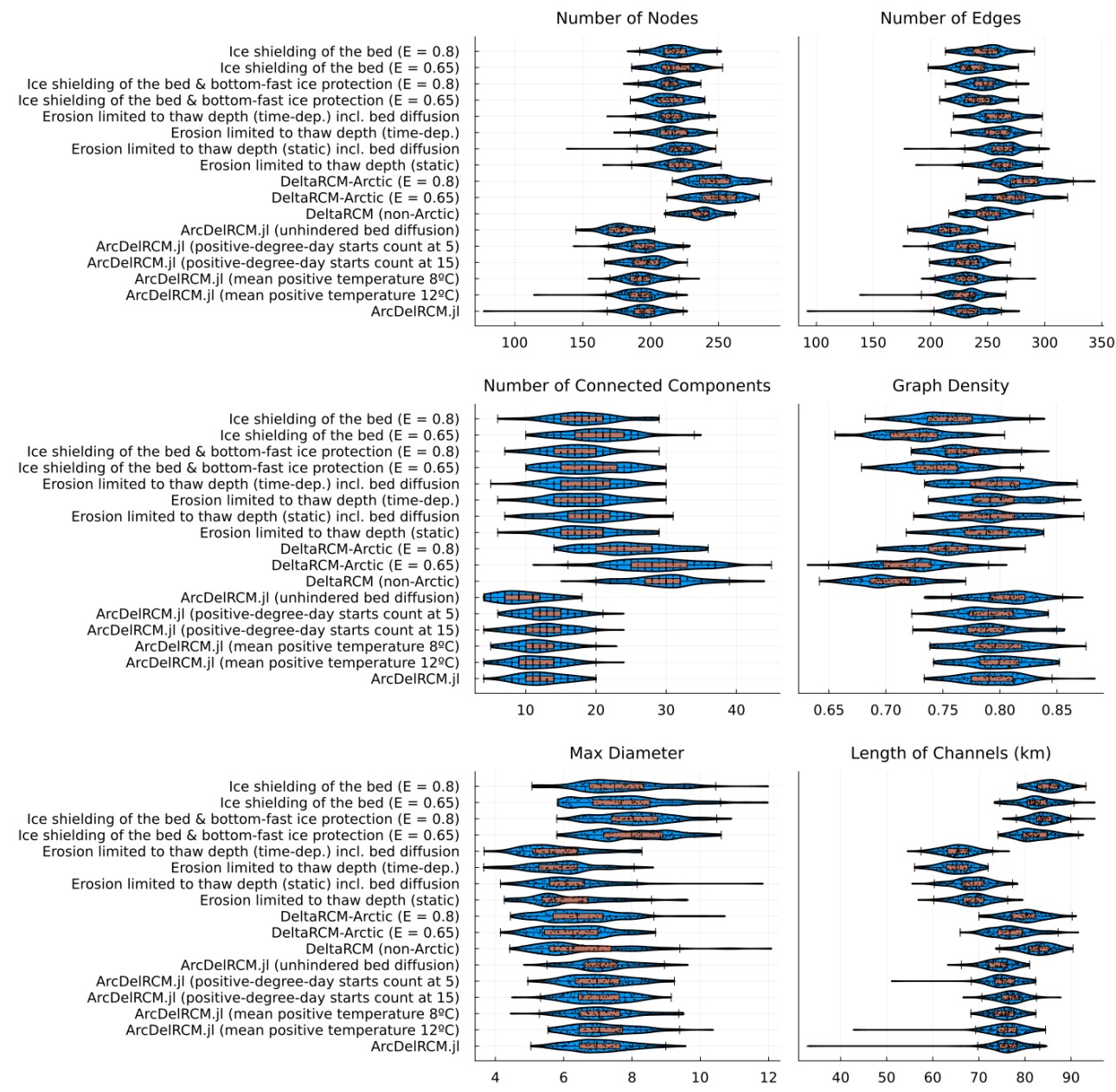

**Figure 6.** Plots showing delta metrics determined by the graph analysis we performed on each of the configurations described in Section 3.2. The blue violin plots show the kernel density estimate, the orange box and whisker plots show the central 50%, maximum, median, and minimum values, and the black dots show the actual values of the individual simulations. The labels along the vertical axes correspond to the titles of each sub-figures in Figure 5. The metrics of the variant cases not explicitly shown in this text (i.e., all cases with $E = 0.65$, ArcDelRCM.jl with unhindered bed diffusion, and the non-Arctic DeltaRCM) are included for comparison.

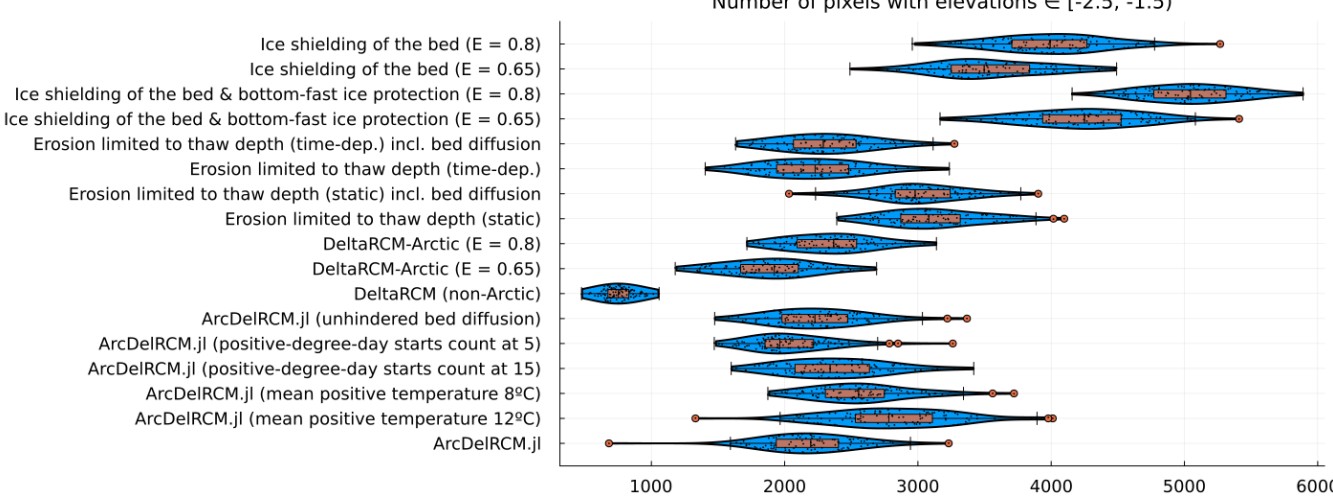

**Figure 7.** An analogus plot to Figure 6, but showing the number of pixels with elevation $\in [-2.5, 1.5)$ m. Since the modelled ramp feature that forms under $h_{ice,max} = 2$ m lies mostly in this elevation range, we use this value as a proxy to the size of the ramp, even though it inevitably include delta pixels that are not part of a ramp feature but happen to be in the same elevation range. This is evident when comparing the DeltaRCM-Arctic (E = 0.8) and ArcDelRCM.jl cases with Figures 5a and b. The pixels with relevant elevations are more concentrated around channel outlets in the former, while they are more distributed along the delta shore in the latter (forming the ramp).

To gauge the size of the ramp feature, which is not straightforward to quantify due to the irregular shape and distance from apex, we use the number of pixels with elevations $\geq -2.5$ and $< -1.5$ m as a proxy (Figure 7). The non-Arctic cases have the lowest counts of such pixels, whilst the cases with both ice shielding of the bed and bottom-fast ice protection have the highest.

Protection of bottom-fast ice appears to increase this count compared to ice-shielding of the bed alone. The cases with erosion limited to thaw depth, including all ArcDelRCM.jl cases, show a similar range in this pixel count, with the cases using a static thaw depth giving higher counts. Notably, both DeltaRCM-Arctic cases occupy the same range as the ArcDelRCM.jl cases, even though the spatial distributions of the pixels are different (see Figure 5a, b). Finally, between cases that use $E = 0.8$ and $E = 0.65$, the cases with $E = 0.8$ show higher counts of pixels within this elevation range.

**3.4 Lena Delta Approximants**

As a test case to mimic a large delta such as the Lena Delta, we ran simulations adopting spatial scales that had never been applied to ice-dominated delta using this family of models before. (However, the non-Arctic DeltaRCM has previously been applied on a similar spatial scale on the Mississippi and Selenga deltas by Moodie and Passalacqua 2021.) Specifically, we use $\delta c = 400$ m, $N_0 = 6$, $h_{0,min} = 10$ m, $h_B = 15$ m, $u_{0,min} = 1$ m s$^{-1}$, $S_{0,min} = 5 \times 10^{-5}$, $Q_s/Q_w = 3 \times 10^{-4}$ (roughly 10

440 times the average volume fraction measured in the Lena Delta; Boike et al., 2019), a sand fraction of 20% (Alekseevsky 2007, as cited in Fedorova et al. 2015), a maximum ice extent of 100%, $h_{ice,max} = 2$ m, $\gamma = 0.135$, and a time step of 1 day.

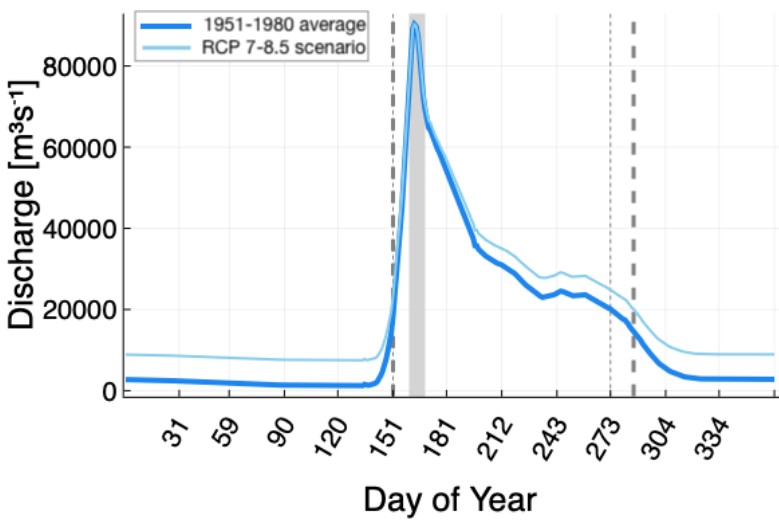

**Figure 8.** Daily discharge measured at STOLB station (thick blue line; GRDC Station Data 2903430, 2018), averaged from 1951 to 1980 inclusive. Discharge remains over 20000 m$^3$ s$^{-1}$ during the months from June to September (between the thin grey dotted lines), which is the simulation period for the 4-month Lena-approximant cases. The grey band spans the peak 10 days of discharge. The light-blue, thinner line shows the same discharge pattern, except the overall discharge has been scaled up by 35% (representing the RCP 7 - 8.5 scenario) whilst keeping the peak value and the shape of the curve the same. The period during which discharge is over 20000 m$^3$ s$^{-1}$ is longer, at 136 days (between the thick grey dash lines).

We would first like to find out the differences arising from assuming a 10-day model year (suitable for smaller Arctic deltas, as adopted by Lauzon et al., 2019; Piliouras et al., 2021) and from assuming a 4-month model year including the summer months (suitable for large Arctic deltas such as Lena Delta and Mackenzie Delta) due to the resulting different total sediment input per year. We ran one batch of simulations for 150 model years. Within this batch, the discharge $Q_w$ is treated differently in order to probe the difference it makes in terms of the number of days per simulation year and thus sediment input per year, given realistic $Q_w(t)$. The "full" simulation cases cover in each model year the 4 months from 1$^{st}$ June to 30$^{th}$ September (122 days). In these cases, $Q_w(t)$ is a time series constructed from daily discharge data from the STOLB station near the main channel into the Lena delta (GRDC Station Data 2903430, 2018), with daily values averaged from 1951 to 1980 inclusive (Figure 8).

The "10-day" cases have two variants: the "constant averaged peak discharge" case (Figure 9a) uses the averaged value from the peak 10 days of $Q_w(t)$ as the constant discharge; and the "time-variable discharge during peak" case (Figure 9b) uses the peak 10 days of the $Q_w(t)$ time series as the input discharge. The 10-day peak period is highlighted by a grey band in Figure 8.

The full 4-month cases are also divided into two variants: one in which the ice-melt period is 10 days (Figure 9c), similar to the aforementioned 10-day cases; and one in which the ice-melt period is 20 days long and is delayed by 20 days from the start of each model year (Figure 9d; motivation described in Sect. 2.2.7). In all cases, flow-induced melting is active

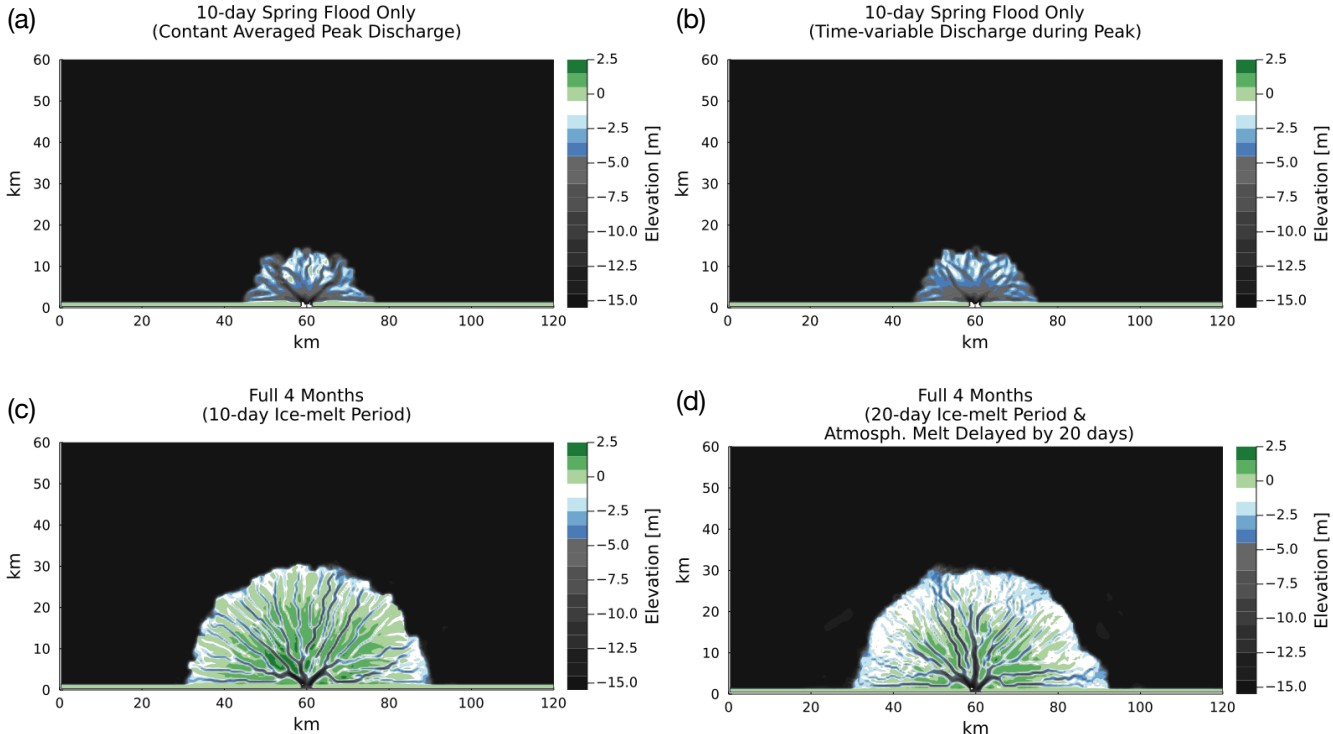

**Figure 9.** Bed elevations of deltas produced by running ArcDelRCM.jl for 150 model years on Lena-Delta-like spatial scales (see text in Sect. 3.4), with input discharge derived from daily measured values from GRDC Station Data 2903430 (2018). The top row (a, b) features deltas produced by running model years of 10 days each, which is also the ice-melt period. Discharge in these 10-day cases are taken from the peak 10 days of the time series and either (a) averaged and used as a constant value or (b) used directly as a 10-day discharge time series. The bottom row (c, d) features deltas that are produced by 4-month model years (June to September), using the full input discharge time series for the corresponding period. The case in panel c kept the ice-melt period of 10 days, whilst the case in panel d has an ice-melt period of 20 days and a delayed onset of atmospherically induced melting by 20 days from the start of each model year. Note the difference in size between the top and bottom rows, and the ramp feature around the delta in panel d.

(where allowed) throughout the whole of the simulation. We use "ice-melt period" to refer to only the time during which the atmospheric contribution is active (i.e., $t_{\mathrm{melt}}$ in Equation 5).

The resulting deltas in the 10-day cases are entirely under water and similar to each other in extent (Figures 9a and 9b). The 4-month cases also produced deltas that are similar in extent with each other (Figures 9c and 9d), but reaching twice as far from the inlet wall (or four times the area) as the 10-day cases. A ramp is also visible around the delta in the case with 20-day ice-melt period delayed by 20 days (Figure 9d), dipping from just below water level towards $2\,\mathrm{m}$, albeit with rougher surface with shallower depths ($\sim 1\,\mathrm{m}$; fourth contour level from the top) than in the small-scale cases in Sect. 3.1.

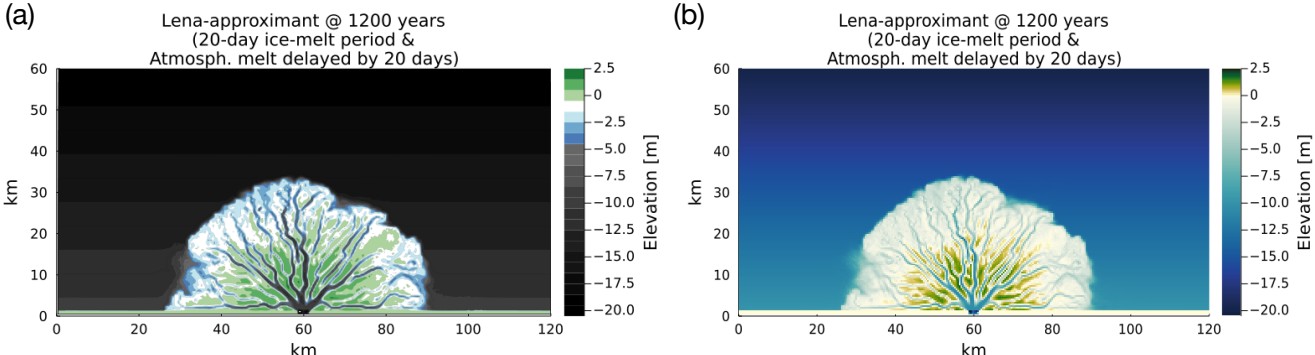

**Figure 10.** A delta produced by ArcDelRCM.jl after 1200 model years with configurations identical to those in Figure 9d, but with the low sediment-to-water volume ratio observed (Boike et al., 2019) and with a tilted ocean-basin bed motivated by the bathymetry of the Laptev Sea coast near the Lena Delta (see text in Sect. 3.4 for details). The panels show (a) the filled-contour view and (b) the gradient-coloured view of the same bed elevations. The latter is to facilitate visual comparisons with Figure 1.

As an additional demonstration of the model in approximating the scale of the Lena Delta, we run the simulation with the same parameters but with $Q_s/Q_w = 3 \times 10^{-5}$, reflecting the measured average volume fraction of sediment to water (Boike et al., 2019). This low sediment-to-water volume fraction requires a much longer run time to produce a delta. Therefore, we ran the simulation for 1200 model years (with the same $\Delta t = 1$ day). To further mimic the underlying ocean bathymetry on the Laptev Sea coast, where the Lena Delta is situated, we introduced a gradual tilt of the ocean basin elevation $h_B$: from 10m at the inlet wall to 20m on the opposite side of the simulation domain. The extent of the tilt and the 20-m maximum $h_B$ is motivated by inspecting the bathymetry of the Laptev Sea coast (Fuchs et al., 2021). The resulting delta is shown in Figure 10. The extent of the delta is similar to the 4-month cases (at 150 model years) in Figure 9, whilst the ramp feature has a clearer dip towards $\sim 2$ m depth than in the other Lena-approximant cases.

### 3.5 Ramp Feature under a Warming Climate

To explore how a warming climate might affect the ramp feature, we continued the simulation of the delta shown in Figure 10 for another 1200 model years. However, for this portion, we adopt an end-member scenario of Representative Concentration Pathway (RCP) roughly 7 to 8.5 (Stocker, 2014), which corresponds to $\sim 4°C$ of global warming by the year 2100. Under this scenario, maximum ice thickness during winter is not expected to reduce drastically (Nummelin et al., 2016; Sun et al., 2018) although some thinning has been suggested (Landrum and Holland, 2020). We therefore adopt $h_{\text{ice,max}} = 1$ m thickness instead of 2 m. The discharge at the Lena Delta has also been observed to be increasing in recent years (Fedorova et al., 2015). For this warming scenario, we also adopt an overall increase of 35 % in total discharge (Mann et al., 2022), whilst the peaks remain the same (motivated by Juhls et al., 2020, in which they found an increased winter discharge; see also Mann et al., 2022). The overall time period during which discharge is over $20000$ m$^3$ s$^{-1}$ is 14 days longer. This modified discharge

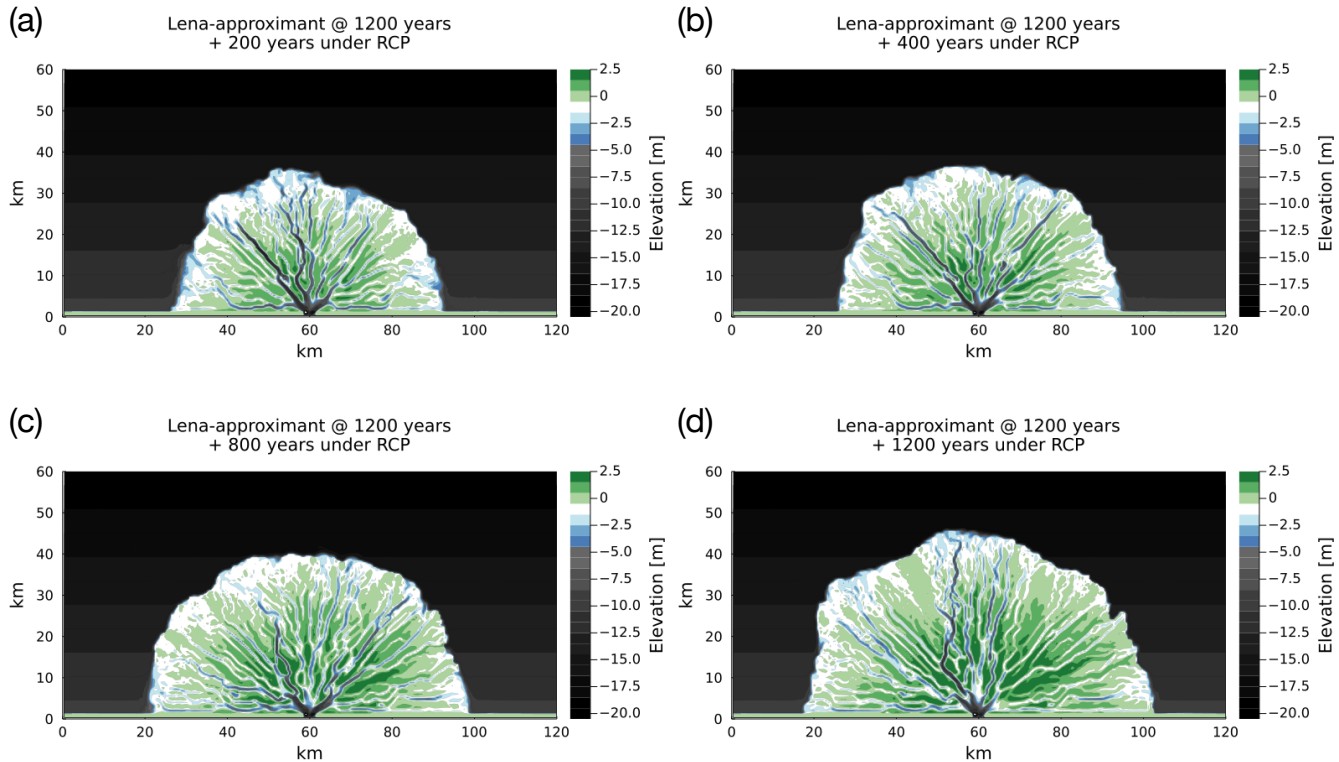

**Figure 11.** The continued evolution of the delta shown in Figure 10, except now under conditions possible in an end-member climate warming scenario (based roughly on RCP 7 to 8.5). As before, coloured contours reflect the bed elevations. The four panels show (a) 200, (b) 400, (c) 800, and (d) 1200 model years into this continued portion of the simulation. Note the degradation and disappearance of the ramp feature.

pattern is shown in Figure 8. Furthermore, based on the surface-temperature increase (and the rate of temperature increase during the spring and summer months; Sun et al., 2018), the atmospheric heat-induced melting of ice cover is brought forward by 10 days. The duration is also shortened by 10 days due to the thinner ice. All other parameters remain the same as described in Sect. 3.4.

Figure 11 shows various snapshots of the continued evolution of the Lena-approximant delta. Under the warm conditions, the ramp feature has diminished by the 200-year mark (Figure 11a), and becomes mostly disrupted by the 400-year mark (Figure 11b). From 800 years onwards, no continuous ramp features remain (Figure 11c and 11d).

## 4  Discussion

The results (Figure 4) demonstrates that ArcDelRCM.jl is able to reproduce the 2-m ramp around Arctic deltas (contrast with Figure 4a), and Figure 4d shows that the ramp is related to the maximum ice thickness ($h_{\text{ice,max}}$). As observed in Lauzon et al.

(2019) and Piliouras et al. (2021), off-shore deposits do occur as tentacle-like features in DeltaRCM-Arctic, especially when thick ice is imposed on a shallow domain ($h_{\text{ice,max}}/h_B = 3/5$; Figure 4c). However, our modifications in ArcDelRCM.jl led to similar features in addition to a continuous ramp not limited to around channel outlets (Figure 4d). Increasing maximum ice extent to 100% led to a more prominent ramp (Figure 4f), further supporting that ice is the driving factor behind the ramp feature (Reimnitz, 2002). However, having a deeper ocean basin (i.e., more accommodation space) appears to impede the development of the ramp, as seen in Figure 4e (in stark contrast to Figure 4d). This suggests that the available space under maximum ice thickness ($h_{\text{ice,max}}$), and thus the dynamics of transport under ice cover around the shore of deltas, plays a determining role in the formation of ramp features. The less space there is, the more flow is constricted and the faster sediments build up against the bottom of the ice cover, forcing lateral deposition, which forms a continuous band that becomes the ramp feature. Previous work by Lauzon et al. (2019) and Piliouras et al. (2021) using DeltaRCM-Arctic observed that off-shore deposits increase with $h_{\text{ice,max}}$ and thus with decreasing accommodation space in the ocean basin. Our results are therefore in agreement.

Figures 5 and 6 together show the various effects of individual modifications detailed in section 2.2. Using the original DeltaRCM-Arctic model (Figure 5a) as a baseline, limiting erosion to thawed layers whilst keeping a static thaw depth (Figures 5c and 5d) and foregoing the erodibility factor have the effect of allowing existing channels to erode more easily down to the thaw depth. The more readily available sediments combined with some retained constriction to the flow (from the thaw-depth limitation) lead to more sediments being carried seaward along the same channel paths, resulting in tentacle-like off-shore deposits at $\sim 2\,\text{m}$ depth (reflected in the pixel count in Figure 7). Further making the thaw depth time-dependent (as in Figures 5e and 5f), erosion can still occur (without the erodibility factor) down to the thaw depth, but the thaw depth now stays shallower throughout (approximately between $0.2$ and $0.3\,\text{m}$). This leads to similar characteristics with the cases in Figures 5c and 5d in the central (ice-free) part of the delta, but the shallower erodible depth leads to less off-shore deposits (Figure 7) and more short channels in the ice-cover zone branching in search of unblocked pathways. The latter helps spread sediments more evenly along the delta shoreline.

The relationship between the extent of off-shore deposition and erodible sediments (either through erodible thaw depth or the erodibility factor) is echoed in all the cases that uses DeltaRCM rules for erosion. Cases with $E = 0.8$ results in more pronounced off-shore deposits at $\sim 2\,\text{m}$ depth—potential building material for the ramp—than those with $E = 0.65$ (Figure 7).

The cases where the thaw-depth erosional limit is included in the bed-diffusion process (Figures 5d and 5f) do not appear to be substantially different from the cases where it is excluded (Figures 5c and 5e). The same is reflected in all the graph and pixel-count metrics (Figures 6 and 7). This shows that bed diffusion has a relatively minor effect on the delta's form compared to the flow-driven erosion and deposition.

The ice-shielding case (Figure 5g) has identical erosion mechanisms as in DeltaRCM-Arctic (Figure 5a), except erosion and deposition are blocked wherever ice is bottom-fast. This enhances flow constriction by ice, which focuses erosion on the few unblocked pathways (e.g. sub-ice channels), and leads to sediment being carried farther seaward. The result is a tentacle-like off-shore deposition pattern similar to Figures 5c and 5d, although the underlying mechanisms differ. The far-reaching tentacles yield a higher total length of channels and a larger maximum-diameter metric (Figure 6).

Given the locations and elevations of the tentacle-like off-shore deposition, either in DeltaRCM-Arctic or the partial ArcDelRCM.jl configurations of Figures 4c-g, they are likely the foundation of the ramp features, as Lauzon et al. (2019) and Piliouras et al. (2021) suggested. As they extend outwards from various channel outlets, however, they are still separated from each other instead of forming a band-like ramp feature around the delta shoreline.

Figure 5h shows that the protection of bottom-fast ice from flow-induced melting (Sect. 2.2.1) is ultimately the process that gave the model the ability to produce the ramp feature. In this case, the erosion/deposition rules are the same as in the ice-shielding case (Figure 5g), giving a similar tentacle pattern (and similarly elevated metrics of maximum diameter and total length of channels; Figure 6). However, the bottom-fast ice survives for longer when protected from melting. This gives rise to two effects: (i) sediment deposition is more evenly spread out along the shore due to constriction of flow by ice (similar to the cases of Figures 5e and 5f); (ii) previously deposited material is protected, allowing depositional lobes to expand and merge subaqueously without being deposited over top. The ramp feature is thus stabilised and sustained as a result.

Considering what we observed in the model outputs so far, the following balance may be at play in the formation of the ramp feature: The existence of ice constricts flow and promotes off-shore deposition (Lauzon et al., 2019; Piliouras et al., 2021). The erosional rules of ArcDelRCM.jl further allow the erodible layers (i.e., thawed layers) to be eroded to form more substantial off-shore deposits. The introduction of time-dependent thaw depth leads to shallower thaw depths within the same 10-day model year, offsetting the previous effect somewhat, but the ice constriction is enhanced. Finally, the shallower thaw depth and ice shielding of the bed work in tandem with the longer survival of bottom-fast ice to both constrict flow and cause it to spread out laterally. This leads to more even deposition along the shore, which is then protected by the longer-surviving bottom-fast ice, resulting in the ramp. As a pair of examples, Figure C1 shows the spatial views of the thaw-depth pattern corresponding to simulations shown in Figures 4b and 4f. These thaw-depth patterns provide an indication on where the aforementioned processes and balances are active.

Many of the individual modifications are made to ensure logical consistency. For instance, the protection of the bed by bottom-fast ice (Sect. 2.2.1) and limiting erosion to only thawed layers (Sect. 2.2.3) directly follow from the protection of bottom-fast ice from flow-induced melting. Time-dependent thaw depth (Sect. 2.2.2) also becomes necessary due to the fact that bottom-fast ice transfers heat conductively during winter months and delays the progression of the thaw depth during a model year, reducing erosion even during summer months. The combined effects of the individual modifications described above are what give rise to the form of the simulated deltas with ramp features in ArcDelRCM.jl (Figures 4b and 5b).

To assess if further insights could be gained through quantitative measure of network structures, we examine the four graph metrics on top of Figure 6. The cases with ice- and erosion-related modifications of ArcDelRCM.jl switched on in isolation occupy the range between full ArcDelRCM.jl cases and the DeltaRCM family of cases. The ArcDelRCM.jl cases appear to have more overlap with either the non-Arctic DeltaRCM or the DeltaRCM-Arctic case with $E = 0.8$, depending on the metric. This may be a manifestation of the hybrid nature of the erosion rules of ArcDelRCM.jl (i.e., erosion is unhindered for thawed vertical sediment cells, but restricted for frozen vertical cells until the thaw depth deepens in the next time step) versus the more constant fractionally scaled (in flow-speed thresholds or the magnitude of bed diffusion) erosion of DeltaRCM-Arctic, as explained in Sect. 2.2.3. In the context of the ramp feature, the better connected channels (graph density) and fewer abandoned

subnetworks (number of connected components) could translate into a more consistent feeding of sediments to all segments of the delta shore, which would help build an evenly distributed ramp. Future work could apply more comprehensive analyses to further interpret the differences or similarities between these different Arctic delta models, but these are out of the scope of this ramp-focused work.

In order to gauge the long-term (multi-year) evolution of ramp features in major Arctic deltas, we first address the duration covered by each model year. To this end, Figure 9 demonstrates the importance of delta activities (and the associated sediment input) outside of the peak flooding season. The discharge of large drainage basins such as the Lena watershed remains significant during the summer months (about 53% of the annual discharge; Holmes et al., 2012), even though it is at a much lower level compared to the peak (GRDC Station Data 2903430, 2018, and Figure 8). The deltas produced by taking into account the summer months are 4 times the area (and more if one considers only above-water areas) of the equivalent ones that take into account only the peak flooding period. Therefore, for our purpose, we adopt a 4-month model year rather than a 10-day one adopted by Lauzon et al. (2019) and Piliouras et al. (2021). Whether a constant discharge or a time-variable one is used during the peak period does not appear to have an impact on overall areal extent (Figures 9a and 9b).

Regarding the ramp feature, not only is it affected by the under-ice depth of the ocean basin, but also by the timing of the ice-melt. This is demonstrated in Figures 9c and 9d. With $15\,\mathrm{m}$ of accommodation space under $h_{\mathrm{ice,max}} = 2\,\mathrm{m}$, deposition that leads to ramp formation is not favoured. However, when an observation-informed timing and duration of bottom-fast ice melt is introduced (Figure 2), the ramp feature begins to emerge (Figure 9d). This corresponds to how bottom-fast ice resting on the ramp remains in place whilst the ice in other areas of the delta is flushed or melted away, and only starts to break up after the peak flood is over (Figure 2). However, the deeper ocean and the different discharge pattern led to slower build-up of deposits during ice cover and more deposition during ice-free summer, resulting in the ramp being more hummocky and unevenly graded than in the small-scale, "benchmark" cases in Sect. 3.1.

In reality, the Lena Delta has a much lower sediment volume discharge (roughly a tenth) than we used in our demonstration cases in Figure 9. The ocean bed on which it formed may not have been flat, but rather tilted from the coast towards the Laptev Sea. Taking these into account, the simulation in Figure 10 took 8 times as long to produce a delta of similar size to the one in Figure 9d, but with a ramp feature that is slightly wider and dips closer to $2\,\mathrm{m}$ in depth. This is consistent with the aforementioned observation that available depth below ice plays an important role in the formation of ramp features.

Continuing the simulation of Figure 10 under a strong warming scenario (for discharge and ice-cover) akin to RCP 7-8.5, we find that an existing ramp feature could degrade on a time scale of centuries (Figures 11a and 11b), and effectively disappear within a millennium (Figures 11c and 11d). Marine processes, such as warming sea water, wave attack, alongshore currents and sea ice entrainment, may accelerate these time scales. The degradation of the ramp could affect transport distance of sediments, impacting the release or sequestration of soil organic carbon (Overeem et al., 2022). The reduction of the shallow-water platform provided by the ramp can also impact the delta ecosystems (Lopez et al., 2006). Deltas will also lose a potentially important buffer against coastal erosion (Dean and Dalrymple, 2002).

We note, however, that important ocean-driven processes are missing in the model, resulting in differences in smoothness and outer-edge shapes between the modelled ramps and those observed in reality (Figure 1). Surges (e.g. during winter storm)

can thicken (from underneath) the ice cover that become bottom-fast over the ramps (Reimnitz, 2002), which could enhance the protection by bottom-fast ice during spring. Moreover, compared to the deltas produced by DeltaRCM-Arctic and ArcDel-
RCM.jl, the observed slopes of the sediment bed beyond the outer-edge of the 2-m ramps are much gentler, typically dipping from 2 m to $> 20$ m over $\mathcal{O}(10)$ km rather than $\mathcal{O}(0.1)$ km (Reimnitz, 2002; Are et al., 2002). The lack of the gentle dipping may have resulted from the limitation of the models having abrupt thresholds for deposition (Sect. 2.1.4) and in the classification of "on-delta" and "ocean" grid cells during the flow routing (Sect. 2.1.2), in which most of the sediments carried in a packet tend to get deposited as soon as it leaves the "on-delta" cells. The gentle slope observed in reality could be a result
of marine processes such as sediment re-suspension by waves (especially during fall storms), which can be effective to water depths exceeding $10$ m (Heim et al., 2014), and transport by currents. Wave-action re-suspends sediment, especially during fall storms. Sediment in the water column acts as nuclei for frazil ice formation and can be integrated into the forming ice pack (Reimnitz et al., 1992). The formation and export of ice during the fall can remove material from the ramp, as can anchoring of ice in winter and spring to the seabed (Krumpen et al., 2020). Once anchored, continued cold temperatures create a seasonally
frozen layer in the sediment beneath the bottom-fast ice (Osterkamp et al., 1989). This material can be exported once the ice is lifted in spring (Reimnitz et al., 1987). Both processes, entrainment of frazil ice and adherence of frozen sediment, lead to ice-rafting (Are and Reimnitz, 2008). Sediment can also be moved by ice-gouging and ice-bulldozing, when thicker ice masses are ploughed into the seabed (Maznev et al., 2019; Ogorodov et al., 2018). In the context of the simulations shown above, these marine and marine-ice processes could affect not just the exact morphology, but also the time scale of ramp formation
(balancing between the sediment supply to build, and the reworking to sculpt the overall ramp), although not prevent it. The same could, of course, also affect the time scale of degradation of the ramp under a changing climate. Future work on the model could focus on improving the capability of off-shore dynamics, such that a full picture of a delta's formation and destruction can be built.

## 5  Conclusions

The 2-m ramp feature is an integral and ubiquitous feature in Arctic deltas. The morphology and location of the ramp feature means that it could play an important role in the surrounding ecosystems (Lopez et al., 2006), in diffusing wave energy to protect the delta from coastal erosion (Dean and Dalrymple, 2002), and in enhancing carbon sequestration (Overeem et al., 2022). Although it has been suggested that the formation and existence of these ramps may be ice-related (Reimnitz, 2002), we set out to explore from a modelling perspective the conditions and processes that could give rise to the ramps.

To this end, we took the approach of using the explanatory insights (Bokulich, 2013) of RCMs, and have written the ArcDelRCM.jl model based on the published descriptions of DeltaRCM (Liang et al., 2015b; Liang, 2015; Perignon, 2018) and DeltaRCM-Arctic (Lauzon et al., 2019; Piliouras et al., 2021). Even though previous work using DeltaRCM-Arctic (Lauzon et al., 2019; Piliouras et al., 2021) have shown tentacle-like off-shore deposits at the right elevation range ($\sim -2$ m) under conditions where the accommodation space between winter ice cover and the ocean bottom is small enough, no continuous
(band-like) ramp feature could be produced.

Therefore, we tested a series of physical-rule modifications to explore the origins of ramp formation. The resulting Ar-cDelRCM.jl contains the following modifications over the base DeltaRCM-Arctic: (i) the protection of bottom-fast ice from flow-induced melting; (ii) the shielding of the bed by bottom-fast ice; (iii) time-dependent thaw depth; (iv) the limiting of all forms of erosion to thawed layers only (instead of using an erodibility factor); (v) the ability for users to specify the time-step size (with internal checks for numerical stability); (vi) the ability to use a time series for input discharge and its related parameters.

We showed that ArcDelRCM.jl can produce the ramp feature. We have found that the modelled ramp feature is indeed related to the winter ice cover, with its depth determined by the maximum thickness of winter ice. We have also found that the prominence of this ramp feature is affected by three factors: (i) the thickness and extent of winter ice, (ii) the available depth in the ocean basin under the winter ice cover (i.e., accommodation space; this confirms previous work by Lauzon et al. 2019 and Piliouras et al. 2021), and (iii) the timing of the melting of bottom-fast ice from atmospheric heat. Simulations of Lena-scale deltas also suggest that a break up of bottom-fast ice (simulated by a delayed onset of atmospheric melting), which is widespread on the ramp feature, plays an important role in the formation and growth of the ramp feature. The bottom-fast ice protects the ramp from degradation during peak flow.

To further elucidate the processes responsible for the ramp feature, and to ensure that the results are not the product of some isolated random events, we ran ensembles of 105 realisations for multiple configurations. These configurations range from the DeltaRCM-Arctic reconstruction with which we began, through individual modifications of ArcDelRCM.jl being enabled in isolation or in groups, to the full ArcDelRCM.jl that includes all the new physical rules.

We found that the protection of bottom-fast ice from flow-induced melting is ultimately responsible for the ability of the model to reproduce the 2-m ramp features. However, the other modifications are not only necessary for internal logical consistency with the first, they also contribute individually to the under-ice deposition pattern and its subsequent preservation required for the ramp feature to form. Specifically, we found that an enhanced constriction leads to more off-shore deposits (consistent with Lauzon et al. 2019 and Piliouras et al. 2021) and to more even spreading of deposits along the delta shore as the flow searches for unblocked pathways. Such constriction results from the combination of shielding of the bed from erosion/deposition by bottom-fast ice and the different timing and magnitude of erosion resulting from the updated erosional rules. Acting together, deltas produced by ArcDelRCM.jl appear to have less abandoned parts of the channel network and the channels are better interconnected, as shown in our graph analyses of the ensemble runs.

Taking the model to explore the ramp features' evolution in large Arctic deltas such as the Lena Delta, we first demonstrated that the months outside of the peak spring-flood season (and thus the sediment input they bring) are significant contributors to an Arctic delta's evolution and cannot be neglected by using a 10-day model year (as was adopted for general, smaller deltas in Lauzon et al., 2019; Piliouras et al., 2021). In a set of Lena-like simulations, we found that the inclusion of summer months (from June to September), instead of limiting to the peak-flood period of 10 days, led to a quadrupling of the delta area under similar conditions.

When compared to bathymetry data, the simulation-produced ramp features that have different elevation smoothness and outer-edge shapes (i.e., the underwater "shorelines"). This may be due to the lack of marine and marine-ice processes in

the model and to the clear distinctions between "delta" and "ocean" grid cells, affecting off-shore sediment transport. These limitations not only impact the exact morphology produced, but may also impact the time scale of formation, growth, or (under climate change) deterioration of the ramps. Future work could focus on addressing these limitations in order to improve the model's capability in predicting the future of Arctic deltas under an increasingly warm climate.

In a sequential pair of simulations (lasting millennia in model time) meant to closely mimic the Lena Delta under present-day and future ice and discharge conditions, we found that a formed ramp feature can degrade and effectively disappear on a time scale of centuries under an extreme climate-warming scenario akin to RCP7-8.5. This time scale could be accelerated further by ocean processes not included in the current model. Such degradation and disappearance of the ramp feature can impact the transport of carbon-carrying sediments, affect the delta ecosystems, and reduce future buffering of Arctic delta shorelines against coastal erosion.

*Code availability.* The source code of the specific version of ArcDelRCM.jl used in this study is published as a supplement citable as Chan (2023) and archived by GFZ Data Services. Any future updates of ArcDelRCM.jl can be found on the GitLab project page at: https://gitlab.com/nhchan/arcdelrcm.jl. The source codes used for the graph analyses are available in a GitHub repository via the link: https://github.com/trettelbach/arctic_delta_analysis.

## Appendix A: Other Feature Additions in ArcDelRCM.jl

### A1 Inheritance of Simulation States

A convenience-motivated feature addition is the ability to start a simulation from an output state given by another simulation. This would allow users to investigate multiple change scenarios that occur after the formation of a delta, such as sudden increase in discharge or in the duration of spring floods. This has been utilised in the case of Sect. 3.5. It could also be used to break very long simulations into stages to mitigate the risk of a computing-system crash.

### A2 Pre-exisiting Island Blocks

Users can specify islands in the ocean basin of the simulation domain, where no physical processes can occur (similar to the inlet wall). This provides the ability to mimic, albeit simplistically, islands such as Arga in the Lena Delta. Geometries such as rectangular and elliptical are available initially; more can be added in the future.

### A3 Bed Geometries

The initial bed of the ocean basin can have non-uniform depths (i.e., variable $h_B$). Simple tilt geometries such as linear (from inlet side wall towards the ocean) and radial (from the centre of the inlet channel outwards) are available, in which users can specify the distance over which the ocean bed varies from a specified depth to the $h_B$ specified in the simulation-domain parameters. This has been utilised in Sect. 3.4 and 3.5.

## Appendix B: The Choice of the Initial Positive Degree Day Index

The simplistic initial starting count of 10 days to calculate the positive-degree-day index, $I$, is chosen by balancing a few factors that can vary over time and specific sites: (i) comparison between the daily temperatures at the Lena Delta extracted from the ERA-Interim reanalysis data (Dee et al., 2011) (Figure B1) and the approximate timing of a typical onset of the spring flood season there (around 1st June); (ii) the measured water temperature in the Lena River showing that it crosses above $0°C$ near the onset of the spring flood (Juhls et al., 2020); and (iii) the fact that all the simulated cases start when discharge is above a certain threshold (either covering the peak 10 days or when the discharge rises above $20000\,\mathrm{m^3s^{-1}}$; details in Section 3), which occur some days after the temperature becomes positive. The temperature we used to calculate $I$ is the average daily temperature (Figure B1) from 1st June to 30th September, which is $4°C$. In addition to the graph metrics of the results presented in Section 3.2, Figure 6 shows the same metrics from the cases identical to the standard ArcDelRCM.jl runs except with $I$ starting count from 5 and 15 days and (independently) the average positive temperature being $8$ and $12°C$. They show that, within the ranges tested, these parameters do not affect the resulting deltas beyond the internal variability of each case. To help illustrate this visually, an example of each case is shown in Figure B2.

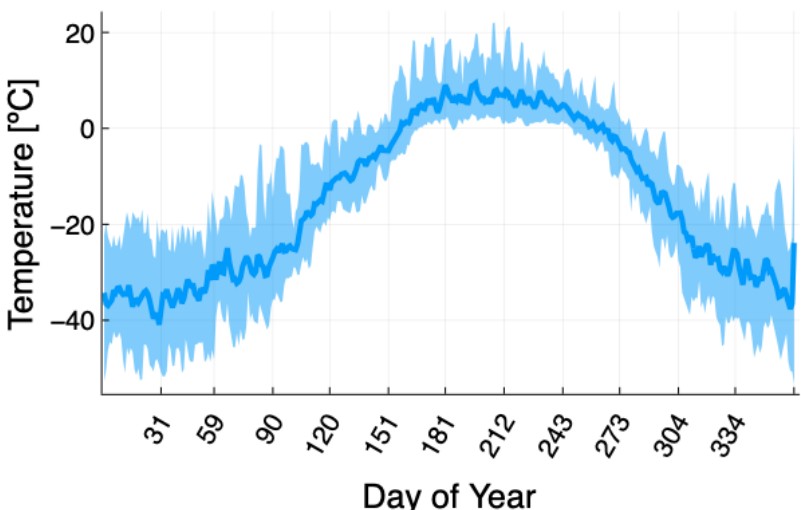

**Figure B1.** Daily temperature extracted from ERA-Interim reanalysis data (Dee et al., 2011) and averaged over all years from 1951 to 1980 (i.e., the same period from which the average daily discharge in Figure 8 is obtained). The lighter-shaded ribbon around the average line shows the range of daily temperatures during the same period.

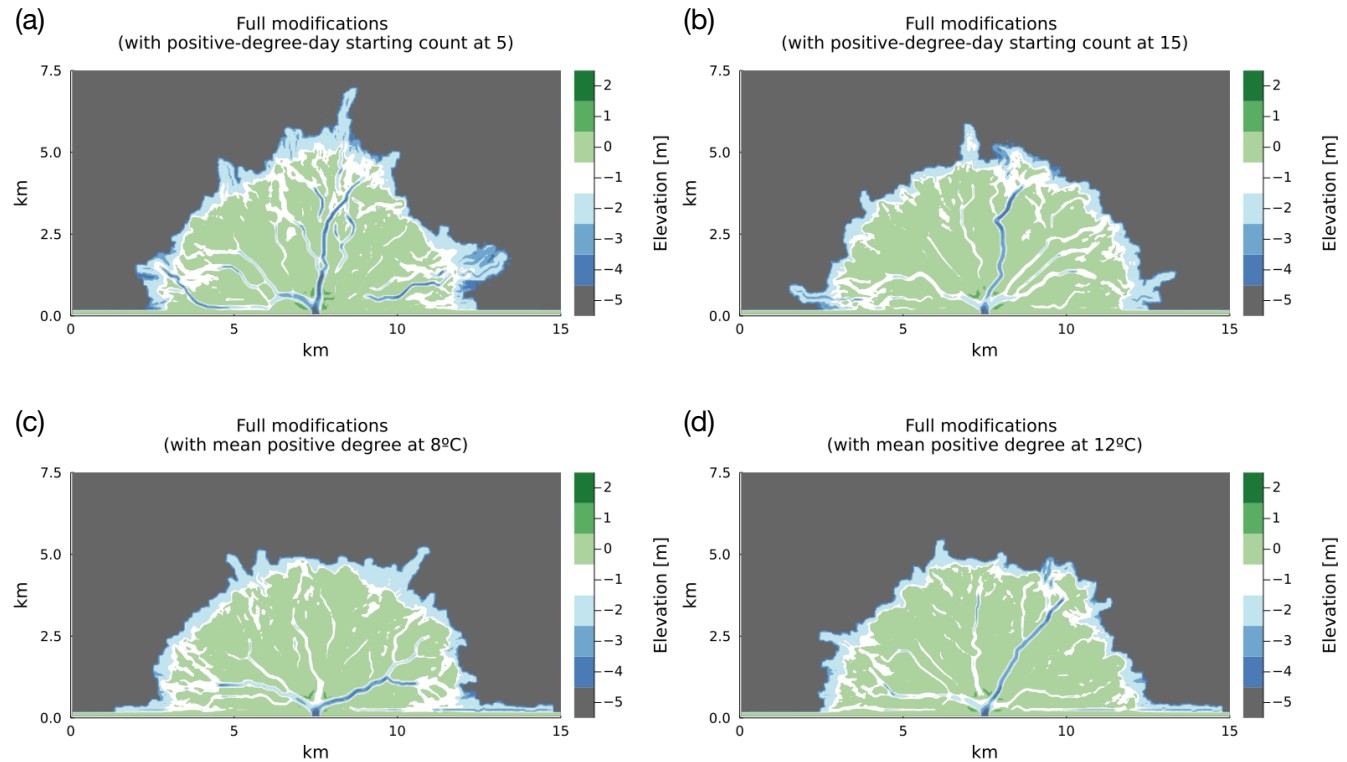

**Figure B2.** Examples of model outputs that use identical setup and parameters as Figure 4b, except for one of the following parameters: the positive-degree-day index, $I$, starts the count at (a) 5 days and (b) 15 days instead of 10; the average positive temperature used to calculate the positive-degree-day index is (c) $8°C$ and (d) $12°C$ instead of $4°C$.

## Appendix C:  Ground Freeze-Thaw Pattern

Figure C1 shows the time-variable thaw depth in the time steps around the final winter in the simulations of Figures 4b and
4f. The time snapshots shown are from the time step before the start of the final model year through to two time steps after
(i.e., from step 4970 to 4973 out of 5000, not counting the 300-step ramp-up phase). The second time snapshot (i.e., step
4971) correspond to the imposition of maximum ice thickness ($h_{\mathrm{ice,max}}$) and extent. Since ArcDelRCM.jl forgo the use of the
"permafrost" label (Sect. 2.2.2 and 2.2.3), direct comparison with distribution maps of "permafrost" or "frozen" cells from
DeltaRCM-Arctic output is not meaningful. To provide a visual sense of how the ground is frozen on the delta bed, we plot the
depth of thaw instead. The thaw-depth patterns provide an indication on where the various processes and balances discussed
in Sect. 4 are active. The figure also shows that the thaw depth is directly influenced by winter ice cover, since bottom-fast ice
leads to the top layer being also frozen in our model. The resulting step division between deeper and shallower thaw depths

does not delineate the ramp feature well, with the shallower thaw depths continuing into the ramp's boundary. The ice-related protective effects are thus active on parts of the ramp itself, as expected.

*Author contributions.* The initial idea came from Langer, Braun, and Huppert. Chan wrote the source codes for all the delta models in Julia, led the scientific modifications that led to ArcDelRCM.jl, obtained the GRDC discharge data, and executed all the model simulations. He also drafted the initial text of the article and made all the figures except 1 and 3. Huppert and Braun provided the background and guidance for Chan during the early phase. Langer participated throughout the project and provided guidance and domain knowledge on permafrost and the Arctic, provided the ERA temperature data, and put Chan in communication with Juhls, Rettelbach, and Overduin. Overduin and Juhls

gave expert knowledge of Arctic deltas and brought the ramp features to the other co-authors' attention. Juhls provided data and references regarding bathymetry, ice breakup, and water temperature. Juhls also made Figures 1 and 2. Rettelbach provided the codes and expertise related to the graph analyses. All authors contributed to subsequent editing of the draft manuscript.

*Competing interests.* The authors have the following competing interests: Co-author Jean Braun is a member of the editorial board of Earth Surface Dynamics.

*Acknowledgements.* Moritz Langer is supported through a grant by the Federal Ministry of Education and Research (BMBF) of Germany (No. 01LN1709A, Research Group PermaRisk). Chan thanks Laurens Bouwer from the Climate Service Center Germany (GERICS), part of the Helmholtz-Zentrum Geesthacht, for pointing him to the GRDC discharge data during a chat in a workshop event. We also acknowledge the use of imagery from the NASA Worldview application (https://worldview.earthdata.nasa.gov/), part of the NASA Earth Observing System Data and Information System (EOSDIS). We thank all the anonymous reviewers and community-comment authors, Jayaram Hariharan and

Lawrence Vulis, for their detailed and constructive reviews that helped to improve our manuscript.

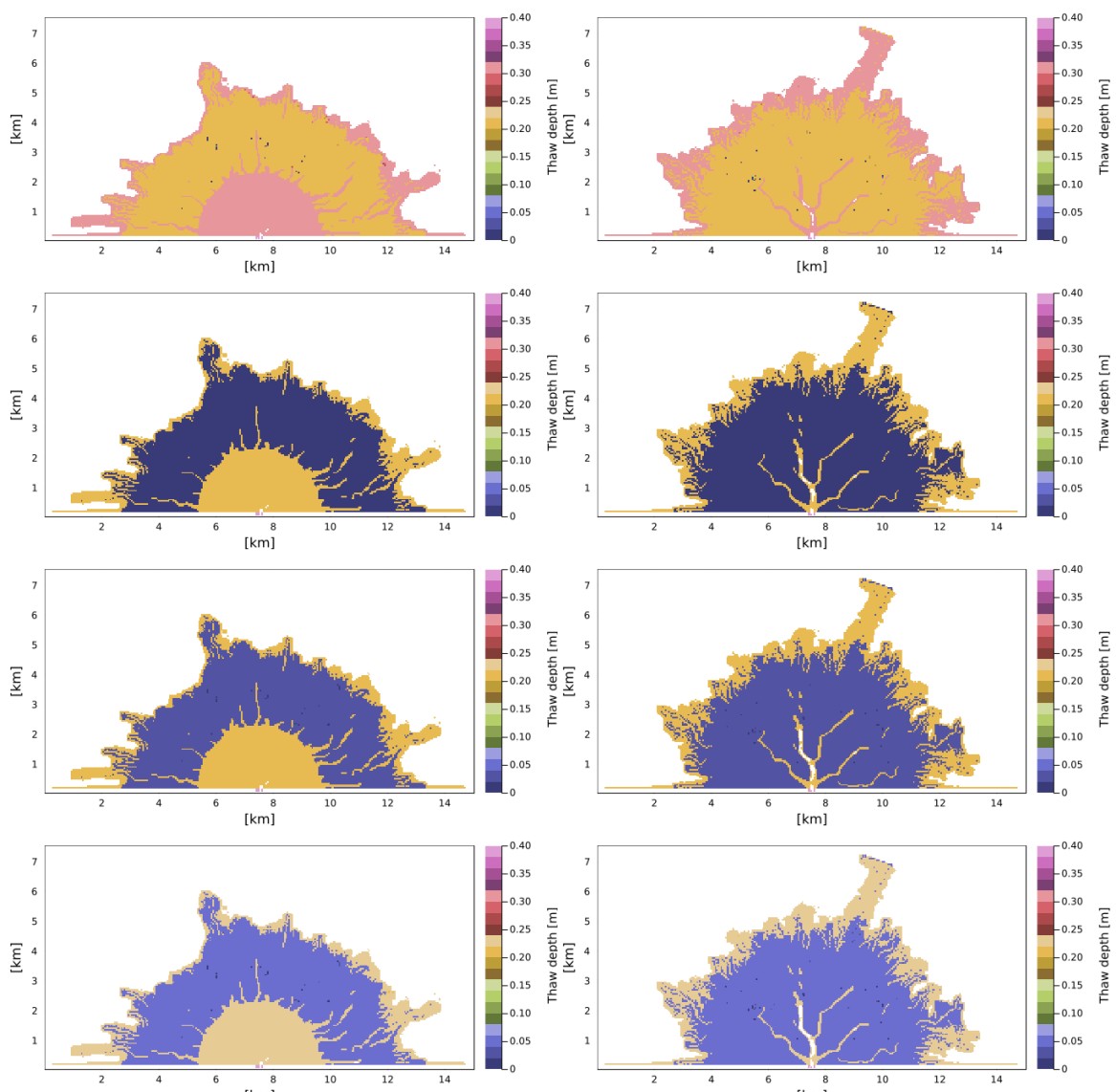

**Figure C1.** Evolution of thaw depths in the time steps around the final ice-cover maximum in the simulations in (left column) Figure 4b and (right column) Figure 4f. The time steps shown here are, from top to bottom, step number 4970 to 4973 out of 5000 (not counting the 300-step ramp-up phase). These correspond to the time step before the last winter ice maximum of the simulation through to two steps afterwards. Thaw depths are defined only where there are at least one vertical grid cell of sediments accumulated. Note that these plots show only the thaw depths from the top of any accumulated sediment column, and do not reflect elevation information. The contrast between the left and right columns, corresponding respectively to ice extents of 40% and 100%, shows that the thaw depths are directly influenced by the winter ice cover in our model. These thaw-depth patterns provide a visual sense of how the interplay and balances of processes in ArcDelRCM.jl, as discussed in Sect. 4, are spatially distributed.

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
