# Peer review of "Arctic Delta Reduced Complexity Model and its Reproduction of Key Geomorphological Structures"

_Earth Surface Dynamics, 2022_

## Referee Comment (RC2)

**Overview**

This paper presents a new morphodynamic model of Arctic delta dynamics, based on DeltaRCM-Arctic, which has been described and analyzed in several previously published studies. The new model, ArcDelRCM.jl, adopts the basic water and sediment routing schemes from DeltaRCM-Arctic, adds several new behaviors (e.g., time dependent thaw depth, a time series of river discharge), and modifies the treatment of permafrost and ice cover from DeltaRCM-Arctic. The paper shows results comparing ArcDelRCM.jl to DeltaRCM-Arctic, using a reconstruction of DeltaRCM-Arctic based on the published studies, as the code is unfortunately not publicly available. They also present results from simulations conducted with ArcDelRCM.jl using parameters meant to mimic the setting of the Lena Delta. The authors find that results from the various simulations presented suggest that bedfast ice, and the protection of nearshore deposits by bedfast ice, is largely responsible for the creation of an extensive sub-ice platform known as the "2 m ramp." While I enjoyed reading this paper and appreciate the extensive effort by the authors to model these complex processes in Arctic deltas, I have several concerns about both the technical aspects of the modeling and the presentation of the results. I am confident this paper can be published and will be a significant contribution to the community following a major revision. I provide several suggestions below that I believe will improve the manuscript.

**Specific comments**

There is a lot of technical focus on the model but comparatively little in terms of scientific questions, hypotheses, results, and discussion. I do not suggest removing any of the methods, as I believe these are necessary to understand how the model works and specifically how it differs from DeltaRCM-Arctic. I do suggest, however, that the authors frame the paper with a science question or a series of science questions and conduct more quantitative analyses that allow them to understand the impact of their changes to the model and the range of results they obtain with the new model.

Similarly, it struck me that nearly all of the results in this paper are qualitative. It seems like a missed opportunity to present a more complete picture of the features and behaviors that can be observed from these new simulations. I suggest the authors try to quantify some of their results, such as the extent of the subaqueous deposits, or the distribution of elevations, etc.

The introduction lacks appropriate referencing for cellular automata and reduced complexity models in the geomorphology community, as previously pointed out by a community comment. Examples include Murray and Paola (1994, 2003) and Murray (2007).

The background on the 2 m ramp is quite short and vague, given the heavy focus on it in the results. Please provide a more thorough description about what we do and do not know about the formation of the 2 m ramp. L47 seems to suggest that the ramps have permafrost, but there is no reference for this and I'm not sure that this is universally true. The 'sub-ice platform' on the Yukon described by Dupre (1980), for example, is reported to not have permafrost.

The methods section is noticeably missing any description of the boundary conditions used and what types of experiments were performed. The authors have stated there will be some comparisons to DeltaRCM and DeltaRCM-Arctic, and also attempts to specifically model the Lena delta. Much of this information is actually in the results section, but I suggest it be moved to the methods section. A table listing all experiments performed and the appropriate parameters or processes that were changed would be helpful.

It would be useful to know which model source code you started with, specifically, as you've listed two (Liang and Perignon). Neither of these is the most recent, though: pyDeltaRCM

L105: this requires more discussion. Was the model tuned to reproduce some specific features using this parameter? Did you try tuning other parameters? Did you perform some sort of error analysis or are you just looking for features by eye? Please provide a physical justification for picking a new value for this parameter that differs from that in established literature.

Section 2.2 begins by stating that the authors refactored the DeltaRCM algorithm 'as we saw fit.' This is a little concerning, as it is hard to know what they mean by this. Did you test a non-refactored version against a refactored version to ensure you're still getting the same results? I also suggest comparing to the most recent version of the model, pyDeltaRCM, as suggested in the community comment.

It seems to me that if the paper is about simulating arctic deltas and specifically about modifications to existing models, then all changes made to the model should be included in the main text, not an appendix.

Section 2.2.2: maybe I'm missing something, but these units don't work out for your time dependent thaw depth. How do you get rid of the Kelvin? Also, is thaw depth not dependent on there being standing water? Just dependent on positive degree days? We know that there are taliks under water bodies, which is what the rules in DeltaRCM-Arctic try to simulate by enforcing some minimum thickness for permafrost cells. Can cells under channels be permafrost in your model?

Since the authors changed both the erosive rules for permafrost and the rules for how permafrost 'forms' in the model, it is not clear how both of these changes in combination affect the results. In order to assess the new permafrost rules, please also provide figures and analysis in the paper of permafrost extent and evolution in the model as compared to DeltaRCM-Arctic. Is there permafrost under the channels? On channel banks? On the ramp?

How sensitive is the model to the choice of I=10? This should be justified further.

I'm particularly concerned about the treatment of permafrost erosion in the new model. Based on my understanding of the text, the model assumes, or rather asserts, that permafrost can never erode, which is not accurate. I assume this because they state that cells can only erode down to the thaw depth. Permafrost riverbanks absolutely do erode. Does this mean that

lateral migration cannot occur if the cells immediately adjacent to a channel are permafrost? If erosion is limited only to the thawed layers, and thawing happens based only on a degree day index, then won't riverbanks always be frozen at depth and therefore completely non-erodible. This is not physically realistic. Please provide justification for this choice and/or a further explanation of the functionality of this choice in the model. L290 also states the authors compare to a DeltaRCM-Arctic run with $E_p = 0.65$, suggesting that this mimics their restrictive erosion. But this is not true, because the ArcDelRCM implementation does not seem to allow *any* permafrost bank erosion.

L233: Please be more specific about changes to delta t ('increased by a factor of a few.'). How did you discover this? Was it unstable for all other values? Is there a range of ok values?

L260: This addresses temporal changes to the input water discharge, but what about sediments? Are they just scaled with Qw? Is that realistic based on Arctic river sediment flux timeseries? Please provide more information on how sediment input is treated and justify this choice.

I think the authors have missed an opportunity in the results section to just directly compare to the publicly available model output from previous DeltaRCM-Arctic runs, noted in the acknowledgments of Piliouras et al., (2021): "Raw model outputs are available in Piliouras et al. (2020) through DOE's ESS-DIVE repository: 10.15485/1682304. "

Information about the use of graph theory/network-based techniques should be included in methods. Extracting channel networks from model output is not a simple task, and it can be rather subjective. Please include details about how channels were extracted, what methods were applied to those channel networks, and how. Also, why not use graph theory methods designed specifically for river and delta channel networks, such as those discussed in Tejedor et al., (2015)? Also, why was 105 chosen as the number of realizations to include? The authors suggest that based on their analysis of some graph theory metrics, they did not find statistically significant differences between the various runs, including between Arctic and non-Arctic runs. How do you reconcile this with the fact that Lauzon et al., (2019) and Piliouras et al., (2021) did show noticeable differences in some aspects of the channels? Those studies have only 3 replicates and did not do a formal statistical analysis, but do you think those findings were not representative? Or is it a matter of the metrics tested?

Regarding the comparisons to the Lena delta, why was this delta chosen over others? Why not other arctic deltas? Is there something unique to the boundary conditions of this system that might result in unique deltas or features compared to other Arctic deltas? Some discussion on this is warranted.

L340: Given the differences in inputs, I'm not sure this comparison is appropriate. The deltas should be compared when they have the same total volume input.

Figure 8: can you show side by side with a control for those variables that changed?

L366: why would the atmospheric melting period be shortened in the future? Please provide a justification for this choice.

The discussion should put the results re: ramp formation back in the context of what we already know from literature. For example, L376 should include some references. L378-380 has also been suggested by previous papers, including older observational studies and Piliouras et al., (2021).

L396-399: Can you quantitatively compare the amount of in-channel erosion between the two cases for locations where h_ice = 99.99% of flow depth? it would be helpful to understand how important this rule is, and how much its effect varies spatially.

L404-407: If the ice is bedfast, how is deposition occurring beneath it? Wouldn't it have to occur in front of it (i.e., upstream of it)? It looks like you have sub-ice channels in the ramp, though you have not discussed this. These are presumably responsible for the construction of the ramp under the ice, no?

L429-430: There are many satellite images of the Lena delta where ice on part of the ramp is already melted while ice in many channels remains (e.g., Figure 7 in Overeem et al., 2022). This is in contrast to your rule that delays melting on the ramp, which you claim is a major reason for its formation. Is it realistic to force the ice to stay on the ramp for a period of time? Doesn't this somewhat contrive an intended result? Shouldn't the ice be melting uniformly everywhere by incoming radiation? What is the justification for delaying it?

L484: Where is this comparison in the text? I do not remember seeing these metrics

**Technical comments/Rephrasing suggestions**
L22: 'key interfaces between permafrost landscapes and the Arctic Ocean.' Are Arctic deltas themselves not permafrost landscapes? I would rephrase this.

L55: This sounds contrived, like you are forcing a result. Presumably you are making modifications to the rules to include more physics or more processes, with the hope that you will reproduce a 2m ramp. Please rephrase. The following paragraph should similarly be rephrased. I'd hope that one purpose of the article is to explore the processes that shape Arctic deltas and to better understand those that might contribute to development of the 2 m ramp. As written, it sounds like the purpose is purely to develop a model and present model output.

L107-109: this should be rephrased to focus on what the model steps are, not what sections are in the cited paper.

Section 2.2.4: I suggest renaming this subsection, as this section does not actually describe shore or bank migration. The phrase shore/bank migration implies some redistribution of sediments, whereas you are describing a modeling step that simply smooths the water surface.

L277: What does this mean? That the atmospheric melting can change over time and that it can be nonlinear in its rate change?

Please include a legend on Figure 6.

Figure 9 (and other figures with this color scale): please adjust the color scale. The ramp feature is not particularly visible. Maybe you can change the colorbar to a log scale so we don't see so many numbers that are all black?

Figure C1: Please label the run names on the plots with rotated text instead of the numbers for the various runs. There is no table in the text, and these numbers/IDs are not used elsewhere, so readers cannot readily identify which is which.

**References**

Dupre, W. (1980). *Yukon delta coastal processes study*. University of Houston.

Murray, A.B. and C. Paola. (1994). A cellular model of braided rivers. *Nature*. doi: 10.1038/371054a0.

Murray, A.B. and Paola, C. (2003), Modelling the effect of vegetation on channel pattern in bedload rivers. Earth Surf. Process. Landforms, 28: 131-143. doi: 10.1002/esp.428

Murray, A.B. (2007). Reducing model complexity for explanation and prediction. *Geomorphology.* doi: 10.1016/j.geomorph.2006.10.020.

Lauzon, R., A. Piliouras, and J. C. Rowland. (2019). Ice and permafrost effects on delta morphology and channel dynamics. *Geophysical Research Letters.* doi: 10.1029/2019GL082792.

Overeem, I., J. Nienhuis, and A. Piliouras. (2022). Ice-dominated Arctic deltas. *Nature Reviews: Earth and Environment.* doi: 10.1038/s43017-022-00268-x

Piliouras, A., R. Lauzon, and J. C. Rowland. (2021). Unraveling the combined influence of ice and permafrost on delta morphodynamics. *Journal of Geophysical Research – Earth Surface.* doi: 10.1029/2020JF005706

Tejedor, A., Longjas, A., Zaliapin, I., and Foufoula-Georgiou, E. (2015), Delta channel networks: 2. Metrics of topologic and dynamic complexity for delta comparison, physical inference, and vulnerability assessment, *Water Resour. Res.*, 51, 4019– 4045, doi: 10.1002/2014WR016604.

---

## Community Comment (CC1)

Discussion Commentary For: "Arctic Delta Reduced Complexity Model and its Reproduction of Key Geomorphological Structures" https://esurf.copernicus.org/preprints/esurf-2022-25/

This paper was very thought-provoking and was a pleasure to read. The advances to the DeltaRCM modeling approach (writing the code in Julia, bed-fast ice protection/shielding, time-step stability criteria, etc.) are well-presented. However we do have suggestions and questions about some elements in this manuscript. We itemize and provide numbered comments below for convenience.

1. The introduction to reduced complexity delta models (L. 29-39) lacks references to the origins of this modeling approach for landscape models (e.g., Murray & Paola 1994) and the on-going debate between explanatory and predictive models in geomorphology (Bokulich 2013).

2. As this paper introduces a new implementation of the DeltaRCM modeling framework, we would like to alert the authors to the latest Python version of the model, pyDeltaRCM (Moodie et al., 2021). pyDeltaRCM has computational runtime improvements over the previous DeltaRCM frameworks (Matlab and Python). If possible, we would suggest comparing the new ArcDelRCM.jl code (in original DeltaRCM mode) to pyDeltaRCM in addition to the runtime comparisons presented in the paper (section 2.2).

3. In Section 3.1 comparison experiments between DeltaRCM-Arctic and ArcDelRCM.jl are described and then shown. Given the lack of access to DeltaRCM-Arctic source code, it is unclear how these comparison experiments were conducted. Was the Julia implementation used to mimic DeltaRCM-Arctic? Some clarification here would be appreciated.

4. The DeltaRCM modeling approach does not simulate a delta foreset. The discussion paragraph which touches on this (L. 439-447) could use some further commentary on how this deficiency might impact the results for the modeled ramps. As the ramps extend from the delta shoreline into the ocean, it seems like the model's inability to accurately model sediment behavior in this region could impact the behavior of the ice ramps, and thus the implications of the results.

5. We would like to alert the authors to the work of Moodie & Passalacqua (2021) in which the same modeling approach is applied to simulate deltas with spatial scales comparable to the Selenga and Mississippi Deltas, this relates to the assertions made on L. 322-323.

6. It was slightly unclear how the sediment is being scaled when changing the input discharges to a time series (Section 2.2.6). We assume a sediment concentration is assumed and therefore the sediment discharge is scaling linearly with the water discharge (per description of Lena Model on L. 324). If this is the case, the comparison in Figure 7 does not seem to be appropriate. As we understand it, the volume of sediment input to the basin between the 10-day and 4 month simulations is not equal,

with a significantly larger sediment volume, both in absolute (m^3 / model year) and relative (m^/3 / model year / m^2 of model domain) terms. The results shown in Figure 7, seem to be more indicative of the total volume of sediment input into the domain rather than the differences due to ice-dynamics. We suggest scaling this comparison such that the total volume of input sediment is the same.

7.  It would be helpful to provide an example plot of what the channel graphs look like (L. 316-320). It is clear from Figure C1 that there are no significant differences between the scenarios, but it would be nice to see planform views of the graphs themselves. From the images of the topography shown in Figure 5, there appear to be differences between the scenarios, although the channel structures and number of active channels seem similar between the cases.

8.  The graph theoretic approach for channel network characterization that was referenced is designed for the analysis of polygonal trough networks, and it would be helpful to the reader to expound on how it was adopted to the distributary channel networks of deltas. In particular a clear definition for what an abandoned versus what an active channel is should be given for the graph analysis. In addition, we would like to alert the authors to the abundant literature on graph theoretic approaches to delta channel network characterization, in particular Tejedor et al. (2017) and references therein and Nesvold (2019).

9.  We appreciated the commentary on how the Lena Delta differs from the analog model simulation (L. 434-438). Looking at the imagery of the Lena Delta as compared to the model simulations, the real channel network appears to be much more complex (greater number of channels, more junctions and tortuous paths) than the model network. How might this influence the results and findings? Is it reasonable to extrapolate results from the model to the real system given these differences (or what are the limits of the extrapolation)?

None of the above points are intended to minimize the contributions made in this work, which we find to be significant. We thank the authors and the Earth Surface Dynamics community for allowing us to comment on this interesting study.

Jay Hariharan and Lawrence Vulis

References

Murray, A., Paola, C. A cellular model of braided rivers. *Nature* 371, 54–57 (1994). https://doi.org/10.1038/371054a0

Bokulich, A. (2013). Explanatory Models Versus Predictive Models: Reduced Complexity Modeling in Geomorphology. In: Karakostas, V., Dieks, D. (eds) EPSA11 Perspectives and Foundational Problems in Philosophy of Science. T*he European Philosophy of Science Association Proceedings*, vol 2. Springer, Cham. https://doi.org/10.1007/978-3-319-01306-0_10

Moodie, A. J., & Passalacqua, P. (2021). When does faulting-induced subsidence drive distributary network reorganization? *Geophysical Research Letters,* 48, e2021GL095053. https://doi.org/10.1029/2021GL095053

Moodie, A., Hariharan, J., Barefoot, E., & Passalacqua, P. (2021). pyDeltaRCM: a flexible numerical delta model. *Journal of Open Source Software*, *6*(64), 3398. https://doi.org/10.21105/joss.03398

Nesvold, E. (2019). Building informative priors for the subsurface with generative adversarial networks and graphs (Order No. 28113269). *Available from ProQuest Dissertations & Theses Global*. (2467640803). Retrieved from https://www.proquest.com/docview/2467640803?pq-origsite=gscholar&fromopenview=true

Tejedor, A., Longjas, A., Edmonds, D. A., Zaliapin, I., Georgiou, T. T., Rinaldo, A., & Foufoula-Georgiou, E. (2017). Entropy and optimality in river deltas. *Proceedings of the National Academy of Sciences*, *114*(44), 11651–11656. https://doi.org/10.1073/pnas.1708404114

---

## Author Comment (AC1)

**Overview of Major Changes**

We followed multiple comments by the referees and the community to reframe our manuscript to be more focused on the 2-m ramp feature. Accordingly, the backgrounds of the ramp feature has been expanded with additional references, and similarly in the Abstract. We have also improved our explanations of how the various modifications and simulation runs tie back to the theme of the ramps. Most importantly, the Discussions and Conclusions sections now have a stronger focus around the ramps than in our original manuscript. Technical details not relevant to the ramps, such as the run-time comparison between different source codes, and Appendix D, have been removed.

During the revision process, we have also discovered an error that concerns the graph-metrics violin plots arising from an unexpected behaviour of Julia's StatsPlots.jl package. Here is a description of this unexpected behaviour:

In the StatsPlots.jl package, to plot a DataFrame (i.e., a tabulated data type/structure comparable to Python's DataFrame of the `pandas` module) as violin/box plots, two inputs are needed per plot: the classes (in our case, they are the simulation names) and the metrics to be plotted (e.g. number of nodes). The latter is passed into the plotting function as a "column" of the DataFrame. The former, we originally thought, is an array/vector of the classes with the length of this array equal to the number of classes. The function gave no warnings or errors and plotted the violin/box plots in the original manuscript. Since the actual values of the metrics plotted were not drastically different, we did not notice any issues immediately. However, during the revision phase, we added another metric (the number of pixels with ~2 m elevations) in our analyses, and the values there do show enough differences that we became suspicious of the plotting function. We found that we were supposed to feed in the "classes" column "as is" (i.e., with the hundreds of repeat entries for each realisation, rather than a short vector with unique entries) for the plots to

display correctly. We felt thankful to have this opportunity during revision to catch this subtle, uncaught issue with the plotting package.

After learning of this unexpected behaviour and taking the appropriate measures, we found that the graph metrics now show some differences between various cases and models examined in our manuscript. We have moved the graphs section from Appendix C in the original manuscript back to the Results section (now Section 3.3) along with updated discussions in Section 4. Our main conclusion of ArcDelRCM.jl reproducing the ramp feature remains unchanged.

During revision, a missing temperature term in the calculation of the positive-degree-day index was discovered by Referee 2. We have changed the description of the positive-degree-day index, *I*, as follows:

> '[...] and *I* is the "positive degree day index", which is the integrated number of days times the positive temperature since winter. For *I* in our simulations, we use a mean temperature of 4ºC to get *I* (see Appendix B for the reason for this choice and the sensitivity of the model output to this value).'

And in Appendix B, we added the following final sentences:

> "The temperature we used to calculate *I* is the average daily temperature (Figure B1) from 1st June to 30th September, which is 4ºC. In addition to the graph metrics of the results presented in Section 3.2, Figure 6 shows the same metrics from the cases identical to the standard ArcDelRCM.jl runs except with *I* starting count from 5 and 15 days and (independently) the average positive temperature being 8 and 12ºC. They show that, within the ranges tested, these parameters do not affect the resulting deltas beyond the internal variability of each case. To help illustrate this visually, an example of each case is shown in Figure B2."

Accordingly, we have re-run all of our simulations. However, we did not observe changes that would necessitate changing our main conclusions. We apologise for this mistake, and

thank Referee 2 for discovering this and alerting us.

We have also made another adjustment to the DeltaRCM-Arctic configuration of ArcDelRCM.jl. After carefully re-writing DeltaRCM (non-Arctic) into Julia, the first author (Chan) followed Lauzon et al. (2019)'s description to extend it to mimic the DeltaRCM-Arctic version due to lack of source-code access. He made detailed notes in the comments of the code where there are potential ambiguities and the interpretations adopted. These comments form the basis of the portions of the model descriptions in Section 2.2 of our manuscript, where we state our assumptions about what we understood from the DeltaRCM-Arctic papers (and thus adopted). During the revision process, Chan re-examined these parts of the code again and re-compared against the now-published Piliouras et al. (2021), since they also re-described the model therein. He found that the descriptions regarding permafrost classification (last paragraph of Section 2.1 of Piliouras et al., 2021) has more clarity compared to Text S2, supplementary material, Lauzon et al., 2019, which reads:

> "For an entire cell to be treated as permafrost, its total permafrost thickness must be greater than 75% of either the 5 m inlet channel depth (3.75 m) or the local deposit thickness, whichever is greater."

and realised that there is a discrepancy that needs to be closed in the source code. This has been done. The use of the erodibility factor, $E$, as a multiplicative factor to scale the flow-speed thresholds of erosion is also reinterpreted to be simpler. Chan used to assume that, in order to avoid division by 0 by erroneous usage, an erodibility of 65% would likely be implemented as increasing the flow-speed threshold by 35%. However, during revision and re-examination of the code, Chan concluded that it makes more sense that the original authors would divide the flow-speed thresholds by 0.65 instead, consistent with the usage of $E$ in the bed-diffusion step. As a result, the $E = 0.65$ used in the original manuscript prior to revision is actually $E = 0.74$ under the updated interpretation (and likely the one used by the original authors of DeltaRCM-Arctic). Chan apologise for this and ask

for all of your kind understanding. All of the affected simulation cases have been re-run and reflected in our updated manuscript. Our main conclusions did not change.

Last but not least, to improve clarity, we have changed our use of terminology to replace "bed-fast" with "bottom-fast".

**Response to Referee #1's Comments**

We quote the referee's original comments in **bold-italics** typeface, and give our responses in light typeface below each section.

***Based on an existing and published reduced complexity model (RCM) that simulates the morphodynamics of Arctic deltas (Rebecca Lauzon et al., 2019; cited in the manuscript) Ngai-Ham Chan and co-authors have developed an extended DeltaRCM that is able to reproduce an important morphological feature: the wide 2-m ramp characteristic for arctic deltas (Erk Reimnitz, 2002; cited in the manuscript). According to the authors, the models published so far cannot reproduce this important characteristic. The changes from the DeltaRCM are that the entire model has been rewritten in the Julia programming language and that significant changes are made to the model in order to improve its ability to account for processes that are climate-sensitive.***

***The submarine ramp, which can reach an extent of up to 30 km off the Lena Delta (Siberian Arctic, Laptev Sea), has a significant influence, for example, on the wave energy impinging on the coast and thus on coastal erosion and sediment transport. Chan et al. found that "the delayed breakup of bed-fast ice on and around the deltas is ultimately responsible for the development of the ramp feature". Finally, they tested a strong climate-warming scenario on the simulated deltas. They have found that ramp features degrade on a time scale of centuries and disappear in less than a millennium. In this context, the RCM presented here is an important step toward realistic prediction of Arctic coastal evolution in the face of global warming.***

***I congratulate the authors on this interesting and important study and believe that the publication of the results contributes significantly to the understanding of the dynamics of the Arctic coasts. I have no doubts about the methodology presented in the manuscript. The results are also comprehensible to me.***

***However, in my opinion, the manuscript focuses too much on the description of the model and too little on the scientific question/hypothesis that is to be answered on the basis of the further developed model. I believe, however, that this weakness can be overcome by a structural revision of the manuscript and that the publication will thereby attract the interest of a wider readership. I therefore recommend the publication of the manuscript after a moderate revision.***

We thank the referee for the nice summary, the positive assessment, and the constructive comments regarding our manuscript. We shall address the referee's points one by one below.

***Below is a list of my comments and questions (the numbers refer to the line numbers of the original manuscript):***

***Abstract:***

***1 – The abstract should be revised to focus more on the underlying question. The abstract in Lauzon et al. (2019) is certainly a good example.***

We changed the abstract to focus more on the underlying question:

'Arctic river deltas define the interface between the terrestrial Arctic and the Arctic Ocean. They are the site of sediment, nutrient, and soil organic carbon discharge to the Arctic Ocean. Arctic deltas are unique globally because they are underlain by permafrost, acted on by river and sea ice, and many are surrounded by a broad shallow ramp. Such ramps may buffer the delta from waves, but as the climate warms and permafrost thaws, the evolution of Arctic deltas will likely take a different course, with implications both local in scale and on the wider Arctic Ocean. One important way to understand and predict the evolution of Arctic deltas is through numerical models. Here we present ArcDelRCM.jl, an improved reduced complexity model (RCM) of arctic delta evolution based on the DeltaRCM-Arctic model (Lauzon et al., 2019), which we have reconstructed using published information. Unlike previous models, ArcDelRCM.jl is able to replicate the ramp

around the delta. We have found that the delayed breakup of the so-called "bottom-fast ice" (i.e., ice that is in direct contact with the bed of the channel or the sea, also known as "bed-fast ice") on and around the deltas is ultimately responsible for the appearance of the ramp feature in our models. However, changes made to the modelling of permafrost erosion and protective effects of bottom-fast ice are also important contributors. Graph analyses of the delta network performed on ensemble runs show that deltas produced by ArcDelRCM.jl have more interconnected channels and contain less abandoned subnetworks. This may suggest a more even feeding of sediments to all sections of the delta shoreline, supporting ramp growth. Moreover, we showed that the morphodynamic processes during the summer months remain active enough to contribute significant sediment input to the growth and evolution of Arctic deltas, thus should not be neglected in simulations gauging the multi-year evolution of delta features. Finally, we tested a strong climate-warming scenario on the simulated deltas of ArcDelRCM.jl, with temperature, discharge, and ice conditions consistent with RCP 7-8.5. We found that the ramp features degrade on the time scale of centuries and effectively disappear in under a millennium. Ocean processes, which are not included in these models, may further shorten the time scale. With the degradation of the ramps, any dissipative effects on wave energy they offered would also decrease. This could expose the sub-aerial parts of the deltas to increased coastal erosion, thus impacting permafrost degradation, nutrients and carbon releases.'

**2 – What exactly does "Arctic deltas... provide key stratigraphic records of permafrost landscape evolution" mean? Why is this important in the context of this study?**

We have replaced the opening three sentences of the abstract with the following:

> "Arctic river deltas define the interface between the terrestrial Arctic and the Arctic Ocean. They are the site of sediment, nutrient, and soil organic carbon discharge to the Arctic Ocean. Arctic deltas are unique globally because they are underlain by permafrost, acted on by river and sea ice, and many are surrounded by a broad

shallow ramp. Such ramps may buffer the delta from waves, but as the climate warms and permafrost thaws, the evolution of Arctic deltas will likely take a different course, with implications both local in scale and on the wider Arctic Ocean."

**6 – "We have rewritten…" Certainly, this was necessary work. But is it so crucial that it has to be mentioned in the abstract?**

We have removed the sentence from the abstract.

**10 – I think that only a few readers know what "bed-fast ice" is. A brief explanation would certainly be helpful, wouldn't it?**

We agree. We have changed our use of terminology to replace "bed-fast" with "bottom-fast". We have also rephrased the sentence as follows:

> 'We have found that the delayed breakup of the so-called "bottom-fast ice" (i.e., ice that is in direct contact with the bed of the channel or the sea, also known as "bed-fast ice") on and around the deltas is ultimately responsible for the appearance of the ramp feature in our models.'

**11/12 Changes made to the modelling is responsible for the development of the ramp feature? You mean for the appearance of the ramp in the model, right?**

Yes, we have modified the wording accordingly when rewriting the sentence above.

**12/14 What does "differences in channel structure" mean?**

We meant differences in the graph metrics computed after converting the channel network into graphs. However, due to an unexpected behaviour of the plotting package we used (detailed in the "Overview of Major Changes" above), this sentence has completely changed.

**14 – "summer month contribute significantly…". Shouldn't it read " the morphodynamic (?) processes occurring during the summer months contribute**

*significantly to the...". Due to these small linguistic inaccuracies, the abstract loses some clarity.*

We have modified the sentence to read:

> "Moreover, we showed that the morphodynamic processes during the summer months remain active enough to contribute significant sediment input to the growth and evolution of Arctic deltas, thus should not be neglected in simulations gauging the multi-year evolution of delta features."

*16 - It would be good if you briefly describe which changed environmental conditions you have assumed in the "strong climate-warming scenario".*

We have modified the relevant sentence in the abstract to become:

> "Finally, we tested a strong climate-warming scenario on the simulated deltas of ArcDelRCM.jl, with temperature, discharge, and ice conditions consistent with RCP 7-8.5."

*Introduction:*

*22 – What do you mean by "key interface"? Please explain.*

We have replaced "key interface" with "key components". The opening sentences now reads:

> "Arctic deltas are key components of the permafrost landscape, connecting the permafrost areas upstream and the Arctic Ocean. They act as records and filters of the particulate and dissolved matter, such as sediments and nutrients, that originated from the Arctic and sub-Arctic regions [...]"

*23 – Perhaps instead of "sediments and nutrients" it would be better to write "particulate and dissolved matter, such as...".*

This has been done.

**26 – "Arctic deltas will likely be affected…" Please specify.**

We have rephrased the sentence to reflect the purpose of communicating an uncertain future (hence the importance of studying Arctic deltas from various aspects in general):

> "As permafrost thaws and the Arctic ocean trends towards being free of ice, especially under amplified warming in polar regions (Stocker, 2014), Arctic deltas face an uncertain future."

**41 bis 52 – I think this paragraph should be at the beginning of the Introduction because it explains the need for "improved" modeling and outlines the scientific question.**

We appreciate the referee's view that the main question of the paper should be at the top of the Introduction. However, we have taken the narrative to "zoom in" in a stepwise manner from the general "why is the Arctic important?" to "why ramps?". We are unsure how to find a middle ground to reconcile these two approaches whilst preserving the flow.

**Figure 1 – Perhaps a small inlay, with an overview map informing about the location of the map section within the Arctic, would be helpful.**

This has been done.

**55/56 and 59/60 – The importance of summer month: In the study, model runs covering 10 days during the high-discharge period are compared with simulation results over the entire 4 summer months (Figure 7). Is it really necessary to emphasize that the summer discharge of one of the ten largest rivers on earth also has a significant influence on the development of the delta?**

We agree. However, we showed this in order to demonstrate the importance to not use a 10-day model year in simulating some deltas (e.g. large, Lena-scale deltas), as was adopted in the more analytical (smaller-scale, more general) deltas by Lauzon et al. (2019) and Piliouras et al. (2021) that used DeltaRCM-Arctic.

***Methods:***

***2.1 Description of DeltaRCM(-Arctic) – The 5-page chapter includes a description of the previous RCM model and is thus a mere repetition of the already published model description (see citations in the manuscript). I think this chapter should be shortened considerably and rather describe the development of the DeltaRCM so far. In particular, it should be described why a further development of the DeltaRCM-Arctic seemed to be necessary.***

We have the same feeling as the referee, and have attempted multiple times (both before the initial submission and in the current revision process) to cut out the parts that repeat information in already-published articles. However, since descriptions of our modifications to make ArcDelRCM.jl heavily depend on some parameters, expressions, and automata rules of the DeltaRCM models, we found that most of the time we ended up "transplanting" the descriptions in Section 2.1 into Section 2.2. Moreover, we have also found (or, rather, re-discovered during the revision process) that Section 2.1 contains a number of observations and information "hidden" in the non-Arctic DeltaRCM source codes that we learned during the coding process. After weighing and trying multiple options, in order to avoid confusion between what is original/inherited and what is new, and to prevent the narrative from becoming too fragmented, we have cut away a large figure (the original Figure 2) and some parts that do not define parameters that we need later. Admittedly, the shortening effect is not large, but this is the best balancing act we could find.

***2.2 The authors state that: "Since we do not have access to the source code of DeltaRCM-Arctic, we have no performance comparisons between the Arctic simulations." On the other hand, the results of the two models are compared in chapter 3. Can you explain how this goes together.***

The comparison referred to the non-Arctic version of DeltaRCM, the source codes of which are available in multiple versions (in MATLAB and Python). We ran ArcDelRCM.jl in the non-Arctic DeltaRCM setting for these comparisons. However, since there is now a

newer pyDeltaRCM version available and the run-time comparisons appear to cause considerable concerns and distractions, we have decided to remove such run-time comparisons altogether.

***Results:***

***The whole chapter is well written and the results clearly presented. However, it is not clear to me on what basis (data/publication?) values for sand fraction (25%) and ice cover (40%) were chosen.***

We have added the following clarification to the reason behind these choices:

> "With the exception of $h_{\text{ice,max}}$, these values are chosen after the demonstrated cases of DeltaRCM-Arctic in Piliouras et al. (2021) and Lauzon et al. (2019)."

***357 – This should already be mentioned in the abstract.***

We have now specified RCP 7-8.5 in the abstract.

***440 – Tidal currents in this area of the Laptev Sea are weak. Perhaps it is better to speak of waves and ocean currents in general. At the beginning of winter, when the newly forming ice is still mobile, the formation of sediment-laden sea ice in shallow water (e.g., anchor ice) and the subsequent export of the "dirty" sea ice plays an important role in the sediment budget of the delta. This should at least be mentioned (various papers by Reimintz and Are).***

We have expanded the relevant paragraph to read:

> 'We note, however, that important ocean-driven processes are missing in the model, resulting in differences in smoothness and outer-edge shapes between the modelled ramps and those observed in reality (Figure 1). Surges (e.g. during winter storm) can thicken from underneath the ice cover that become bottom-fast over the ramps (Reimnitz, 2002), which could enhance the protection by bottom-fast ice during spring. Moreover, compared to the deltas produced by DeltaRCM-Arctic and ArcDelRCM.jl, the observed slopes of the sediment bed beyond the outer-edge of

the 2-m ramps are much gentler, typically dipping from 2 m to > 20 m over $O(10)$ km rather than $O(0.1)$ km (Reimnitz, 2002; Are et al., 2002). The lack of the gentle dipping may have resulted from the limitation of the models having abrupt thresholds for deposition (Sect. 2.1.4) and in the classification of "on-delta" and "ocean" grid cells during the flow routing (Sect. 2.1.2), in which most of the sediments carried in a packet tend to get deposited as soon as it leaves the "on-delta" cells. The gentle slope observed in reality could be a result of marine processes such as sediment re-suspension by waves, which can be effective to water depths exceeding 10 m (Heim et al., 2014), and transport by currents. Wave-action re-suspends sediment, especially during fall storms. Sediment in the water column acts as nuclei for frazil ice formation and can be integrated into the forming ice pack (Reimnitz et al., 1992). The formation and export of ice during the fall can remove material from the ramp, as can anchoring of ice in winter and spring to the seabed (Krumpen et al., 2020). Once anchored, continued cold temperatures create a seasonally frozen layer in the sediment beneath the bottom-fast ice (Osterkamp et al.,1989). This material can be exported once the ice is lifted in spring (Reimnitz et al., 1987). Both processes, entrainment of frazil ice and adherence of frozen sediment, lead to ice-rafting (Are and Reimnitz, 2008). Sediment can also be moved by ice-gouging and ice-bulldozing, when thicker ice masses are ploughed into the seabed (Maznev et al., 2019; Ogorodov et al., 2018). In the context of the simulations shown above, these marine and marine-ice processes could affect not just the exact morphology, but also the time scale of ramp formation (balancing between the sediment supply to build, and the reworking to sculpt the overall ramp), although not prevent it. The same could, of course, also affect the time scale of degradation of the ramp under a changing climate. Future work on the model could focus on improving the capability of off-shore dynamics, such that a full picture of a delta's formation and destruction can be built.' (starting from line 585)

***Conclusion:***

***The whole chapter reads like a redundant summary of the discussion. Perhaps the parts from the discussion that describe the possible future development of the model should appear here.***

***465 – Was this a random result or was the existing DeltaRCM-Artcic further developed to reproduce the 2m ramp? Wasn't the scientific goal then to describe which modification of the model is "responsible" for the formation of the ramp? The Conclusions is, in my opinion, too much written from the technical point of view of the model improvement. Wouldn't it be better to start the Conclusions with the scientific question (what leads to the formation of the 2m ramp?), then outline the need why the existing model had to be extended (i.e. DeltaRCM-Arctic does not reproduce a ramp) and then explain what was modified in the model and how you proceeded methodically?***

We have rewritten the conclusion section to follow the suggestions made by the referee, whilst also keeping the "summary" nature of Conclusions sections. Technical details have been significantly slimmed down in particular. The new conclusion is as follows:

[revised manuscript text omitted]

We thank the referee again for the positive assessment and the effort spent on improving our manuscript.

Response to Community Comment 1 by

Jayaram Hariharan and Lawrence Vulis

We quote the original comments in **bold-italics** typeface, and give our responses in light typeface below each section.

***This paper was very thought-provoking and was a pleasure to read. The advances to the DeltaRCM modeling approach (writing the code in Julia, bed-fast ice protection/shielding, time-step stability criteria, etc.) are well-presented. However we do have suggestions and questions about some elements in this manuscript. We itemize and provide numbered comments below for convenience.***

We thank Dr. Hariharan and Dr. Vulis for spending the time and effort to contribute to the improvement of our manuscript. We shall respond point by point below.

***1. The introduction to reduced complexity delta models (L. 29-39) lacks references to the origins of this modeling approach for landscape models (e.g., Murray & Paola 1994) and the on-going debate between explanatory and predictive models in geomorphology (Bokulich 2013).***

We apologise for this omission. We have corrected this; the relevant sentence in the Introduction section now reads:

"To address these issues, the second class of models -- reduced complexity models (RCMs) -- simulate phenomenological processes of arctic delta evolution using rule based trajectories of cellular automata (originating works in this field include, e.g., Murray and Paola, 1994, 2003; Murray, 2007)." (line 54)

The philosophical explorations of Bokulich (2013) is very interesting, and we thank the referee for the suggestion to include. We have incorporated it into our text as follows:

'As Bokulich (2013) pointed out, there are on-going debates about the predictive power and explanatory insights offered by these classes of models, and their usage

by researchers are typically with "division of cognitive labour"—to use a term coined by Bokulich—in mind. Within this context, we take the RCM approach in order to explore what physical-process rules could be favourable to the formation of a ramp feature.' (line 60)

***2. As this paper introduces a new implementation of the DeltaRCM modeling framework, we would like to alert the authors to the latest Python version of the model, pyDeltaRCM (Moodie et al., 2021). pyDeltaRCM has computational runtime improvements over the previous DeltaRCM frameworks (Matlab and Python). If possible, we would suggest comparing the new ArcDelRCM.jl code (in original DeltaRCM mode) to pyDeltaRCM in addition to the runtime comparisons presented in the paper (section 2.2).***

Thank you for alerting us to the new Python version of DeltaRCM. We take note that pyDeltaRCM is re-organised professionally and contains significant usage improvements. We have removed all runtime comparisons between ArcDelRCM.jl and the older MATLAB and Python source codes, as they are not important points in the manuscript (especially after the revision to focus it more on questions surrounding the 2m ramp feature).

***3. In Section 3.1 comparison experiments between DeltaRCM-Arctic and ArcDelRCM.jl are described and then shown. Given the lack of access to DeltaRCM-Arctic source code, it is unclear how these comparison experiments were conducted. Was the Julia implementation used to mimic DeltaRCM-Arctic? Some clarification here would be appreciated.***

Indeed, after first writing the non-Arctic DeltaRCM in Julia, carefully cross-checking Liang's MATLAB and Perignon's Python versions, we had to rely on the text descriptions of Lauzon et al. (2019) with additional information from Piliouras et al. (2021) to mimic DeltaRCM-Arctic. This is another reason (in addition to succinctness and the Julia naming conventions) why we chose a slightly different name for our model instead of adding to the "DeltaRCM-Arctic" root name. Admittedly, it would have been ideal if we could work directly on top of the authentic DeltaRCM-Arctic source code. However, given our focus

on exploring what physical-process rule(s) is (are) favourable to the formation of the ramp feature, by having ArcDelRCM.jl both perform as itself and mimic DeltaRCM-Arctic at least ensures that the "with-ramp" and "without-ramp" results are not due to some small, undocumented computational detail differences between the authentic DeltaRCM-Arctic and our reproduction of it. To address this concern, we have made the following changes to our manuscript:

- We have modified the wording of the relevant sentence in the Abstract to read: "Here we present ArcDelRCM.jl, [...] based on the DeltaRCM-Arctic model (Lauzon et al., 2019), which we have reconstructed using published information."
- To the sentence "We call the modified model ArcDelRCM.jl for succinctness and in keeping with conventions of Julia code packages", we have added the subclause "and to signify that it is a reconstruction of the Arctic extension of DeltaRCM based on published articles and not a direct translation of the original DeltaRCM-Arctic source code (which is not publicly available)." (line 75)
- We have added the sentence to the opening paragraph of Section 2.2: 'Due to the non-availability of the source code of DeltaRCM-Arctic, whenever we present simulations results of "DeltaRCM-Arctic", we mean the DeltaRCM-Arctic configuration of ArcDelRCM.jl.' (line 197)

**4. The DeltaRCM modeling approach does not simulate a delta foreset. The discussion paragraph which touches on this (L. 439-447) could use some further commentary on how this deficiency might impact the results for the modeled ramps. As the ramps extend from the delta shoreline into the ocean, it seems like the model's inability to accurately model sediment behavior in this region could impact the behavior of the ice ramps, and thus the implications of the results.**

The lack of ocean-driven processes (which includes influencing how delta-sourced deposition is carried into the delta foreset) could both aid and slow the ramp-formation processes. On one hand, tidal or storm surges help both to smooth the ramps' top surface and to increase the thickness of the bottom-fast ice as water infiltrates the gap between

the ice cover and the ramp. On the other hand, the reworking of sediments in the foreset could be an impediment in the formation of the ramp, as the deposits that would fill the accommodation space below the ice is spread across a larger area. However, this in itself would not prevent the ramp from growing, as sediments would eventually build up to reach the ice bottom. This is a complex interaction that is not straightforward to summarise or speculate in a definitive tone. To expand on it in the main text, we have expanded the relevant paragraph as follows:

> 'We note, however, that important ocean-driven processes are missing in the model, resulting in differences in smoothness and outer-edge shapes between the modelled ramps and those observed in reality (Figure 1). Surges (e.g. during winter storm) can thicken from underneath the ice cover that become bottom-fast over the ramps (Reimnitz, 2002), which could enhance the protection by bottom-fast ice during spring. Moreover, compared to the deltas produced by DeltaRCM-Arctic and ArcDelRCM.jl, the observed slopes of the sediment bed beyond the outer-edge of the 2-m ramps are much gentler, typically dipping from 2 m to > 20 m over $O(10)$ km rather than $O(0.1)$ km (Reimnitz, 2002; Are et al., 2002). The lack of the gentle dipping may have resulted from the limitation of the models having abrupt thresholds for deposition (Sect. 2.1.4) and in the classification of "on-delta" and "ocean" grid cells during the flow routing (Sect. 2.1.2), in which most of the sediments carried in a packet tend to get deposited as soon as it leaves the "on-delta" cells. The gentle slope observed in reality could be a result of marine processes such as sediment re-suspension by waves, which can be effective to water depths exceeding 10 m (Heim et al., 2014), and transport by currents. Wave-action re-suspends sediment, especially during fall storms. Sediment in the water column acts as nuclei for frazil ice formation and can be integrated into the forming ice pack (Reimnitz et al., 1992). The formation and export of ice during the fall can remove material from the ramp, as can anchoring of ice in winter and spring to the seabed (Krumpen et al., 2020). Once anchored, continued cold temperatures create a seasonally frozen layer in the sediment beneath the bottom-fast ice

(Osterkamp et al.,1989). This material can be exported once the ice is lifted in spring (Reimnitz et al., 1987). Both processes, entrainment of frazil ice and adherence of frozen sediment, lead to ice-rafting (Are and Reimnitz, 2008). Sediment can also be moved by ice-gouging and ice-bulldozing, when thicker ice masses are ploughed into the seabed (Maznev et al., 2019; Ogorodov et al., 2018). In the context of the simulations shown above, these marine and marine-ice processes could affect not just the exact morphology, but also the time scale of ramp formation (balancing between the sediment supply to build, and the reworking to sculpt the overall ramp), although not prevent it. The same could, of course, also affect the time scale of degradation of the ramp under a changing climate. Future work on the model could focus on improving the capability of off-shore dynamics, such that a full picture of a delta's formation and destruction can be built.' (starting from line 585)

**5. We would like to alert the authors to the work of Moodie & Passalacqua (2021) in which the same modeling approach is applied to simulate deltas with spatial scales comparable to the Selenga and Mississippi Deltas, this relates to the assertions made on L. 322-323.**

We thank you for alerting us to this work. Accordingly, we have modified the opening sentences to Section 3.4 as follows:

> "As a test case to mimic a large delta such as the Lena Delta, we ran simulations adopting spatial scales that had never been applied to ice-dominated delta using this family of models before. (However, the non-Arctic DeltaRCM has previously been applied on a similar spatial scale on the Mississippi and Selenga deltas by Moodie and Passalacqua, 2021.)"

**6. It was slightly unclear how the sediment is being scaled when changing the input discharges to a time series (Section 2.2.6). We assume a sediment concentration is assumed and therefore the sediment discharge is scaling linearly with the water discharge (per description of Lena Model on L. 324). If this**

**is the case, the comparison in Figure 7 does not seem to be appropriate. As we understand it, the volume of sediment input to the basin between the 10-day and 4 month simulations is not equal, with a significantly larger sediment volume, both in absolute (m^3 / model year) and relative (m^/3 / model year / m^2 of model domain) terms. The results shown in Figure 7, seem to be more indicative of the total volume of sediment input into the domain rather than the differences due to ice-dynamics. We suggest scaling this comparison such that the total volume of input sediment is the same.**

Indeed, the difference shown in the now-Figure 9 (original Figure 7) is the result of the additional sediment input during summer months. That is actually our point. Since all the previous demonstrations in Lauzon et al. (2019) and Piliouras et al. (2021) used a 10-day model year, which were suitable for their purposes and for the size of the deltas they simulated, we are emphasising the point that we must use a much longer model year to include summer months in order to gauge the evolution of large deltas (including the ramp feature) over multiple years (laying the ground for our multi-year simulations to gauge the ramp feature's evolution).

We have modified the second paragraph of Section 3.4 to help clarify this:

> "We would first like to find out the differences arising from assuming a 10-day model year (suitable for smaller Arctic deltas, as adopted by Lauzon et al., 2019; Piliouras et al., 2021) and from assuming a 4-month model year including the summer months (suitable for large Arctic deltas such as Lena Delta and Mackenzie Delta) due to the resulting different total sediment input per year."

We have also modified the opening sentence of the relevant paragraph in Section 4 (Discussions) to:

> "In order to gauge the long-term (multi-year) evolution of ramp features in major Arctic deltas, we first address the duration covered by each model year. To this

end, Figure 9 demonstrates the importance of delta activities (and the associated sediment input) outside of the peak flooding season."

**7. It would be helpful to provide an example plot of what the channel graphs look like (L. 316-320). It is clear from Figure C1 that there are no significant differences between the scenarios, but it would be nice to see planform views of the graphs themselves. From the images of the topography shown in Figure 5, there appear to be differences between the scenarios, although the channel structures and number of active channels seem similar between the cases.**

We agree that an image of the graphs in comparison to the underlying delta would be very helpful in conveying the context better. We have thus included an example from a run parameterized to Figure 3.

**8. The graph theoretic approach for channel network characterization that was referenced is designed for the analysis of polygonal trough networks, and it would be helpful to the reader to expound on how it was adopted to the distributary channel networks of deltas. In particular a clear definition for what an abandoned versus what an active channel is should be given for the graph analysis. In addition, we would like to alert the authors to the abundant literature on graph theoretic approaches to delta channel network characterization, in particular Tejedor et al. (2017) and references therein and Nesvold (2019).**

Thank you for pointing this out, we agree that the adaptation of the approach to extract and analyze the graph as introduced by Rettelbach et al. 2021 was not made clear in the earlier version of this manuscript. We have added a comprehensive subsection with details on the methodology, including the clarification on how the 'polygonal trough network' approach can be adapted to work for characterizing deltas as well. We have also included a clarification on our definitions of active and abandoned channels, and set our approach into the context of further literature, including the publications by Tejedor et al. (2015) and Nesvold (2019):

'2.3 Graph Analyses on Ensemble Runs

'To assess quantitatively and statistically the effects of the various modifications described in Sect. 2.2, we apply graph theory to derive metrics on a collection of ensemble runs, each set with a different process (corresponding to Sect. 2.2.1 to 2.2.4), or combination thereof, enabled. Previous work by, for example, Smart and Moruzzi (1971), Edmonds et al. (2011), Tejedor et al. (2015), and Nesvold (2019) has shown that the topologies of deltas can be described with quantitative graph metrics, such as the "loopiness" and the structural overlapping of the subnetworks, the "recombination factor" describing the ratio between the number of junctions and the number of forks in the delta systems, or the fractal dimension characterising a delta's self-similarity. While these metrics give interesting insights to the environmental properties of the real-world deltas, we sought a holistic approach that would quantify the differences between simulation results with simplest, yet meaningful descriptors. We thus made use of the approach introduced by Rettelbach et al. (2021), which provides an end-to-end approach starting from the extraction of the graph from the delta images to providing the quantitative metrics of interest for the comparison of the ensemble runs with different parameters and forcings. While the approach by Rettelbach et al. was initially developed for characterising hydrological networks in polygonal permafrost landscapes, the methodology of automated graph extraction from underlying imagery remains exactly the same. In the original publication, the authors used binarised digital elevation models to compute the skeleton of the channel network, while we here extracted the graph from the deltas binarised based on the location of the channels (see Figure 3). [...]'

**9. We appreciated the commentary on how the Lena Delta differs from the analog model simulation (L. 434-438). Looking at the imagery of the Lena Delta as compared to the model simulations, the real channel network appears to be much more complex (greater number of channels, more junctions and tortuous**

***paths) than the model network. How might this influence the results and findings? Is it reasonable to extrapolate results from the model to the real system given these differences (or what are the limits of the extrapolation)?***

Indeed, due to the reduced complexity of all the models involved, processes such as peat growth, thermo-erosional niches, and block collapses are not explicitly modelled. Rather, the RCMs take an approach of simulating the overall effects of all the specific processes in order to gain understanding and explainability of the bigger picture. This may have impacted the detailed channel-network arrangements of the model output when compared with a real-life counterpart. However, the DeltaRCM family of models (Liang et al., 2015b; Lauzon et al., 2019; Piliouras et al., 2021) have been demonstrated to simulate the overall evolution of deltas well (e.g. planform, area growth, trends in channel mobility and quantity). Therefore, we expect the results from the ramp analyses to be applicable to explain reality in terms of the mechanisms and conditions favourable to ramp formation and growth, effects of climate on the survival of the ramp, etc. That said, we would not use this class of models to recreate very specific appearances of specific deltas, as they may be influenced by the aforementioned specific processes in which the local, fine-scale details are relevant, and also because of the random-walk nature of these RCMs.

***None of the above points are intended to minimize the contributions made in this work, which we find to be significant. We thank the authors and the Earth Surface Dynamics community for allowing us to comment on this interesting study.***

We thank Dr. Hariharan and Dr. Vulis again for their positive assessment and constructive comments that helped improve our manuscript.

**Response to Referee #2's Comments**

We quote the original comments in **bold-italics** typeface, and give our responses in light typeface below each section.

**Overview**

*This paper presents a new morphodynamic model of Arctic delta dynamics, based on DeltaRCM-Arctic, which has been described and analyzed in several previously published studies. The new model, ArcDelRCM.jl, adopts the basic water and sediment routing schemes from DeltaRCM-Arctic, adds several new behaviors (e.g., time dependent thaw depth, a time series of river discharge), and modifies the treatment of permafrost and ice cover from DeltaRCM-Arctic. The paper shows results comparing ArcDelRCM.jl to DeltaRCM-Arctic, using a reconstruction of DeltaRCM-Arctic based on the published studies, as the code is unfortunately not publicly available. They also present results from simulations conducted with ArcDelRCM.jl using parameters meant to mimic the setting of the Lena Delta. The authors find that results from the various simulations presented suggest that bedfast ice, and the protection of nearshore deposits by bedfast ice, is largely responsible for the creation of an extensive sub-ice platform known as the "2 m ramp." While I enjoyed reading this paper and appreciate the extensive effort by the authors to model these complex processes in Arctic deltas, I have several concerns about both the technical aspects of the modeling and the presentation of the results. I am confident this paper can be published and will be a significant contribution to the community following a major revision. I provide several suggestions below that I believe will improve the manuscript.*

We thank the referee for the affirmative assessment of our manuscript, and especially for the detailed effort spent on improving our manuscript. We describe our responses and the associated efforts below.

***Specific comments***

***There is a lot of technical focus on the model but comparatively little in terms of scientific questions, hypotheses, results, and discussion. I do not suggest removing any of the methods, as I believe these are necessary to understand how the model works and specifically how it differs from DeltaRCM-Arctic. I do suggest, however, that the authors frame the paper with a science question or a series of science questions and conduct more quantitative analyses that allow them to understand the impact of their changes to the model and the range of results they obtain with the new model.***

We have followed the referee's suggestion to reframe the manuscript to focus on the question: What leads to the formation of the ramp and what is the fate of these features under a warming climate? Our modelling approach of using RCM for its explainability, the modifications we explored that resulted in ArcDelRCM.jl, the simulation cases we ran, and the interpretations of the results are all now more tightly focused around the question of the ramp feature.

***Similarly, it struck me that nearly all of the results in this paper are qualitative. It seems like a missed opportunity to present a more complete picture of the features and behaviors that can be observed from these new simulations. I suggest the authors try to quantify some of their results, such as the extent of the subaqueous deposits, or the distribution of elevations, etc.***

Due to the nature of the ramp, many of the observations are necessarily qualitative. However, we take note of the referee's point. The graph analyses performed on the ensemble runs, which were included already in the original manuscript, were an attempt to gain a more quantitative and statistical picture. Now that the unexpected behaviour of the plotting package has been resolved (see the Overview of Major Changes for details; the same are also included on page 22 of this response), the graph metrics are now more informative. As an additional quantitative measure using the ensemble runs, we have counted the number of pixels that are at the expected depth of a ramp feature (i.e., -2 ±

0.5 m elevation) as a proxy to the size of the ramp (or at least the "off-shore depositions" of DeltaRCM-Arctic that are likely the building-block material for a ramp feature, as suggested in Lauzon et al., 2019 and Piliouras et al., 2021). Due to the extensiveness of analyses that could be performed, as evident in both Lauzon et al. and Piliouras et al. (especially the latter, of which the results and discussion sections alone are 10 pages long), we chose to focus on the ramps as the primary topic, and defer the more general analyses to a separate study.

***The introduction lacks appropriate referencing for cellular automata and reduced complexity models in the geomorphology community, as previously pointed out by a community comment. Examples include Murray and Paola (1994, 2003) and Murray (2007).***

The citations have now been added. We apologise for this omission. The relevant sentence in the Introduction now reads:

> "To address these issues, the second class of models – reduced complexity models (RCMs) – simulate phenomenological processes of arctic delta evolution using rule based trajectories of cellular automata (originating works in this field include, e.g., Murray and Paola, 1994, 2003; Murray, 2007)." (line 54)

***The background on the 2 m ramp is quite short and vague, given the heavy focus on it in the results. Please provide a more thorough description about what we do and do not know about the formation of the 2 m ramp. L47 seems to suggest that the ramps have permafrost, but there is no reference for this and I'm not sure that this is universally true. The 'sub-ice platform' on the Yukon described by Dupre (1980), for example, is reported to not have permafrost.***

There may be some confusion on what is meant by permafrost in this setting - the seabed has been shown to freeze seasonally due to the presence of bottom-fast ice. When seasonal freezing exceeds thawing in subsequent years, permafrost develops. We changed the text of the 2nd paragraph in the Introduction to:

'An important feature ubiquitous to Arctic deltas is the 2-m ramps, which we will interchangeably refer to as "ramp features", or simply "ramps" (Are and Reimnitz, 2000). Figure 1 shows such a ramp feature surrounding the Lena Delta, which dips gradually from below the sea surface to roughly 2-metre depth, but with localised variations on the order of a metre (Fuchs et al., 2021). These ramps extend from the above-water shoreline of Arctic deltas over tens of kilometres towards the open ocean (Reimnitz, 2002). The shallow inclination of the ramp serves to diffuse wave energy off-shore and may protect the delta from direct wave impact. The shallow water depth of the ramp corresponds to the range of maximum winter ice thickness (Wegner et al., 2017), meaning that ice freezes to the seabed at some point during winter, which is called "bottom-fast" or "bed-fast" ice (Dammann et al., 2018, 2019; Eicken et al., 2005), and that a seasonally frozen layer can develop in the sediment. When seasonal freezing exceeds thawing in subsequent years, permafrost can develop below the seabed (Osterkamp et al., 1989). Land- or bottom-fast ice and seasonal or permanent freezing may all act to stabilise the ramp (Overeem et al., 2022). Therefore, the ramp features, aside from being an integral part of Arctic deltas, may also play an important role in protecting Arctic shorelines from coastal erosion (Dean and Dalrymple, 2002) and could enhance carbon sequestration (Overeem et al., 2022). Moreover, the shallow-water platform provided by the ramp could play an important role in the surrounding ecosystems (Lopez et al., 2006). The origin, evolution, and stability of the ramp features are therefore an important feature of Arctic deltas. A better understanding of all three are required to predict how changing sea ice cover and air temperatures will impact Arctic delta morphology.'

**The methods section is noticeably missing any description of the boundary conditions used and what types of experiments were performed. The authors have stated there will be some comparisons to DeltaRCM and DeltaRCM-Arctic, and also attempts to specifically model the Lena delta. Much of this information is actually in the results section, but I suggest it be moved to the methods**

**section. A table listing all experiments performed and the appropriate parameters or processes that were changed would be helpful.**

The wall and ocean boundaries are treated identically to DeltaRCM, and no process changes outside of those described in Section 2.2 are used. To clarify this point to include boundary conditions, we have modified the sentence in the opening paragraph of Section 2 to become:

> "Any remaining parameters and conditions not explicitly listed take on values and specifications given in Liang et al. (2015b)."

We note that it is a balancing act between trying to include the model specifications close to the corresponding results (and not having to repeat it; i.e., the current state of the text) and putting the specifications as an addendum to the design of the model rules themselves (i.e., as another subsection of "Methods" as the referee suggested). After some additional experiments in moving the case specifications around, we have found that, for the purpose of keeping the length of the article shorter and less fragmented, it is preferable to keep the text.

Simulations that compare with DeltaRCM-Arctic configuration and simulations that are "Lena-like" each have their own set of parameters that are largely identical amongst themselves. Typically, there are just 1 or 2 changed parameters or processes in each variant from the "base" model of each cluster. These changes are explicitly given in the title of each plot (Figures 4, 5, 9 of the revised manuscript). Due to some parameters (and the processes being turned on/off in Section 3.2) being not a simple number, and the number of parameters being very large, it is not clear to us how to put everything in a table that is easy to read and does not require the reader to still go back to the text to find additional descriptions or explanations.

**It would be useful to know which model source code you started with, specifically, as you've listed two (Liang and Perignon). Neither of these is the most recent, though: pyDeltaRCM**

Aside from examining the two published articles by Liang and colleagues on DeltaRCM, we also studied and compared both Liang's MATLAB and Perignon's Python source codes, first to ensure consistency between them and later for clarifications if one has unclear portions. Therefore, we have not based our re-writing in Julia on a single source code. Due to the different nature of MATLAB, Python, and Julia in terms of how the compilers translate to machine codes, it is not generally realistic (or wise) to translate codes from MATLAB or Python to Julia in a one-to-one manner. (We further elaborate this point below in our response to concerns about the "refactoring" wording in our original manuscript.) We now take note, thanks to the comments and reviews, of pyDeltaRCM, which is professionally edited and has enhanced user-facing features, great documentations, and performance improvements. However, we expect the scientific output (and thus the scientific aspect of the algorithm) of pyDeltaRCM, which is also non-Arctic, to be identical to the older source codes. In fact, a substantial portion of the computation functions in the "backend" are inherited directly from Perignon's Python code, with some standardisation in syntax formatting.

***L105: this requires more discussion. Was the model tuned to reproduce some specific features using this parameter? Did you try tuning other parameters? Did you perform some sort of error analysis or are you just looking for features by eye? Please provide a physical justification for picking a new value for this parameter that differs from that in established literature.***

We agree that more clarification is needed on this. The gamma parameter was presented by Liang et al. (2015b) as a free parameter that "is usually a small value". In the companion "Part 2" paper (Liang et al., 2015a), they gave the expression reproduced in equation 4 of our manuscript as "a guideline". The parameter plays a determining role in the delta planform: whether it is elongated, as in muddy deltas, or fan shaped, as in sandy deltas. This influence actually comes from the slope term (proxy to the backwater slope) in the guideline expression, since the slope term itself is determined by the sand-mud fraction. In all of our simulations in which we compare with DeltaRCM-Arctic, we used the guideline

expression to determine gamma. However, since gamma contains delta-c (size of a grid cell) and the slope, we found that this value needs to be adjusted for the scale and slope for the Lena delta-approximant cases. In Liang et al. (2015a, b), they also tested a range of manually input gamma, from 0.02 to 0.15, to demonstrate how they resulted in different planforms. We found that the value of approximately 0.135 (± approximately 0.02) mimics the planform of the Lena Delta in terms of its deviation from a perfect semi-circular fan shape (considering the portion away from Arga Island). It does not affect our conclusion that the ramp feature forms under the modifications we present in our manuscript.

To make clear of the range this free parameter can take (and has been given in various test cases) in established literature, we have changed the text following equation 4 in our manuscript:

> "as a guideline to choose an appropriate value, where $g$ is the gravitational acceleration. The latter expression may have arisen from taking the ratio between the pressure gradient and inertia terms (without local acceleration) of the shallow water equations. The latter expression is the version for γ implemented in the source codes of DeltaRCM (Liang, 2015; Perignon, 2018), which would result in, for example, a value of 0.098 in the demonstration cases with 50% sand fraction instead of 0.05 in Liang et al. (2015b). We note that this remains a free parameter that has been given a range of values (e.g. 0.02 to 0.15) in various tests of its influence on delta planforms (Liang et al., 2015b, supplementary material)."

***Section 2.2 begins by stating that the authors refactored the DeltaRCM algorithm 'as we saw fit.' This is a little concerning, as it is hard to know what they mean by this. Did you test a non- refactored version against a refactored version to ensure you're still getting the same results? I also suggest comparing to the most recent version of the model, pyDeltaRCM, as suggested in the community comment.***

We regret the original phrasing that caused the concern. We have removed the "refactoring" wording. The refactoring only concerns purely computational aspects, since Julia is a C-like language, in which loops are more efficient without having to vectorise as in

the case of MATLAB or Python. Moreover, loops (e.g. "for" loops) are not only present in the MATLAB source code in abundance without vectorisation, but also nested in a way that is unfavourable for memory-access speed. (The MATLAB code is also not divided into methods/functions, acting more like a single script.) We made changes to the coding such that: (i) loops are used instead of vectorisation (which allocates memory, hampering speed in Julia), and (ii) the indexing of arrays inside the loops are iterating along dimensions along which the neighbouring elements are closest in computer-memory addresses. When, in the original source codes, multiple loops are created to iterate over multiple arrays in succession (especially in the MATLAB source code), we try to combine them into a single loop whenever possible, such that memory access is more efficient during runtime. We have taken great care to ensure that mathematical (thus "physical") operations performed in the model occur strictly in the same sequence as in the source codes by Liang (2015) and Perignon (2018). This is what we meant by "refactoring".

Since pyDeltaRCM is a setup-efficiency and user-experience feature improvement of DeltaRCM, the output of pyDeltaRCM should not differ from that of the previous source codes of DeltaRCM. In fact, a large part of pyDeltaRCM's "backend" computational code is the same as in Perignon (2018), but with additional standardisation of syntax formatting. We take note of the professionally reorganised and "cleaned-up" pyDeltaRCM, and have removed all runtime-performance comparisons with ArcDelRCM.jl from our manuscript to avoid distracting from the main topic of 2-m ramps.

***It seems to me that if the paper is about simulating arctic deltas and specifically about modifications to existing models, then all changes made to the model should be included in the main text, not an appendix.***

The feature changes in the appendices are purely for convenience in potential future use-cases. They are not actually utilised in the results of our manuscript. However, since we wrote these options into the code during testing, we felt that it may be useful to inform readers about them in an appendix.

**Section 2.2.2: maybe I'm missing something, but these units don't work out for your time dependent thaw depth. How do you get rid of the Kelvin?**

Indeed, there was a missing temperature term in the calculation of the positive-degree-day index. We have changed the description of '$I$' as follows:

> '[...] and $I$ is the "positive degree day index", which is the integrated number of days times the positive temperature since winter. For $I$ in our simulations, we use a mean temperature of 4°C to get $I$ (see Appendix B for the reason for this choice and the sensitivity of the model output to this value).'

And in Appendix B, we added the following final sentences:

> "The temperature we used to calculate $I$ is the average daily temperature (Figure B1) from 1st June to 30th September, which is 4°C. In addition to the graph metrics of the results presented in Section 3.2, Figure 6 shows the same metrics from the cases identical to the standard ArcDelRCM.jl runs except with $I$ starting count from 5 and 15 days and (independently) the average positive temperature being 8 and 12°C. They show that, within the ranges tested, these parameters do not affect the resulting deltas beyond the internal variability of each case. To help illustrate this visually, an example of each case is shown in Figure B2."

We have re-run all of our simulations accordingly, without observing changes that would necessitate changing our main conclusions. We have also added model runs for 8°C and 12°C to demonstrate that changes in this range do not change our conclusions either.

We apologise for this mistake, and are grateful for the referee discovering this and alerting us. Thank you!

**Also, is thaw depth not dependent on there being standing water? Just dependent on positive degree days? We know that there are taliks under water bodies, which is what the rules in DeltaRCM-Arctic try to simulate by enforcing**

**some minimum thickness for permafrost cells. Can cells under channels be permafrost in your model?**

The thaw depth depends on the time elapsed since temperature becomes positive and the positive temperature in that duration. We note that DeltaRCM-Arctic does not take into account standing water or lack of it in terms of thaw depth or active layer, which is static throughout. Similar to DeltaRCM-Arctic (according to Lauzon et al., 2019, Text S2 in the supplementary material), we update the frozen status at every time step. For DeltaRCM-Arctic, it is through the fraction of (vertical) "permafrost" cells in a sediment column and switching on/off the "permafrost" label (thus the erodibility factor, *E*) for *horizontal* grid cells accordingly. For ArcDelRCM.jl, it is through updating the thaw depth in a sediment column, above which erosion can proceed unhindered (i.e., without scaling by an erodibility factor). The minimum-thickness enforcement concerns "permafrost" labels for the *horizontal* grid cells, for which there are no such labels in ArcDelRCM.jl. Erosion is limited or allowed only based on whether a *vertical* sediment cell is frozen or thawed. Vertical (sediment) cells directly under flowing channels cannot be a "frozen cell" in ArcDelRCM.jl, unless it is during *only* the first time step in a scenario where we set the positive-degree-day index to begin with 0, which is *not* the case in any of our model usages (we start the count with 10). The only way the top cell of a sediment column can remain frozen is if there is bottom-fast ice sitting directly on top. As soon as the ice ceases to be bottom-fast, the thaw depth progresses down the sediment column, and the top cells are "thawed" (free to be eroded by passing water packets).

To clarify how each model uses or forgoes the use of the "permafrost" label for grid cells, we have modified the last sentences of Section 2.2.2 ("Time-dependent thaw depth") to read:

'As a result of the time-variable thaw depth, we forgo the "permafrost" label for horizontal (or planar) grid cells. In usages where the classification of vertical sediment cells as permafrost is relevant, it can be defined as the vertical cells that stayed below the maximum thaw depth (instead of a static depth of 0.5 m in

DeltaRCM-Arctic) for at least 2 years, although we do not use such labels in this study.'

We agree that taliks are not explicitly modelled by any of these models.

***Since the authors changed both the erosive rules for permafrost and the rules for how permafrost 'forms' in the model, it is not clear how both of these changes in combination affect the results. In order to assess the new permafrost rules, please also provide figures and analysis in the paper of permafrost extent and evolution in the model as compared to DeltaRCM-Arctic. Is there permafrost under the channels? On channel banks? On the ramp?***

Since ArcDelRCM.jl forgoed the use of the erodibility factor, the use of a "permafrost" label for a (horizontal, or planar) grid cell is also dropped. (We have elaborated on this in our response to the previous point.) Instead of scaling down the erosion of "permafrost" (horizontal) grid cells, ArcDelRCM.jl approaches the process by allowing unhindered erosion (i.e., without an erodibility factor) of thawed (vertical) sediment-column cells. DeltaRCM-Arctic uses a fractional measure of frozen versus active-layer thicknesses to flag a horizontal grid cell as "permafrost", and then uses a fractional factor to reduce erosion on that cell. ArcDelRCM.jl allows erosion to proceed unhindered on thawed vertical sediment-column cells, but restricts erosion of frozen vertical cells, resulting in a passive control on erosion by the fraction of frozen versus thawed layers in individual horizontal grid cells. At the same time, the thawed-frozen boundary progresses downwards with each time step in a single model year. In effect, both models allow erosion in horizontal grid cells with substantial vertical frozen sediment cells. The difference is that DeltaRCM-Arctic allows scaled-down erosion persistently in horizontal grid cells where there are enough vertical frozen grid cells to make the binary classification as "permafrost" (horizontal) cell. ArcDelRCM.jl allows unhindered erosion "up-front" until the thawed (vertical) cells are exhausted (but more will be available again in the next time step). Both are effectively fractional erosion (compared to completely unhindered non-Arctic cases), but the timing of when the top layers can erode is different.

To clarify this point, we have expanded the text at the end of Section 2.2.3:

> 'We thus forgo the use of the erodibility factor, E (and, as mentioned in Sect. 2.2.2, the labelling of any pixel as "permafrost").

> 'Conceptually, both DeltaRCM-Arctic and ArcDelRCM.jl allow erosion of sediment columns with frozen (vertical) cells. Consider a sediment column with a significant fraction of its cells frozen during some number of time steps (i.e., with a shallow thaw depth in ArcDelRCM.jl, or with the corresponding horizontal grid cell is labelled as "permafrost" in DeltaRCM-Arctic). DeltaRCM-Arctic would allow for erosion with a scaled-up flow-speed threshold through E (and a similarly scaled-down bed diffusion); the cumulative erosional effect on the corresponding column can thus be thought of as a fraction of the equivalent no-permafrost case (i.e., E = 1). By allowing only thawed (vertical) sediment cells to be eroded, the same sediment column in ArcDelRCM.jl also undergoes erosion that is a fraction of an equivalent case without any frozen (vertical) cells. However, the main difference between the two is the timing of the erosional events. ArcDelRCM.jl allows the thawed vertical cells to be eroded unhindered and thus earlier (considering the chance of having lower versus higher flow speed across the domain at any given moment), but then delay the erosion of the at-the-time frozen vertical cells, which would become thawed in the next time step and freely erodible again.'

Due to the lack of "permafrost" labels for horizontal grid cells as used in DeltaRCM-Arctic, and the resulting treatment imposed by this label, it is unclear if generating permafrost-extent maps (in the horizontal, plan-view sense) using DeltaRCM-Arctic rules on the outputs of ArcDelRCM.jl would result in a meaningful comparison.

At 4ºC of positive average temperature, the time-dependent thaw depth indeed remains between 0.2 and 0.3 m between positive-degree-day 10 and 20 (since the 'l' count starts from 10), which is the simulation length prescribed by Lauzon et al. (2021) and Piliouras et al. (2021). However, for the simulation lengths that include the summer months (122 days) in ArcDelRCM.jl, the 0.5-m active-layer depth prescribed by DeltaRCM-Arctic is reached a

week before the halfway point (54 days) and goes on to reach 0.75 m by the end of the model year, following a square-root function shape in time. According to Piliouras et al. (2021), the 0.5-m active-layer depth is consistent with the characteristic active-layer depth in the zone of continuous permafrost, citing Walker (1998). However, this characteristic depth may not be reachable within the first 10 days of the spring-flood conditions, and therefore the treatment is not necessarily self-consistent for all Arctic deltas. This possible inconsistency may have been mitigated by the use of erodibility factor-scaled erosion. This is one of the reasons why we included a subsection to emphasise the importance of including not only 10 days in each model year.

Since lateral erosions are simulated in an approximate manner by the "bed diffusion" step (Liang et al., 2015; we elaborate more in our response to a similar point raised by the referee—the point after the next), we made comparison runs in which the bed diffusion step allows unhindered erosion (i.e., not even an erodibility factor or thaw-depth limitations) versus the limited-to-thaw-depth treatment of standard ArcDelRCM.jl. We show this in the violin plots (Figures 6 and 7, the cases labelled "unhindered bed diffusion"). There are no differences between them.

In terms of the 2-m ramp formation, which is our focus, as shown in our higher-temperature cases (thus faster thaw-depth progressions) or our static thaw-depth cases (which has deeper thaw depth than the time-dependent cases), allowing more erosion may slightly encourage off-shore depositions that would increase the prominence of a 2-m ramp (Figures 5c-f, 6, and 7).

We agree that this subject is interesting and believe that it is substantial enough to warrant its own, separate manuscript in the style of Piliouras et al. (2021).

***How sensitive is the model to the choice of I=10? This should be justified further.***

We have run simulations with $I = 5$ and $I = 15$, and included their graph metrics in the violin-plot figure (Figure 6) and a sample of each in Appendix B. The violin plot shows that the cases for $I = 5$, $I = 10$ (the standard we used), and $I = 15$ are statistically

indistinguishable. Due to the correction of the positive-degree-day index (with the temperature term added back in), we also tested the cases where the average temperature (let's call it T⁺) is 8 and 12 degrees Celsius instead of 4 and show them in the same figures and sections. These too show no discernible differences in the graph-metrics. In terms of our main focus of the size of the 2-m ramp feature, the variations of *I* and the average temperature (T⁺) also do not produce statistically significant differences (see the "number of pixels with elevations in [-2.5, -1.5)" plot, Figure 7). However, there appears to be a very slight tendency (though remaining well within the scatter range) for the ramp to increase in size as *I* and T⁺ increase.

***I'm particularly concerned about the treatment of permafrost erosion in the new model. Based on my understanding of the text, the model assumes, or rather asserts, that permafrost can never erode, which is not accurate. I assume this because they state that cells can only erode down to the thaw depth. Permafrost riverbanks absolutely do erode. Does this mean that lateral migration cannot occur if the cells immediately adjacent to a channel are permafrost? If erosion is limited only to the thawed layers, and thawing happens based only on a degree day index, then won't riverbanks always be frozen at depth and therefore completely non-erodible. This is not physically realistic. Please provide justification for this choice and/or a further explanation of the functionality of this choice in the model.***

We would like to clarify that, unlike DeltaRCM-Arctic, we do not classify horizontal cells as "permafrost" or "not permafrost". Operationally, the only classification relevant to erosional limits is whether a vertical sediment-column cell is "frozen" (below thaw depth at the current time step) or "not frozen". Therefore, sediment columns in pixels adjacent to channels *can* erode.

The models DeltaRCM and DeltaRCM-Arctic, and thus inherited by ArcDelRCM.jl, erode through two mechanisms: (i) removal of sediments by packets whose routes pass over the horizontal grid cell, and (ii) the "bed diffusion" process in which bed elevations are in effect

smoothed according to the local sand flux and slope (Liang et al., 2015b, Equation 24; Lauzon et al., 2019, bold-fonted equation in the middle of the third page; Piliouras et al., 2021, unnumbered equation at the bottom of page 3). These rules are further conditioned by the rule, from the original DeltaRCM model (Liang et al., 2015a, b), that the sediment column cannot change in thickness by over 25% of the water depth in a time step.

None of these rules explicitly model block collapse due to thermo-erosional niches, nor other forms of water/sediment packets acting laterally to erode from adjacent cells.

The lateral redistribution of the sediment bed comes only from the bed-diffusion process. This was described by Liang et al. (2015b) as:

> "This topographic diffusion also introduces lateral erosion by allowing sediment on the bank to be removed and added to the channels. This lateral erosion gives channels the mobility to migrate or even to meander."

To make this clear to the readers, we have expanded the opening sentence of Section 2.1.5 to:

> 'Immediately after each round of sediment packet routing, a bed-diffusion process is applied "to take into account the influence of topographical slope on sediment flux", and to introduce "lateral erosion by allowing sediment on the bank to be removed and added to the channels" (Liang et al., 2015b).'

Our model outputs (Figure 5c and 5e versus 5d and 5f, and the graph metrics in Figure 6 labelled as "unhindered bed diffusion", which indicates that neither an erodibility factor nor a thaw-depth limit was used) show no significant impacts by the bed-diffusion process, whether or not erosion is limited (vertically) to thawed layers.

The rules limiting erosion (vertically) to (vertical) layers above the thaw depth (which deepens at every time step) are introduced such that the rules are logically consistent with the protection by bottom-fast ice, which can bond with and partially freeze the substrate through conduction (Reimnitz, 2002). The treatment of bottom-fast ice necessitates

considerations of a shallow thaw depth at the time steps right after the bottom-fast ice ceases to be bottom-fast. Thaw-depth progression is then considered in all of the simulation domain with a sediment column to preserve internal consistency of the model. Theoretically, one *could* re-introduce an erodibility factor for the (vertical) grid cells below the thaw depth, if its re-introduction could be shown to have significant impact to model outputs that are directly called for by observations. However, in terms of our focus of the 2-m ramp, the main effect (whether a ramp has formed or not) is ice-related. Changing how initial thaw depth (in pixels with no bottom-fast ice; i.e., the *l* index) and changing the speed at which it deepens, i.e., changing *l* and $T^+$, do not lead to significant effects on our conclusions.

To clarify this point further in the text, we have modified the final sentences of Section 2.2.2 to read:

> 'As a result of the time-variable thaw depth, we forgo the "permafrost" label for horizontal (or planar) grid cells. In usages where the classification of vertical sediment cells as permafrost is relevant, it can be defined as the vertical cells that stayed below the maximum thaw depth (instead of a static depth of 0.5 m in DeltaRCM-Arctic) for at least 2 years, although we do not use such labels in this study.'

We have also added the parenthesised qualifier in the following sentence in Section 2.2.3:

> "If the calculated erosion reaches deeper than the available thawed layers, the erosional depth is limited to the thawed (vertical-cell) layers only."

**L290 also states the authors compare to a DeltaRCM-Arctic run with E_p = 0.65, suggesting that this mimics their restrictive erosion. But this is not true, because the ArcDelRCM implementation does not seem to allow any permafrost bank erosion.**

We have run additional simulations mimicking DeltaRCM-Arctic where $E_p = 0.8$, and found no significant differences to the metrics or our conclusions. We have now switched to

using the $E_p = 0.8$ case as our standard case for comparison in the figures (but retain the $E_p = 0.65$ case in our graph analyses in addition to the $E_p = 0.8$ case). This, as we found, is more favourable for DeltaRCM-Arctic when it comes to off-shore depositions that provide material for the ramp feature, but is not enough to form the ramp.

(We have responded to the lateral erosion point above.)

**L233: Please be more specific about changes to delta t ('increased by a factor of a few.'). How did you discover this? Was it unstable for all other values? Is there a range of ok values?**

The discovery was due to frequent occurrences of numerically unstable simulations either crashing the code or producing "deltas" that are "stuck" having only one channel bending towards the sides of the simulation domain. The guideline expression for delta-t given by Liang et al. (2015b) for DeltaRCM is intended also for numerical stability. For this guideline, they used the ratio of inlet size and the input sediment discharge, divided by 10. We found that for our Lena-scale simulations, we need to increase the factor in the denominator to at least 40 to not have consistent numerical failure of runs (it became quite safe in the neighbourhood of 80, but the small time steps lengthen the run time). This is admittedly somewhat of a trial-and-error, as the numerical stability can be affected by individual random seeds near the threshold range. That is why we did not want to specify a hard threshold. Accordingly, we have modified the following sentences in Section 2.2.5 ("Time-step size") to read:

"Specifically, $\Delta t = \frac{N_0^2 h_0 \partial_c^2}{10 Q_{s_0}}$ is given as a guideline by Liang et al. (2015b) to prevent too much sediments from entering in each time step relative to the accommodation space for deposition in the grid cells. We discovered in our simulations intended to mimic the Lena Delta, where the grid-cell dimensions (which are terms in the numerator) are several times larger than in Liang et al. (2015b) and Lauzon et al. (2019), that the 10 in the denominator of expression for $\Delta t$ needed to be increased

by a factor of a few (e.g., to 40 – 80, depending on the volume of sediments entering the domain in each time step in the specific simulation)."

And added the following sentence to the end of the subsection:

"Due to the random-walk nature of the RCMs, this threshold is not sharp."

***L260: This addresses temporal changes to the input water discharge, but what about sediments? Are they just scaled with Qw? Is that realistic based on Arctic river sediment flux timeseries? Please provide more information on how sediment input is treated and justify this choice.***

Based on the data shown in Holmes et al. (2002), which is also cited in Overeem et al. (2022), for Lena delta (and several others such as Ob and Yenisey) the sediment fluxes scale rather directly with water discharge. Therefore, we follow the DeltaRCM approach to simply apply a scaling factor to translate water discharge into sediment discharge. We have added the following sentences to the end of the subsection (2.2.6):

"The sediment discharge, $Q_{s0}$, can also be input as a time series. However, many large Arctic deltas have sediment fluxes that scale approximately with water discharge (Holmes et al., 2002). Therefore, following DeltaRCM, a simple multiplicative factor is used to translate $Q_{w0}$ to $Q_{s0}$."

***I think the authors have missed an opportunity in the results section to just directly compare to the publicly available model output from previous DeltaRCM-Arctic runs, noted in the acknowledgments of Piliouras et al., (2021): "Raw model outputs are available in Piliouras et al. (2020) through DOE's ESS-DIVE repository: 10.15485/1682304. "***

We noted the existence of the model outputs, and did consider whether to use them for Section 3.2 of our analyses. However, considering that we are focusing on cases with 2-m ice thickness (due to the ramp-feature focus) and that the published outputs only contains 0.5m, 3m, 4.5m cases (which are suitable choices for their purposes), and that they have

triplicate runs while we are comparing an ensemble of 105 each, we had decided to run our own. Moreover, neither their outputs nor our attempt in reconstructing the DeltaRCM-Arctic configuration produced the band-like 2-m ramp feature (although both produced tentacle-like off-shore deposits). Our purpose is to start from a well-performing base model (DeltaRCM-Arctic, or as close as we can reconstruct it with only published information) that does not produce the 2-m ramp, and explore what self-consistent changes in physical processes/rules would favour or lead to the formation of the ramp feature. We take the referee's suggestion as an idea to devote time and resources for a separate manuscript in the style of Piliouras et al. (2021), which is substantial on its own, as mentioned in another point above.

***Information about the use of graph theory/network-based techniques should be included in methods. Extracting channel networks from model output is not a simple task, and it can be rather subjective. Please include details about how channels were extracted, what methods were applied to those channel networks, and how. Also, why not use graph theory methods designed specifically for river and delta channel networks, such as those discussed in Tejedor et al., (2015)? Also, why was 105 chosen as the number of realizations to include? The authors suggest that based on their analysis of some graph theory metrics, they did not find statistically significant differences between the various runs, including between Arctic and non-Arctic runs. How do you reconcile this with the fact that Lauzon et al., (2019) and Piliouras et al., (2021) did show noticeable differences in some aspects of the channels? Those studies have only 3 replicates and did not do a formal statistical analysis, but do you think those findings were not representative? Or is it a matter of the metrics tested?***

Thank you for pointing out the missing description of how we extracted these channels. We have added the following subsection and figure into the methods section (Section 2) describing details on the methodology to extract the graphs from the delta runs and the methods applied to compare the ensemble runs of the different modifications:

'2.3 Graph Analyses on Ensemble Runs

[revised manuscript text omitted]

Further, the code to compute the delta graph metrics proposed by Smart & Moruzzi (1971), Edmonds et al. (2011), and Tejedor et al. (2015) has not been made available to the reader, making a direct implementation of their approach considerably more difficult and time-consuming. As the metrics described here, adapted from Rettelbach et al. (2021), already give a good foundation for comparison of the different parameterized simulations, we decided to reside with these.

Concerning the number of runs per ensemble, 105, the reasoning is as follows: We had a target of 100 realisations each. However, to insure against rare but possible failures due to individual cores or nodes on the computing cluster, we added 5 to each so that we can

ensure that a minimum of 100 realisations each can arise from a single execution (with random seeds) rather than a patchwork of multiple executions.

Regarding the graph metrics, we would also like to report an error in the original violin plot arising from an unexpected behaviour of Julia's StatsPlots.jl package. In case the details are relevant, here is a description of this unexpected behaviour (also included in the Overview of Major Changes at the beginning of this response document):

In the StatsPlots.jl package, to plot a DataFrame (i.e., a tabulated data type/structure comparable to Python's DataFrame of the `pandas` module) as violin/box plots, two inputs are needed per plot: the classes (in our case, they are the simulation names) and the metrics to be plotted (e.g. number of nodes). The latter is passed into the plotting function as a "column" of the DataFrame. The former, we originally thought, is an array/vector of the classes with the length of this array equal to the number of classes. The function gave no warnings or errors and plotted the violin/box plots in the original manuscript. Since the actual values of the metrics plotted were not drastically different, we did not notice any issues immediately. However, during the revision phase, we added another metric (the number of pixels with ~ 2 m elevations) during our analyses, and the values there do show enough differences that we became suspicious of the plotting function. We found that we were supposed to feed in the "classes" column "as is" (i.e., with the hundreds of repeat entries for each realisation, rather than a short vector with unique entries) for the plots to display correctly. We felt lucky to have this opportunity of revision to catch this subtle, uncaught issue with the plotting package.

The graph metrics now show some differences, although still significantly overlapping in scatter range, between various cases and models examined in our manuscript. We have moved the graphs section from Appendix C in the original manuscript back to the Results section (now Section 3.3) along with updated discussions in Section 4. However, while there are some tendencies for the non-Arctic DeltaRCM to have a higher number of connected components and a correspondingly lower graph density than DeltaRCM-Arctic, their spread overlaps with each other. The outputs presented by Lauzon et al. (2019) and

Piliouras et al. (2021) indeed show some differences between various degrees of ice coverage, erodibility, and ice thickness. However, we believe this does not necessarily conflict with the outcome of our graph analyses. In Piliouras et al. (2021), they wrote the following in their discussions:

> "Previous experiments showed that permafrost tended to decrease the number of channels due to the difficulty of incising into resistant deposits while ice increased the number of channels, owing to the increased tendency for overbank flow (Lauzon et al., 2019). When ice and permafrost were combined in these experiments, these two effects balanced each other, such that the average number of channels on deltas with both ice and permafrost was not notably different from that on deltas without ice and permafrost. This suggests that the number of active channels on Arctic deltas may be not be readily distinguishable from that on temperate deltas of comparable discharges."

Lauzon et al. (2019) earlier also made a similar observation in their analyses of channel dynamics between full, low, and high erodibility. These observations are the ones connected to what our graph metrics cover, and they do not conflict with each other.

Another possible hint that our results and theirs are reconcilable could be found in Figure 7c of Piliouras et al. (2021), which plots the number of channels against the various cases they tested. In the plot, the error bars (from the triplicate runs of each case) show clear overlaps with each other. This suggests that, while they may well be differences, these differences may also be drowned out by the random nature of the RCM when many instances are realised.

***Regarding the comparisons to the Lena delta, why was this delta chosen over others? Why not other arctic deltas? Is there something unique to the boundary conditions of this system that might result in unique deltas or features compared to other Arctic deltas? Some discussion on this is warranted.***

This is a good point — we have not done a good job of establishing the basis for our comparison. Frankly, comparison to the full spectrum of Arctic delta morphologies is beyond the scope of this paper. Many of the other Arctic river mouths (Ob, Yenisei, Mackenzie, for example) are estuarine or geologically confined, meaning that they poorly match the modelled geometry. Other Arctic deltas could have been used (e.g. Olenyok, Colville), but are much smaller and therefore more likely to be influenced in their growth by processes other than those modelled here, fluvial sediment delivery in a cold climate setting.

We have added to text to the introduction:

> "The largest Arctic delta is the Lena Delta, which has fan-shaped morphology with multiple channels leading discharge from its epicentre to the delta edge. Although complicated by neotectonics (Are and Reimnitz, 2000), its large-scale structure is probably determined by regional relative sea level effects on fluvial sediment budgets within the delta and at its edge (Whitehouse et al., 2007)." (line 30)

> "Through these outputs, we demonstrate the model's capability in reproducing the 2-m ramp, identify the processes that led to their formation in the model and make an exemplary comparison of these outputs with a real-world case, the Lena Delta." (line 80)

**L340: Given the differences in inputs, I'm not sure this comparison is appropriate. The deltas should be compared when they have the same total volume input.**

Our argument was to not limit the number of model days per model year to 10, precisely because larger deltas remain active outside of the spring-flood season, with additional sediment input. We completely agree that the differences are due to the different total volume input. We are trying to demonstrate the differences it makes by taking 150 years with only 10 days of evolution per year, and 150 years with 4 months of evolution per year. This demonstration is in response to previous use cases of DeltaRCM-Arctic assuming that 10 days per year is suitable to model a smaller Arctic deltas, which is alright for the

purposes of Lauzon et al. and Piliouras et al., but is not suitable for our Lena-Delta approximant cases. To clarify the point that our purpose is to show the difference in days-per-model-year and thus the cumulative volume of sediment input, we have modified the first sentences of the second paragraph in Section 3.4:

> "We would first like to find out the differences arising from assuming a 10-day model year (suitable for smaller Arctic deltas, as adopted by Lauzon et al., 2019; Piliouras et al., 2021) and from assuming a 4-month model year including the summer months (suitable for large Arctic deltas such as Lena Delta and Mackenzie Delta) due to the resulting different total sediment input per year. We ran one batch of simulations for 150 model years.Within this batch, the discharge $Q_w$ is treated differently in order to probe the difference it makes in terms of the number of days per simulation year and thus sediment input per year, given realistic $Q_w(t)$."

Our intended take-away point is that, in order to investigate delta-feature (e.g. ramp) growth and evolution over years, centuries, or longer, the summer activities of large deltas (and thus the associated sediment input) should not be omitted. We use this demonstration to lay the ground for our usage of longer model years.

**Figure 8: can you show side by side with a control for those variables that changed?**

Unfortunately, due to the length and output-file sizes of these 1200-year simulations, we did not (and could not within the response period amidst other re-runs of simulations) produce a control version of such duration. The only differences between this (now Figure 10 in the revised manuscript) and the would-be "control" case is that: (i) the sediment-to-water volume fraction is the ratio observed (Boike et al., 2019), which is 1/10th of the cases in Figure 9 of the revised manuscript (Figure 7 in the original); and (ii) the ocean depth (or accommodation space) is linearly tiled from 10 m at the inlet wall to 20 m at the far end of the simulation domain, as opposed to a uniform 15 m. Since the primary difference is due to a scaled sediment volume input, the "control" case with 10 times the sediment input would have grown much faster and hit the simulation-domain

boundaries much earlier. The tiled seabed, which is mimicking the bathymetry of the Laptev sea near but outside of the Lena Delta, is the reason the ramp is as visible in the output. Having a uniform 20-m accommodation depth (i.e., if we take a far off-shore ocean depth as our uniform ocean-basin depth) would cause the ramp to be very diminished, similar to Figure 4e and 9c, which is one of our conclusions regarding the ramp feature — a large accommodation depth can prevent the ramp from forming.

***L366: why would the atmospheric melting period be shortened in the future? Please provide a justification for this choice.***

This is due to the thinner ice thickness. The rate of atmospheric heat-induced melting is controlled by the duration of the melting period in all Arctic models discussed in this manuscript. We have rephrased the sentence to read:

> "[...] the atmospheric heat-induced melting of ice cover is brought forward by 10 days. The duration is also shortened by 10 days due to the thinner ice."

***The discussion should put the results re: ramp formation back in the context of what we already know from literature. For example, L376 should include some references. L378-380 has also been suggested by previous papers, including older observational studies and Piliouras et al., (2021).***

We have added reference to Reimnitz (2002) to the original L376 (now L474). We have also added the following sentence after the original L378-381 (now L492+):

> "Previous work by Lauzon et al. (2019) and Piliouras et al. (2021) using DeltaRCM-Arctic observed that offshore deposits increase with $h_{ice,max}$ and thus with decreasing accommodation space in the ocean basin. Our results are therefore in agreement with theirs."

***L396-399: Can you quantitatively compare the amount of in-channel erosion between the two cases for locations where h_ice = 99.99% of flow depth? it***

***would be helpful to understand how important this rule is, and how much its effect varies spatially.***

In lieu of a modification of the source code that requires care (and considerable amount of time) and would impact performance, we instead give the measure of pixels with the condition "$h_{ice} = 0.9999\,h$ AND $u > 0.3$ m/s" (where $h$ is the flow depth and $u$ is the flow speed) versus pixels with "$u > 0.3$ m/s" (Lauzon et al., 2019 used this threshold to identify fluvial channels with at least enough flow speed to erode) at the last time step that has maximum ice coverage. For this, we ran additional quintuplet simulations where every time slice is saved (other runs we performed had long intervals between writing snapshots to file in order to circumvent disk-space limitations) with the configuration identical to those that gave Figure 4b and 5b in our revised manuscript.

All of the cases have 40% maximum ice extent, meaning the inner area within 60% of the delta radius from the inlet is ice free (this is simply a case frequently used in previous publications of DeltaRCM-Arctic, especially in Piliouras et al., 2021). Across the quintuplet, an average of 7852 pixels belong to active channels that have erosive power with $u > 0.3$ m/s (including ice-free areas), of which 1336 are active channel pixels with $h_{ice} = 0.9999\,h$, or 17% of all channel pixels with erosive power (the min-max range of the quintuplet is from 15.5% to 18.5%). With a few exceptions (14 pixels on average), all of the average 1336 pixels have computed flow speed that is clipped at $u_{max}$, which is set to 2.0 m/s as per the original DeltaRCM model, representing the maximum erosive power allowed in the model. The 17% value would become higher if the maximum ice extent is greater.

To visualise the spatial distribution of these pixels, the following four figures show one of the quintuplet cases at the last maximal ice cover, in the following order: (i) the ice thickness, (ii) the pixels with $h_{ice} = 0.9999\,h$ (1 for true, or 0 for false), (iii) the pixels with $h_{ice} = 0.9999\,h$ AND $u > 0.3$ m/s (colours are $u$ in m/s), and (iv) all pixels regardless of ice thickness with $u > 0.3$ m/s (1 for true, or 0 for false).

[Figure]

[Figure]

[Figure]

[Figure]

**L404-407: If the ice is bedfast, how is deposition occurring beneath it? Wouldn't it have to occur in front of it (i.e., upstream of it)? It looks like you have sub-ice channels in the ramp, though you have not discussed this. These are presumably responsible for the construction of the ramp under the ice, no?**

Indeed, sub-ice channels exist and contribute to ferrying sediments that would otherwise be blocked by bottom-fast ice towards the shoreline of the delta. We found that the place to add the mention of sub-ice channels would be two paragraphs before, when we used the term "unblocked pathways":

> "This enhances flow constriction by ice, which focuses erosion on the few unblocked pathways (e.g. sub-ice channels), and leads to sediments being carried farther seaward." (line 516)

**L429-430: There are many satellite images of the Lena delta where ice on part of the ramp is already melted while ice in many channels remains (e.g., Figure 7 in Overeem et al., 2022). This is in contrast to your rule that delays melting on the ramp, which you claim is a major reason for its formation. Is it realistic to force the ice to stay on the ramp for a period of time? Doesn't this somewhat contrive an intended result? Shouldn't the ice be melting uniformly everywhere by incoming radiation? What is the justification for delaying it?**

When ice is bottom-fast along the shore of the delta, most flood water flows over the ice instead of under (despite the existence of under-ice channels), leading to the appearance

of ice retreat from the delta shore when in fact the darker water is merely covering the ice underneath (and depressing it). Both stream temperature and air temperature are still close to the freezing point at that flood time (e.g. Juhls et al., 2021, Yang et al., 2005). The flood water comes from thaw and melts from upstream regions (in the south) that became warmer earlier than at the delta. The ice seen in the channels in Figure 7 of Overeem et al. (2022) appears to be the so-called "serpentine ice", which are floating ice (Juhls et al., 2021). Without seeing the satellite time series, it is difficult to be certain at which stage of the spring flood is the Lena Delta at the time. However, from the visual appearance of the on-delta areas, it is likely that it is at the tail end of the spring flood. Our co-authors Juhls and Overduin are experts of the Lena Delta and also have strong field experience there in addition to studying it using remote-sensing data. The delayed break-up of bottom-fast ice sitting atop the 2-m ramp is based on observations (also described in Reimnitz, 2002, especially Figures 10 and 12) and shown in the satellite-image time series in Figure 2 of our updated manuscript (Figure 3 in the original). The fact that ice on the ramp remains bottom-fast for longer is not a contrived condition, and may be an important element in the formation of the ramp feature. That being said, we would also like to stress that the results in our Figures 4 and 5 do not have this delay.

**L484: Where is this comparison in the text? I do not remember seeing these metrics**

This was shown in Figure 9 (Figure 7 in the original manuscript). We have also added a parentheses to the relevant sentence:

> "[...] but reaching twice as far from the inlet wall (or four times the area) as the 10-day cases."

**Technical comments/Rephrasing suggestions**

**L22: 'key interfaces between permafrost landscapes and the Arctic Ocean.' Are Arctic deltas themselves not permafrost landscapes? I would rephrase this.**

We have rephrased it such that we are not inadvertently excluding Arctic deltas from permafrost landscapes: "Arctic deltas are key components of the permafrost landscape, connecting the permafrost areas upstream and the Arctic Ocean."

**L55: This sounds contrived, like you are forcing a result. Presumably you are making modifications to the rules to include more physics or more processes, with the hope that you will reproduce a 2m ramp. Please rephrase. The following paragraph should similarly be rephrased. I'd hope that one purpose of the article is to explore the processes that shape Arctic deltas and to better understand those that might contribute to development of the 2 m ramp. As written, it sounds like the purpose is purely to develop a model and present model output.**

Indeed, it was an unfortunate phrasing. We were interested in the 2-m ramp feature and wondered if we could find the processes/conditions conducive to it through a modelling process. Since we arrived at a model that can reproduce the ramp feature, it is unavoidable that one of the main components of the manuscript is to present the model itself. We do acknowledge and take note of the tone-problem pointed out by the referee. To mitigate this issue, we have rephrased the sentence to become the following:

> "We then explored the effects of modifying the rules to include additional processes and physics, with the hope of identifying and understanding the circumstances that favour the formation of the ramp feature."

The following paragraph has been rephrased to:

> "The purpose of this article is two fold. First, we present and motivate the physical-processes rule changes we made to our reconstructed version of DeltaRCM-Arctic to arrive at the ramp-producing ArcDelRCM.jl version. Second, we present the model outputs, including ones that are intended to simulate the evolution of large-scale Arctic deltas. Through these outputs, we demonstrate the model's capability in reproducing the 2-m ramp, identify the processes that led to

their formation in the model, and make exemplary comparisons of these outputs

with a real-world case, the Lena Delta. [...]"

**L107-109: this should be rephrased to focus on what the model steps are, not what sections are in the cited paper.**

We have modified the sentence to become: "We note that, in practice, the unit-discharge vector field serves as the flow-direction field and thus computed during the routing of the water packets. Its underrelaxation by the free parameter γ is done directly when computing the routing weights (Equation 3)."

**Section 2.2.4: I suggest renaming this subsection, as this section does not actually describe shore or bank migration. The phrase shore/bank migration implies some redistribution of sediments, whereas you are describing a modeling step that simply smooths the water surface.**

The first author was affected by sea-level work to which he has previously been exposed and thought of the horizontal correction of water coverage after a vertical level change as "shore migration". However, we understand the potential for confusion with different terminologies here and decided to simply adopt the terminology used in the original source code, "flood correction".

**L277: What does this mean? That the atmospheric melting can change over time and that it can be nonlinear in its rate change?**

Correct. However, since we are defocusing on the technical descriptions of the model itself, and since we have not used this feature (i.e., hyperbolic tangent time profile of melting) in any of our results, we find this sentence distracting and have thus removed it. Users working with the model will still see this option mentioned in the comments/doc-strings of the source code.

**Please include a legend on Figure 6.**

A legend has now been added.

***Figure 9 (and other figures with this color scale): please adjust the color scale. The ramp feature is not particularly visible. Maybe you can change the colorbar to a log scale so we don't see so many numbers that are all black?***

In Figure 9 (now Figure 11), the ramp features are either partially disrupted or fully gone. For all the figures, the emphasis of the colour scale is in the range around 0 to 3 m depth, which is where the ramp feature is supposed to be (i.e., dipping from just under the sea surface down to roughly 2 m depth, in the case of 2-m ice thickness). The colour scale therefore contains colour-blind distinguishable colours from 2, 1, 0, -1, -2, -3, and -4 m. We agree that having the rest of the colour scale be all black is less than ideal. Therefore, we made small adjustments so that all values from -5 m downwards would have a darker step of grey every 2 m until black (RGB 0, 0, 0 at -20 m). We have applied filters to our monitors to verify that individual grades can still be distinguished in various colour-blindness settings.

***Figure C1: Please label the run names on the plots with rotated text instead of the numbers for the various runs. There is no table in the text, and these numbers/IDs are not used elsewhere, so readers cannot readily identify which is which.***

We have completely redone this figure, partly due to the aforementioned unexpected behaviour of the plotting package. The new figure now uses complete, descriptive labels that match the sub-figure titles of Figures 5, rather than number-letter combinations as 'IDs'.

We thank the referee again for the affirmative assessment, the time and effort spent on reviewing our manuscript, and the constructive suggestions.

---

## Author Response (AR2)

**Response to Referee #2's Comments**

We quote the original comments in **_bold-italics_** typeface, and give our responses in light typeface below each section.

**_The authors have made significant revisions and improvements to this paper. I appreciate their attention to the reviewer comments, but still have the following concerns/suggestions, warranting minor revision:_**

We thank the referee for the positive assessment of our manuscript and the affirmative view of our first revision. We describe our responses and the associated efforts below.

**_The authors still do not provide any results showing what areas are frozen/unfrozen. It would be helpful to include, at least in supplemental, some information to help the reader visualize freezing and thawing patterns on the deltas and how this is or is not related to ramp development._**

We have added the thaw-depth plots of two simulations (corresponding to those shown in Figures 4b and 4f) at time steps 4970 to 4973 (out of 5000, not counting the 300-step ramp-up phase). These time steps correspond to the step before the final maximum winter ice cover, through to two time steps afterwards. Due to the lack of "permafrost" labels in the sediment pixels in ArcDelRCM.jl (as described in Sect. 2.2.2 and 2.2.3 of the manuscript), thaw-depth values are the closest way to display the frozen/thawed pattern of the delta bed. The new figure (C1) is included in a new Appendix C, which also includes a short text description of what is shown and its purpose.

References to the new Appendix C and its figure (C1) are added in the main text on lines 546-548:

> "As a pair of examples, Figure C1 shows the spatial views of the thaw-depth pattern corresponding to simulations shown in Figures 4b and 4f. These thaw-depth

patterns provide an indication on where the aforementioned processes and balances are active."

And line 389:

"A visual representation of the thaw-depth patterns corresponding to Figures 4b and 4f is shown in Appendix C."

***I appreciate the explanation of how bank erosion is treated, showing that their model requires sediment to be thawed in order to be mechanically eroded and that thaw proceeds in the next time step, allowing more cells to be eroded. The modifications to the text in this regard are adequate. Related, however, the reviewers in their response file suggest there is some 'alternate' definition of permafrost they intended (e.g., seasonally frozen ground). Permafrost, by definition, is at a temperature of <= 0C for at least two years. If the authors do not intend to model or describe permafrost but only frozen vs. unfrozen, then the authors should refer only to frozen ground rather than permafrost to avoid confusion.***

Nearly all of the usage of the term "permafrost" in the manuscript refers to general descriptions of the landscapes or settings of Arctic deltas, or the "permafrost" label and its associated erosional rules in the DeltaRCM-Arctic model. We have already been using the "frozen"/"thawed" terminology when discussing the simulation rules used in ArcDelRCM.jl. Nevertheless, we have identified a few remaining places where the referee's suggestion is applicable, and made the changes described below.

We have added the following sentence to the end of Sect. 2.2.2:

'To avoid confusion, we will use the terms "frozen" or "thawed" (as opposed to "permafrost") in the context of the erosional rules in ArcDelRCM.jl.'

The title of Sect. 2.2.3 has also been changed from "Permafrost Erosion" to "Erosion of Frozen Ground". To the last sentence of the first paragraph of Sect. 2.2.3, we added the following clarifying sub-clause:

> "[...], and consider only the depth of the boundary between frozen and thawed grounds."

***The authors did not actually address my comment regarding how channel networks were extracted. They simply state that graphs were extracted based on the locations of the channels, but determining the locations of the channels was my actual concern, as this is also somewhat subjective and rather difficult! Did you use a velocity threshold? The topography? A wet-dry map? Some combination? Based on Figure 3, it looks like you used bed elevation. Was it a simple threshold? How was the threshold selected?***

We have added clarifying sentences to where we describe the graph-extraction process in Sect. 2.3, on lines 336-339:

> 'In this context, channel pixels are defined as those having a water depth of 0.1 m or over, which is the threshold value for "dry"/"wet" pixel labels used in DeltaRCM (Liang et al., 2015b) and inherited by both DeltaRCM-Arcticand ArcDelRCM.jl. "Open-ocean" pixels without any depositions (i.e., with depth equalling the ocean-basin depth, $h_B$) are excluded.'

***I appreciate the authors' attempt to include more justification on the Lena delta modeling, but more information would be appreciated in the manuscript text. Even including the information written in the response to reviewer file (the non-quoted part) would help. This would also allow you to elaborate in the text on whether you think these results, in terms of the processes, are applicable to other Arctic deltas or not, which I suggest you include.***

As the referee suggested, we have added the following elaboration (based on the non-quoted part of the previous response to reviewer) to lines 82-87 in the Introduction:

> "Aside from being the largest Arctic delta, Lena Delta is chosen for the real-world example because of its fit to the modelled geometry. Many of the other Arctic river mouths (e.g., Ob, Yenisei, Mackenzie) are estuarine or geologically confined, and thus match the modelled geometry poorly, which may confound the analyses herein. Other Arctic deltas such as Olenyok and Colville are much smaller, and therefore more likely to be influenced in their growth by processes other than those modelled here (we discuss some of these processes at the end of Sect. 4). Through the exemplary case of the Lena Delta, [...]"

This, in combination with the existing discussion on the model limitations at the end of Sect. 4, should provide readers with a sense on the extent of extrapolation that can be safely applied to our results.

***Re: ice remaining on the ramp longer - My suggestion was not that the ice remaining there was contrived, but rather that if you force the ice to stay on the ramp longer, and then claim that the fact that the ice stays there longer is responsible for preserving/forming the ramp, then this seems contrived because you created a rule that causes the ice to stay there and a rule that causes the ice to shield the ramp. Nonetheless, I think the modifications to the text sufficiently address this issue and we can consider it resolved.***

***Figure 6-7, I appreciate having this in the manuscript but the box and whisker plots are not visible at this scale. Removing the points for individual runs may help, or rearrange the figure/make it larger.***

The size of the black dots, representing the individual realisations of the simulations, have been reduced in Figures 6 and 7. The box and whisker plots should now be visible despite the black dots.

We thank the referee again for the affirmative assessment, the time and effort spent on twice-reviewing our manuscript, and the constructive suggestions.

**Other Changes**

Aside from the changes responding to Referee #2's comments, we have also made the following pair of minor changes:

- We have added one recent reference, Rantanen et al. (2022) to the sentence about amplified warming in the Arctic (on line 29).
- We have also modified the capitalisation of the title of Appendix B to "title case".
- We have moved the hosting location of the ArcDelRCM.jl source code from the first author's personal GitLab instance to an institutional instance hosted at the GeoForschungsZentrum German Research Centre for Geosciences (GFZ). The web link has been modified accordingly.